# TOWARDS GENERAL-PURPOSE MODEL-FREE REINFORCEMENT LEARNING

**Scott Fujimoto, Pierluca D'Oro, Amy Zhang, Yuandong Tian, Michael Rabbat**
Meta FAIR

## ABSTRACT

Reinforcement learning (RL) promises a framework for near-universal problem-solving. In practice however, RL algorithms are often tailored to specific benchmarks, relying on carefully tuned hyperparameters and algorithmic choices. Recently, powerful model-based RL methods have shown impressive general results across benchmarks but come at the cost of increased complexity and slow run times, limiting their broader applicability. In this paper, we attempt to find a unifying model-free deep RL algorithm that can address a diverse class of domains and problem settings. To achieve this, we leverage model-based representations that approximately linearize the value function, taking advantage of the denser task objectives used by model-based RL while avoiding the costs associated with planning or simulated trajectories. We evaluate our algorithm, MR.Q, on a variety of common RL benchmarks with a single set of hyperparameters and show a competitive performance against domain-specific and general baselines, providing a concrete step towards building general-purpose model-free deep RL algorithms.

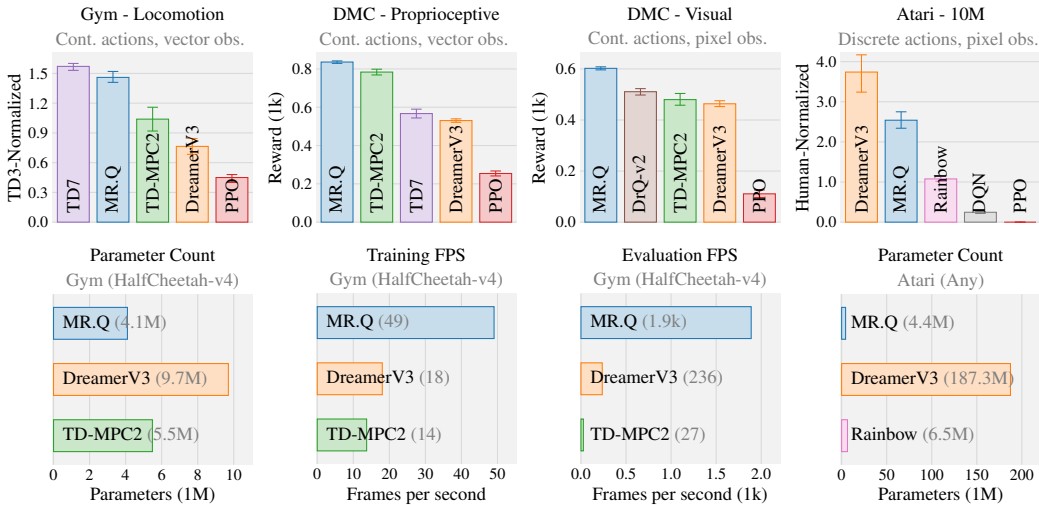

Figure 1: **Summary of results.** Aggregate mean performance across four common RL benchmarks and 118 environments featuring diverse characteristics (e.g., observation and action spaces, task types). Error bars capture a 95% stratified bootstrap confidence interval. Our algorithm, MR.Q, achieves a competitive performance against both state-of-the-art domain-specific and general baselines, while using a single set of hyperparameters. Notably, MR.Q accomplishes this with fewer network parameters and substantially faster training and evaluation speeds than general-purpose model-based methods.

## 1 INTRODUCTION

The conceptual premise of RL is inherently general-purpose—an RL agent can learn optimal behavior with only two basic elements: a well-defined objective and data describing its interactions with

---

Correspondence: sfujimoto@meta.com. Code: https://github.com/facebookresearch/MRQ.

the environment. In reality, however, most RL algorithms are anything but general-purpose. Instead, RL algorithms are highly specialized and typically characterized by specific problem classes, such as discrete versus continuous actions or vector versus pixel observations, with each category requiring its own set of algorithmic choices and hyperparameters. For example, Rainbow and TD3 (Hessel et al., 2018; Fujimoto et al., 2018), common methods for Atari and MuJoCo respectively (Bellemare et al., 2013; Todorov et al., 2012), have more differences than similarities in their shared hyperparameters (Table 1)—without accounting for further algorithmic differences.

Table 1: Hyperparameter differences between Rainbow (Hessel et al., 2018) and TD3 (Fujimoto et al., 2018). TD3 uses an expected moving average (EMA) update with an effective frequency of $\frac{1}{1-0.995} = 200$.

| Hyperparameter | Rainbow | TD3 |
|---|---|---|
| Discount factor | 0.99 | 0.99 |
| Optimizer | Adam | Adam |
| Learning Rate | $6.25 \cdot 10^{-5}$ | $10^{-3}$ |
| Adam $\epsilon$ | $1.5 \cdot 10^{-4}$ | $10^{-8}$ |
| Replay buffer size | 1M | 1M |
| Minibatch size | 32 | 100 |
| Target network update | Iterative | EMA |
| Effective target update freq. | 8k | 200 |
| Initial random steps | 20k | 1k |

To some extent, general-purpose algorithms do exist—policy gradient methods (Williams, 1992; Schulman et al., 2015; 2017) and many evolutionary approaches (Rechenberg, 1978; Back, 1996; Rubinstein, 1997; Salimans et al., 2017) require few assumptions on the underlying problem. Unfortunately, these methods often offer poor sample efficiency and asymptotic performance compared more domain-specific approaches, and in some instances, can require extensive re-tuning over numerous implementation-level details (Engstrom et al., 2020; Huang et al., 2022).

Recently, DreamerV3 (Hafner et al., 2023) and TD-MPC2 (Hansen et al., 2024), have showcased the potential of general-purpose model-based approaches, achieving impressive single-task performance on a diverse set of benchmarks without re-tuning hyperparameters. However, despite their success, model-based methods also introduce substantial algorithmic and computational complexity, making them less practical than lightweight domain-specific model-free algorithms.

This paper presents a general model-free RL algorithm that leverages model-based representations to achieve the sample efficiency and performance of model-based methods, without the computational overhead. A recent surge of high-performing model-free RL algorithms with dynamics-based representations (Guo et al., 2020; 2022; Schwarzer et al., 2020; 2023; Zhao et al., 2023; Fujimoto et al., 2024; Zheng et al., 2024; Scannell et al., 2024) has showcased the potential of this family of algorithms when tailored for a single benchmark. Recognizing the similarity between these model-based and model-free approaches, our hypothesis is that the true benefit of model-based objectives is in the implicitly learned representation, rather than the model itself, and thus prompting the question:

*Can model-based representations alone enable sample-efficient general-purpose learning?*

Our proposed approach is based on learning features that approximately capture a linear relationship between state-action pairs and value. To do so, we draw heavily from modern dynamics-based representation learning methods (see Related Work) as well as the work of Parr et al. (2008), who show that both model-based and model-free objectives converge to the same solution in linear space. By mapping states and actions into a single, unified embedding, we eliminate any environment-specific characteristics of the input space and allow for a standardized set of hyperparameters.

We evaluate our method, MR.Q, on four widely used RL benchmarks and 118 environments, and achieve competitive performance against state-of-the-art domain-specific and general baselines without algorithmic or hyperparameter changes between environments or benchmarks.

## 2 RELATED WORK

**General-purpose RL.** Although many traditional RL methods are general-purpose in principle, practical constraints often force assumptions about the task domain. For example, algorithms like Q-learning and SARSA (Watkins, 1989; Rummery & Niranjan, 1994) can be conceptually extended to continuous spaces, but are typically implemented using discrete lookup tables. In practice, early examples of general decision-making approaches can be found in on-policy methods with function approximation. For instance, both evolutionary algorithms (Rechenberg, 1978; Back, 1996; Rubinstein, 1997; Salimans et al., 2017) and policy gradient methods (Williams, 1992; Sutton et al., 1999;

Schulman et al., 2015; 2017) offer update rules with convergence guarantees and independence to the input space. However, despite their generality, these methods are also hindered by poor sample efficiency and are prone to local minima, limiting their suitability for many practical applications.

In contrast, the design of deep RL algorithms tends to favor more specialized approaches that align closely with a single benchmark—e.g., DQN↔Atari (Bellemare et al., 2013; Mnih et al., 2015), DDPG↔MuJoCo (Todorov et al., 2012; Lillicrap et al., 2015), or AlphaGo↔Go (Silver et al., 2016). Generalizing beyond these initial benchmarks can often require significant engineering, tuning, or algorithmic discovery (Luong et al., 2019; Schrittwieser et al., 2020; Haydari & Yılmaz, 2020; Ibarz et al., 2021). In imitation learning, GATO achieved generalist behavior, but relied on large expert datasets (Reed et al., 2022). Recently, DreamerV3 (Hafner et al., 2023) demonstrated a strong capability over many benchmarks without re-tuning, but used costly large models and simulated rollouts. Our objective is to discover a lightweight model-free approach to general-purpose learning.

**Dynamics-based representation learning.** Building representations from system dynamics is a long-standing approach for adaptation, partial observability, and feature selection (Dayan, 1993; Littman & Sutton, 2001; Parr et al., 2008). Numerous model-free methods have been developed to learn representations by predicting future latent states (Munk et al., 2016; Van Hoof et al., 2016; Zhang et al., 2018; Gelada et al., 2019; Lee et al., 2020; Guo et al., 2020; 2022; Schwarzer et al., 2020; 2023; Zintgraf et al., 2021; Yu et al., 2021; 2022; Fujimoto et al., 2021; 2024; McInroe et al., 2021; Seo et al., 2022; Kim et al., 2022; Tang et al., 2023; Zhao et al., 2023; Zheng et al., 2024; Ni et al., 2024; Scannell et al., 2024). Unsurprisingly, these model-free approaches closely relate to model-based counterparts which learn a latent dynamics model for planning or value estimation (Watter et al., 2015; Finn et al., 2016; Karl et al., 2017; Ha & Schmidhuber, 2018; Schrittwieser et al., 2020; 2021; Ye et al., 2021; Hansen et al., 2022; 2024; Hafner et al., 2019; 2023; Wang et al., 2024). Our approach, MR.Q, is most closely related to the state-action representation learning in TD7 (Fujimoto et al., 2024). At a high level, MR.Q differs from TD7 by discarding the original input and including losses over the reward and termination. MR.Q also differs significantly in implementation, drawing inspiration from prior work to determine a set of design choices that performs well across benchmarks, including multi-step returns, unrolled dynamics, and categorical losses.

Our motivation also relates to linear MDPs (Jin et al., 2020; Agarwal et al., 2020) and linear spectral representation (Ren et al., 2022; 2023; Zhang et al., 2022; Shribak et al., 2024). The latter aims to learn a low-rank decomposition of the transition dynamics of the MDP and recover a linear relationship between an embedding and the value function. Similarly, our work connects to two-stage linear RL, where a non-linear embedding is learned for linear RL (Levine et al., 2017; Chung et al., 2019).

**State abstraction.** Our work is closely related to bisimulation metrics (Ferns et al., 2004; 2011; Castro, 2020) and MDP homomorphisms (Ravindran, 2004; van der Pol et al., 2020a;b; Rezaei-Shoshtari et al., 2022) which rely on measures of similarity in reward and dynamics for state or action abstraction. These concepts have inspired practical approximations to bisimulation metrics as a means of shaping representations in deep RL agents, particularly those using image-based observations (Zhang et al., 2020; Castro et al., 2021; Zang et al., 2022).

## 3  BACKGROUND

Reinforcement learning (RL) problems are described by a Markov Decision Process (MDP) (Bellman, 1957), which we define by a tuple $(S, A, p, R, \gamma)$ of state space $S$, action space $A$, dynamics function $p$, reward function $R$ and discount factor $\gamma$. Value-based RL methods learn a value function $Q^\pi(s, a) := \mathbb{E}_\pi[\sum_{t=0}^\infty \gamma^t r_t | s_0 = s, a_0 = a]$ that models the expected discounted sum of rewards $r_t \sim R(s_t, a_t)$ by following a policy $\pi$ which maps states $s$ to actions $a$.

The true value function $Q^\pi$ is estimated by an approximate value function $Q_\theta$. We use subscripts to indicate the network parameters $\theta$. Target networks, which are used to introduce stationarity in prediction targets, have parameters denoted by an apostrophe, e.g., $Q_{\theta'}$. These parameters are periodically synchronized with the current network parameters ($\theta' \leftarrow \theta$).

## 4  MODEL-BASED REPRESENTATIONS FOR Q-LEARNING

This section presents the MR.Q algorithm (Model-based Representations for Q-learning), a model-free RL algorithm that learns an approximately linear representation of the value function through

model-based objectives. Value-based RL algorithms learn a value function $Q$ that maps state-action pairs $(s, a)$ to values in $\mathbb{R}$ and a policy $\pi$ that maps states $s$ to actions $a$. Like many representation learning methods for RL, MR.Q adds an initial step that transforms states and state-action pairs into embeddings $\mathbf{z}_s$ and $\mathbf{z}_{sa}$, which serves as inputs to the downstream policy and value function.

$$f_\omega : s \to \mathbf{z}_s, \qquad\qquad g_\omega : (s, a) \to \mathbf{z}_{sa}, \qquad\qquad (1)$$

$$\pi_\phi : \mathbf{z}_s \to a, \qquad\qquad Q_\theta : \mathbf{z}_{sa} \to \mathbb{R}. \qquad\qquad (2)$$

While neither the value function nor policy require explicit representation learning, using intermediate embeddings has two main benefits:

1. Introducing an explicit representation learning stage can enable richer alternative learning signals that are grounded in the dynamics and rewards of the MDP, as opposed to relying exclusively on non-stationary value targets used in both value and policy learning.
2. Representation learning can transform the input into a unified, abstract space that is decoupled from the original input characteristics, e.g., images or action spaces. This abstraction allows us to filter irrelevant or spurious details and use unified downstream architectures, improving robustness to environment variations.

To learn these embeddings, we draw inspiration from linear feature selection, revisiting the work of Parr et al. (2008), as well as MDP homomorphisms (Ravindran & Barto, 2002). In Section 4.1 we highlight how model-based objectives can be used to learn features that share an approximately linear relationship with the true value function. Then in Section 4.2, we relax our theoretical motivation for a practical algorithm based on recent advances in dynamics-based representation learning.

## 4.1 THEORETICAL MOTIVATION

Consider a linear decomposition of the value function, where the value function $Q(s, a)$ is represented by features $\mathbf{z}_{sa}$ and linear weights $\mathbf{w}$:

$$Q(s, a) = \mathbf{z}_{sa}^\top \mathbf{w}. \qquad\qquad (3)$$

Our primary objective is to learn features $\mathbf{z}_{sa}$ that share an approximately linear relationship with the true value function $Q^\pi$. However, since this relationship is only approximate, we use these features as input to a non-linear function $\hat{Q}(\mathbf{z}_{sa})$, rather than relying solely on linear function approximation.

We start by exploring how to find features that can linearly represent the true value function. Given a dataset $D$ of tuples $(s, a, r, s', a')$, we consider two possible approaches for learning a value function $Q$: A model-free update based on semi-gradient TD (Sutton, 1988; Sutton & Barto, 1998):

$$\mathbf{w} \leftarrow \mathbf{w} - \alpha \mathbb{E}_D \left[ \nabla_\mathbf{w} \left( \mathbf{z}_{sa}^\top \mathbf{w} - |r + \gamma \mathbf{z}_{s'a'}^\top \mathbf{w}|_{\text{sg}} \right)^2 \right]. \qquad\qquad (4)$$

A model-based approach to learn $\mathbf{w}_{\text{mb}}$, based on rolling out estimates of the dynamics and reward:

$$\mathbf{w}_{\text{mb}} := \sum_{t=0}^\infty \gamma^t W_p^t \mathbf{w}_r, \qquad\qquad (5)$$

$$\mathbf{w}_r := \underset{\mathbf{w}}{\arg\min} \, \mathbb{E}_D \left[ \left( \mathbf{z}_{sa}^\top \mathbf{w} - r \right)^2 \right], \qquad W_p := \underset{W}{\arg\min} \, \mathbb{E}_D \left[ \left( \mathbf{z}_{sa}^\top W - \mathbf{z}_{s'a'} \right)^2 \right]. \qquad (6)$$

Closely following Parr et al. (2008) and Song et al. (2016), we can show that these approaches converge to the same solution (proofs for this section can be found in Appendix A).

**Theorem 1.** *The fixed point of the model-free approach (Equation 4) and the solution of the model-based approach (Equation 5) are the same.*

From the insight of Theorem 1, we can connect the value error VE, the difference between an approximate value function $Q$ and the true value function $Q^\pi$,

$$\text{VE}(s, a) := Q(s, a) - Q^\pi(s, a) \qquad\qquad (7)$$

to the accuracy of reward and dynamics components of the estimated model (Theorem 2).

**Theorem 2.** *The value error of the solution described by Theorem 1 is bounded by the accuracy of the estimated dynamics and reward:*

$$|\text{VE}(s, a)| \leq \frac{1}{1 - \gamma} \max_{(s,a) \in S \times A} \left( \left| \mathbf{z}_{sa}^\top \mathbf{w}_r - \mathbb{E}_{r|s,a}[r] \right| + \max_i |\mathbf{w}_i| \sum \left| \mathbf{z}_{sa}^\top W_p - \mathbb{E}_{s',a'|s,a}[\mathbf{z}_{s'a'}] \right| \right). \qquad (8)$$

Parr et al. (2008) and Song et al. (2016) use a related insight regarding the Bellman error to infer an approach for feature selection. However, with the advent of deep learning, we can instead directly learn the features $\mathbf{z}_{sa}$ by jointly optimizing them alongside the linear weights $\mathbf{w}_r$ and $W_p$. This is accomplished by treating the features and linear weights as a unified end-to-end model and balancing the losses in Equation 6 with a hyperparameter $\lambda$:

$$\mathcal{L}(\mathbf{z}_{sa}, \mathbf{w}_r, W_p) = \underbrace{\mathbb{E}_D\left[\left(\mathbf{z}_{sa}^\top \mathbf{w}_r - r\right)^2\right]}_{\text{Reward learning}} + \underbrace{\lambda \mathbb{E}_D\left[\left(\mathbf{z}_{sa}^\top W_p - \mathbf{z}_{s'a'}\right)^2\right]}_{\text{Dynamics learning}}. \tag{9}$$

However, the resulting Equation 9 has some notable drawbacks.

**Dependency on $\pi$.** The dynamics target $\mathbf{z}_{s'a'}$ depends on an action $a'$ determined by the policy $\pi$. In policy optimization problems, this introduces non-stationarity, where the target embedding must be continually updated to reflect changes in the policy. This creates an undesirable interdependence between the policy and encoder.

**Undesirable local minima.** Jointly optimizing both the features $\mathbf{z}_{sa}$ and the dynamics target can lead to undesirable local minima, similar to the issues encountered with Bellman residual minimization (Baird, 1995; Fujimoto et al., 2022). This can result in collapsed or trivial solutions when the dataset does not fully cover the state and action space or when the reward is sparse.

To address these issues, we suggest relaxations on our proposed, theoretically grounded approach:

$$\mathcal{L}(\mathbf{z}_{sa}, \mathbf{w}_r, W_p) = \mathbb{E}_D\left[\left(\mathbf{z}_{sa}^\top \mathbf{w}_r - r\right)^2\right] + \lambda \mathbb{E}_D\left[\left(\mathbf{z}_{sa}^\top W_p - \underbrace{\bar{\mathbf{z}}_{s'}}_{\text{Adjustment}}\right)^2\right]. \tag{10}$$

We propose two key modifications to alleviate the aforementioned issues. Firstly, we use a state-dependent embedding $\mathbf{z}_{s'}$ as the dynamics target, rather than the state-action embedding $\mathbf{z}_{s'a'}$. This eliminates any dependency on the current policy while still capturing the environment's dynamics.

Secondly, to mitigate the issue of local minima, we use a target network $f_{\omega'}(s')$ to generate the dynamics target $\bar{\mathbf{z}}_{s'}$, where the parameters $\omega'$ are periodically updated to track the current network parameters $\omega$. Empirical evidence from prior work suggests that this approach can yield significant performance gains (Grill et al. (2020); Assran et al. (2023), see Related Work), although it no longer guarantees convergence to a fixed point.

Due to these two changes, even if the modified objective defined by Equation 10 is minimized, we can no longer assume there is a *linear* relationship between the embedding $\mathbf{z}_{sa}$ and the value function. However, we can instead allow for a *non-linear* relationship, replacing linear weights $\mathbf{w}$ with a non-linear function $\hat{Q}(\mathbf{z}_{sa})$. We can show that this relationship exists as long as the features are sufficiently rich (i.e., such that a MDP homomorphism is satisfied (Ravindran & Barto, 2002)).

**Theorem 3.** *Given functions $f(s) = \mathbf{z}_s$ and $g(\mathbf{z}_s, a) = \mathbf{z}_{sa}$, then if there exists functions $\hat{p}$ and $\hat{R}$ such that for all $(s, a) \in S \times A$:*

$$\mathbb{E}_{\hat{R}}[\hat{R}(\mathbf{z}_{sa})] = \mathbb{E}_R\left[R(s, a)\right], \qquad \hat{p}(\mathbf{z}_{s'}|\mathbf{z}_{sa}) = \sum_{\hat{s}:\mathbf{z}_{\hat{s}}=\mathbf{z}_{s'}} p(\hat{s}|s, a), \tag{11}$$

*then for any policy $\pi$ where there exists a corresponding policy $\hat{\pi}(a|\mathbf{z}_s) = \pi(a|s)$, there exists a function $\hat{Q}$ equal to the true value function $Q^\pi$ over all possible state-action pairs $(s, a) \in S \times A$:*

$$\hat{Q}(\mathbf{z}_{sa}) = Q^\pi(s, a). \tag{12}$$

*Furthermore, Equation 11 guarantees the existence of an optimal policy $\hat{\pi}^*(a|\mathbf{z}_s) = \pi^*(a|s)$.*

Consequently, even if the features $\mathbf{z}_{sa}$ do not linearly represent the true value function, i.e., the loss in Equation 9 cannot be not exactly minimized, $\mathbf{z}_{sa}$ can still be used in a non-linear relationship to represent the value function. Furthermore, Theorem 3 outlines a similar objective as the original linear objective defined in Equation 9, in learning the reward and dynamics of the MDP.

These results motivates the practical algorithm discussed in the following section. Using the adjusted loss defined in Equation 10, we will aim to learn features with an approximately linear relationship to the true value function, but use a non-linear value function with those features to account for the error induced by our approximations.

### 4.2 ALGORITHM

We now present the details of MR.Q (Model-based Representations for Q-learning). Building on the insights from the previous section, our key idea is to learn a state-action embedding $\mathbf{z}_{sa}$ that is approximately linear with the true value function $Q^\pi$. To account for approximation errors, these features are used with *non-linear* function approximation to determine the value.

The state embedding vector $\mathbf{z}_s$ is obtained as an intermediate component by training end-to-end with the state-action encoder. MR.Q handles different input modalities by swapping the architecture of the state encoder. Since $\mathbf{z}_s$ is a vector, the remaining networks are independent of the observation space and use feedforward networks.

Given the transition $(s, a, r, d, s')$ from the replay buffer:

| **Output** MR.Q | |
|---|---|
| *Trained end-to-end* | |
| State Encoder | $\mathbf{z}_s = f_\omega(s)$ |
| State-Action Encoder | $\mathbf{z}_{sa} = g_\omega(\mathbf{z}_s, a)$ |
| MDP predictor | $\tilde{\mathbf{z}}_{s'}, \tilde{r}, \tilde{d} = \mathbf{z}_{sa}^\top \mathbf{m}$ |
| *Decoupled RL* | |
| Value | $\tilde{Q}_i = Q_\theta(\mathbf{z}_{sa})$ |
| Policy | $a_\pi = \pi_\phi(\mathbf{z}_s)$ |

| **Update** MR.Q |
|---|
| **if** $t \mathbin{\%} T_{\text{target}} = 0$ **then** |
|     Target networks: $\theta', \phi', \omega' \leftarrow \theta, \phi, \omega$. |
|     Reward scaling: $\bar{r}' \leftarrow \bar{r}, \bar{r} \leftarrow \text{mean}_D r$. |
|     **for** $T_{\text{target}}$ time steps **do** |
|         Encoder update: Equation 14. |
| Value update: Equation 19. |
| Policy update: Equation 20. |

The encoder loss is composed of three terms based on the reward, dynamics and terminal signal that are unrolled over a short horizon. The value function and policy are trained independently, using standard losses (Silver et al., 2014; Fujimoto et al., 2018). We use LAP (Fujimoto et al., 2020) to sample transitions with priority according to their TD errors (Schaul et al., 2016), the absolute difference between the predicted value and the target value in Equation 19.

The target network, reward scaling (defined in Equation 19), and the encoder are updated periodically every $T_{\text{target}}$ time steps. This synchronized update schedule keeps the input and target output fixed for the downstream value function and policy within each iteration, thus reducing non-stationarity in the optimization (Fujimoto et al., 2024).

#### 4.2.1 ENCODER

The encoder loss is based on unrolling the dynamics of the learned model over a short horizon. Given a subsequence of an episode $(s_0, a_0, r_1, d_1, s_1, ..., r_{H_{\text{Enc}}}, d_{H_{\text{Enc}}}, s_{H_{\text{Enc}}})$, the model is unrolled by encoding the initial state $s_0$, then by repeatedly applying the state-action encoder $g_\omega$ and linear MDP predictor $\mathbf{m}$:

$$\tilde{\mathbf{z}}^t, \tilde{r}^t, \tilde{d}^t := g_\omega(\tilde{\mathbf{z}}^{t-1}, a^{t-1})^\top \mathbf{m}, \quad \text{where } \tilde{\mathbf{z}}^0 := f_\omega(s_0). \tag{13}$$

The final loss is summed over the unrolled model and balanced by corresponding hyperparameters:

$$\mathcal{L}_{\text{Encoder}}(f, g, \mathbf{m}) := \sum_{t=1}^{H_{\text{Enc}}} \lambda_{\text{Reward}} \mathcal{L}_{\text{Reward}}(\tilde{r}^t) + \lambda_{\text{Dynamics}} \mathcal{L}_{\text{Dynamics}}(\tilde{\mathbf{z}}_{s'}^t) + \lambda_{\text{Terminal}} \mathcal{L}_{\text{Terminal}}(\tilde{d}^t). \tag{14}$$

$\lambda_{\text{Terminal}}$ is set to 0 until the first terminal transition (i.e., $d = 0$) is viewed. This approach is commonly used in model-based RL (Oh et al., 2015; Hafner et al., 2023; Hansen et al., 2024), as well as dynamics-based representation learning (Schwarzer et al., 2020; 2023; Scannell et al., 2024).

**Reward loss.** While our theoretical analysis suggests using the mean-squared error to train the predicted reward, we find that a categorical representation of the reward is more effective in practice for predicting sparse rewards and is robust to reward magnitude. This empirical benefit is consistent with prior work (Schrittwieser et al., 2020; Hafner et al., 2023; Hansen et al., 2024; Wang et al., 2024). Our reward loss function uses the cross entropy CE between the predicted reward $\tilde{r}$ and a two-hot encoding of the reward $r$:

$$\mathcal{L}_{\text{Reward}}(\tilde{r}) := \text{CE}\left(\tilde{r}, \text{Two-Hot}(r)\right). \tag{15}$$

To handle a wide range of reward magnitudes without prior knowledge, the locations of the two-hot encoding are spaced at increasing non-uniform intervals, according to $\mathrm{symexp}(x) = \mathrm{sign}(x)(\exp(x) - 1)$ (Hafner et al., 2023).

**Dynamics loss.** The dynamics loss minimizes the mean-squared error between the predicted next state embedding $\tilde{\mathbf{z}}_{s'}$ and the next state embedding $\bar{\mathbf{z}}_{s'}$ from the target encoder $f_{\omega'}$:

$$\mathcal{L}_{\text{Dynamics}}(\tilde{\mathbf{z}}_{s'}) := (\tilde{\mathbf{z}}_{s'} - \bar{\mathbf{z}}_{s'})^2 . \tag{16}$$

As discussed in the previous section, using the next state embedding $\mathbf{z}_{s'}$ eliminates the dependency on the policy that would occur when using a state-action embedding target.

**Terminal loss.** The predicted scalar terminal signal $\tilde{d}$ is trained simply using a MSE loss with the binary terminal signal $d$:

$$\mathcal{L}_{\text{Terminal}}(\tilde{d}) := (\tilde{d} - d)^2. \tag{17}$$

### 4.2.2 VALUE FUNCTION

Value learning is primarily based on TD3 (Fujimoto et al., 2018). Specifically, we train two value functions and take the minimum output between their respective target networks to determine the value target. Similar to TD3, the target action is determined by the target policy $\pi_{\phi'}$, perturbed by small amount of clipped Gaussian noise:

$$a_\pi = \begin{cases} \mathrm{argmax}\, a' & \text{for discrete } A, \\ \mathrm{clip}(a', -1, 1) & \text{for continuous } A, \end{cases} \quad \text{where } a' = \pi_{\phi'}(s') + \mathrm{clip}(\epsilon, -c, c), \quad \epsilon \sim \mathcal{N}(0, \sigma^2). \tag{18}$$

Discrete actions are represented by a one-hot encoding, where the Gaussian noise is added to each dimension. Action noise and the clipping is scaled according the range of the action space.

We modify the TD3 loss in a few ways. Firstly, following numerous prior work across benchmarks (Hessel et al., 2018; Barth-Maron et al., 2018; Yarats et al., 2022; Schwarzer et al., 2023), we predict multi-step returns over a horizon $H_Q$. Secondly, we use the Huber loss instead of mean-squared error to eliminate bias from prioritized sampling (Fujimoto et al., 2020). Finally, the target value is normalized according to the average absolute reward $\bar{r}$ in the replay buffer:

$$\mathcal{L}_{\text{Value}}(\tilde{Q}_i) := \mathrm{Huber}\left(\tilde{Q}_i, \frac{1}{\bar{r}}\left(\sum_{t=0}^{H_Q-1} \gamma^t r_t + \gamma^{H_Q}\tilde{Q}_j'\right)\right), \quad \tilde{Q}_j' := \bar{r}' \min_{j=1,2} Q_{\theta_j'}(\mathbf{z}_{s_{H_Q}a_{H_Q,\pi}}). \tag{19}$$

The value $\bar{r}'$ captures the *target* average absolute reward, which is the scaling factor used to the most recently copied value functions $Q_{\theta_j'}$. This value is updated simultaneously with the target networks $\bar{r}' \leftarrow \bar{r}$. Maintaining a consistent reward scale keeps the loss magnitude constant across different benchmarks, thus improving the robustness of a single set of hyperparameters.

### 4.2.3 POLICY

For both continuous and discrete action spaces, the policy is updated using the deterministic policy gradient (Silver et al., 2014):

$$\mathcal{L}_{\text{Policy}}(a_\pi) := -0.5 \sum_{i=\{1,2\}} \tilde{Q}_i(\mathbf{z}_{sa_\pi}) + \lambda_{\text{pre-activ}}\mathbf{z}_\pi^2, \quad \text{where } a_\pi = \mathrm{activ}(\mathbf{z}_\pi). \tag{20}$$

To make the loss universal between action spaces, we use Gumbel-Softmax (Jang et al., 2017; Lowe et al., 2017; Cianflone et al., 2019) for discrete actions, and Tanh for continuous actions. A small regularization penalty is added to the square of the pre-activations $\mathbf{z}_\pi$ before the policy's final activation to help avoid local minima when the reward, and value, is sparse (Bjorck et al., 2021).

For exploration, Gaussian noise is added to each dimension of the action (or one-hot encoding of the action). Similar to Equation 18, the resulting action vector is clipped to the range of the action space for continuous actions. For discrete actions, the final action is determined by the $\mathrm{argmax}$ operation.

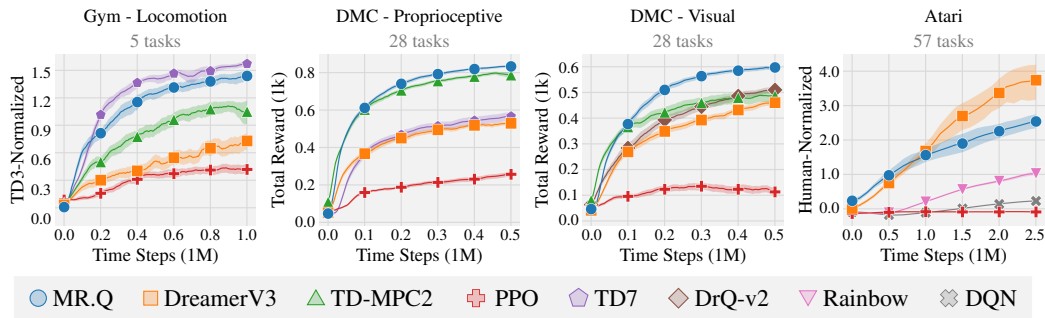

Figure 2: **Aggregate learning curves.** Average performance over each benchmark. Results are over 10 seeds. The shaded area captures a 95% stratified bootstrap confidence interval. Due to action repeat, 500k time steps in DMC correspond to 1M frames in the original environment and 2.5M time steps in Atari corresponds to 10M frames in the original environment.

## 5 EXPERIMENTS

We evaluate MR.Q on four popular RL benchmarks and 118 environments, and compare its performance against strong domain-specific baselines, general model-based approaches, DreamerV3 (Hafner et al., 2023) and TD-MPC2 (Hansen et al., 2024), and a general model-free algorithm, PPO (Schulman et al., 2017). Rather than establish MR.Q as the state-of-the-art approach in any particular benchmark, our objective is to demonstrate its broad applicability and effectiveness across a diverse set of tasks with a single set of hyperparameters. The baselines use author-suggested default hyperparameters and are fixed across environments. Additional details can be found in Appendix B.

### 5.1 MAIN RESULTS

Aggregate learning curves are displayed in Figure 2, with full results displayed in Appendix C.

**Gym - Locomotion.** This subset of the Gym benchmark (Brockman et al., 2016; Towers et al., 2024) considers 5 locomotion tasks in the MuJoCo simulator (Todorov et al., 2012) with continuous actions and low level states. Agents are trained for 1M time steps without any environment preprocessing. We evaluate against three baselines: TD7 (Fujimoto et al., 2024), a state-of-the-art (or near) approach for this benchmark, as well as TD-MPC2, DreamerV3, and PPO. To aggregate results, we normalize using the performance of TD3 (Fujimoto et al., 2018).

**DMC - Proprioceptive.** The DeepMind Control suite (DMC) (Tassa et al., 2018) is a collection of continuous control robotics tasks built on the MuJoCo simulator. These tasks use the proprioceptive states as the observation space, meaning that the input is a vector, and limit the total reward for each episode at 1000, making it easy to aggregate results. We report results on all 28 default tasks that were used by either TD-MPC2 or DreamerV3. Agents are trained for 500k time steps, equivalent to 1M frames in the original environment due to action repeat. For comparison, we evaluate against the same three algorithms as in the Gym benchmark, with TD-MPC2 considered state-of-the-art (or near) for this benchmark. We also include TD7 due to its strong performance in the Gym benchmark.

**DMC - Visual.** The visual DMC benchmark includes the same 28 tasks as the proprioceptive benchmark, but uses image-based observations instead. Agents are trained for 500k time steps. For baselines, we include DrQ-v2 (Yarats et al., 2022), given its state-of-the-art (or near) performance in model-free RL, alongside TD-MPC2, DreamerV3, and PPO.

**Atari.** The Atari benchmark is built on the Arcade Learning Environment (Bellemare et al., 2013). This benchmark uses pixel observations and discrete actions and includes the 57 games used by DreamerV3. We follow standard preprocessing steps, including sticky actions (Machado et al., 2018) (full details in Appendix B.3). Agents are trained for 2.5M time steps (equivalent to 10M frames), a setting which has been considered by prior work (Sokar et al., 2023). For comparison, we evaluate against three baselines: the model-based approach DreamerV3, as well as model-free approaches, DQN (Mnih et al., 2015), Rainbow (Hessel et al., 2018), and PPO. Results are aggregated by normalizing scores against human performance.

**Discussion.** Throughout our experiments, we find the presence of "no free lunch", where the top-performing baseline in one benchmark fails to replicate its success in another. Regardless, MR.Q achieves the highest performance in both DMC benchmarks, showcasing its ability to handle different observation spaces. Although it falls slightly behind TD7 in the Gym benchmark, MR.Q is the strongest method overall across all continuous control benchmarks. In Atari, while DreamerV3 outperforms MR.Q, it relies on a model with 40 times more parameters and struggles comparatively in the remaining benchmarks. When compared to the model-free baselines, MR.Q surpasses PPO, DQN, and Rainbow, demonstrating its effectiveness with discrete action spaces.

## 5.2 DESIGN STUDY

To better understand the impact of certain design choices and hyperparameters, we attempt variations of MR.Q, and report the aggregate results in Table 2.

Table 2: **Design study.** Average difference in normalized performance from varying design choices across each benchmark over 5 seeds. Negative changes are highlighted lightly $[-0.01, -0.2)$. Damaging changes are highlighted moderately $[-0.2, -0.5)$. Catastrophic changes are highlighted boldly $(\leq -0.5)$. Positive changes are similarly highlighted $(> 0.01)$.

| Design | Gym - Locomotion TD3-Normalized | DMC - Proprioceptive Reward (1k) | DMC - Visual Reward (1k) | Atari - 1M Human-Normalized |
|---|---|---|---|---|
| | | Relaxations | | |
| Linear value function | -1.17 [-1.19, -1.15] | -0.58 [-0.59, -0.56] | -0.41 [-0.42, -0.39] | -1.35 [-1.41, -1.29] |
| Dynamics target | -0.10 [-0.17, -0.04] | -0.15 [-0.15, -0.15] | -0.05 [-0.05, -0.04] | -0.38 [-0.81, 0.05] |
| No target encoder | -0.53 [-0.60, -0.46] | -0.35 [-0.35, -0.34] | -0.15 [-0.15, -0.15] | -0.86 [-0.89, -0.83] |
| Revert | -1.47 [-1.54, -1.39] | -0.72 [-0.73, -0.72] | -0.52 [-0.52, -0.51] | -1.69 [-1.70, -1.67] |
| Non-linear model | -0.01 [-0.07, 0.03] | -0.00 [-0.02, 0.01] | -0.01 [-0.02, -0.00] | -0.07 [-0.32, 0.18] |
| | | Loss functions | | |
| MSE reward loss | 0.10 [-0.02, 0.19] | -0.06 [-0.08, -0.05] | -0.05 [-0.07, -0.04] | -0.79 [-0.86, -0.73] |
| No reward scaling | -0.04 [-0.09, 0.02] | -0.01 [-0.02, 0.00] | -0.00 [-0.01, 0.01] | 0.18 [-0.25, 0.56] |
| No min | -0.09 [-0.16, -0.01] | -0.01 [-0.02, 0.01] | 0.00 [-0.01, 0.01] | 0.13 [-0.10, 0.58] |
| No LAP | -0.10 [-0.24, -0.00] | 0.00 [-0.00, 0.01] | -0.01 [-0.02, -0.01] | -0.13 [-0.38, 0.14] |
| No MR | -0.56 [-0.69, -0.43] | -0.19 [-0.19, -0.18] | -0.07 [-0.09, -0.03] | -0.78 [-0.88, -0.69] |
| | | Horizons | | |
| 1-step return | -0.33 [-0.46, -0.21] | -0.04 [-0.05, -0.02] | -0.03 [-0.03, -0.02] | -0.70 [-0.81, -0.59] |
| No unroll | 0.07 [0.01, 0.14] | -0.01 [-0.01, -0.00] | -0.04 [-0.06, -0.01] | -0.33 [-0.41, -0.28] |

**Relaxations.** In Section 4.1, we outlined a loss (Equation 9) that, if globally minimized, would provide features that are linear with the true value function. MR.Q in practice relaxes this theoretical result by modifying the loss and using a non-linear value function. In **Linear value function**, we replace the non-linear value function with a linear function. In **Dynamics target**, we replace the state embedding dynamics target with a state-action embedding $\bar{\mathbf{z}}_{s'a'}$ determined from the target state-action encoder $g_\omega$. In **No target encoder**, we use the current encoder to generate the dynamics target $\mathbf{z}_{s'a'}$, and jointly optimize it within the encoder loss. In **Revert**, we consider all of the aforementioned changes simultaneously, using linear value functions and setting the dynamics target as a state-action embedding determined by the current encoder. In **Non-linear model**, we replace the linear MDP predictor with individual networks that predict each component separately from $\mathbf{z}_{sa}$.

**Loss functions.** MR.Q's loss functions use several unconventional choices. In **MSE reward loss**, we replace the categorical loss function on the predicted reward in Equation 15 with the mean-squared error (MSE). In **No reward scaling**, we remove the reward scaling in Equation 19, setting $\bar{r} = \bar{r}' = 1$. In **No min**, we take the mean over the target value functions instead of the minimum in Equation 19. In **No LAP**, we remove prioritized sampling (Fujimoto et al., 2020) and use the MSE instead of the Huber loss in the value update. Lastly, in **No MR**, we remove model-based representation learning and train the encoder end-to-end with the value function.

**Horizons.** Finally, we consider the role of extended predictions. In **1-step return**, we remove multi-step value predictions and use TD learning. In **No unroll**, we remove the dynamics unrolling in Equation 14, by setting the encoder horizon $H_{\text{Enc}} = 1$.

**Discussion.** The results of our design study show the benefit of balancing theory with practical relaxations. The experiments further validate our design choices and hyperparameters. We highlight two results in particular: (1) increasing the model capacity in the "non-linear model" experiment, does not improve performance. This outcome suggests that maintaining an approximately linear relationship with the value function can be more impactful than increased capacity. (2) Our study also reveals a key distinction between the Gym and Atari benchmarks—while the "MSE reward loss" and "No unroll" variants offer moderate performance gains in Gym, they significantly degrade performance in Atari. This discrepancy highlights how hyperparameters can overfit to individual benchmarks, emphasizing the importance of evaluating algorithms across multiple benchmarks.

## 6 DISCUSSION AND CONCLUSION

This paper introduces MR.Q, a general model-free deep RL algorithm that achieves strong performance across diverse benchmarks and environments. Drawing inspiration from the theory of model-based representation learning, MR.Q demonstrates that model-free deep RL is a promising avenue for building general-purpose algorithms that achieve high performance across environments, while being simpler and less expensive than model-based alternatives.

Our work also reveals insights on which design choices matter when building general-purpose model-free deep RL algorithms and how common benchmarks respond to these design choices.

**Model-based and model-free RL.** MR.Q integrates model-based objectives with a model-free backbone during training, effectively blurring the boundary between traditional model-based and model-free RL. While MR.Q could be extended to the model-based setting by incorporating planning or simulated trajectories with the state-action encoder, these components can add significant execution time and increase the overall complexity and tuning required by a method. Moreover, the performance of MR.Q in these common RL benchmarks demonstrates that these model-based components may be simply unnecessary—suggesting that the representation itself could be the most valuable aspect of model-based learning, even in methods that do use planning. This argument is echoed by DreamerV3 and TD-MPC2, which rely on short planning horizons and trajectory generation, while including both value functions and traditional model-free policy updates. As such, it may be necessary to examine more complex settings, to reliably see a benefit from model-based search or planning, e.g., (Silver et al., 2016).

**Universality of RL benchmarks.** Our results demonstrate that there is a striking lack of positive transfer between benchmarks. For example, despite the similarities in tasks and the same underlying MuJoCo simulator, the top performers in Gym and DMC fail to replicate their success on the opposing benchmark. Similarly, although DreamerV3 excels at Atari, these performance benefits do not translate to continuous control environments, underperforming TD3 in Gym and outright failing to learn the Dog and Humanoid tasks in DMC (see Appendix C). These findings show the limitations of single-benchmark evaluations, indicating that success on one benchmark may not translate easily to others, and highlights the need for more comprehensive benchmarks.

**Limitations.** MR.Q is only the first step towards a new generation of general-purpose model-free deep RL algorithms. Many challenges remains for a fully general algorithm. In particular, MR.Q is not equipped to handle settings such as hard exploration tasks or non-Markovian environments. Another limitation is our evaluation only considers standard RL benchmarks. Although this allows direct comparison with other methods, established algorithms such as PPO have demonstrated their effectiveness in highly unique settings, such as team video games (Berner et al., 2019), drone racing (Kaufmann et al., 2023), and large language models (Achiam et al., 2023; Touvron et al., 2023). To demonstrate similar versatility, new algorithms must undergo the same rigorous testing across a range of tasks that is beyond the scope of any single study.

As the community continues to push the boundaries of what is possible with deep RL, we believe that building simpler general-purpose algorithms has the potential to make this technology more accessible to a wider audience, ultimately enabling users to train agents with ease. Perhaps one day — with just the click of a button.

ACKNOWLEDGMENTS

We would like to thank Brandon Amos, Mikhael Henaff, Luis Pineda, Paria Rashidinejad, and Qinqing Zheng for insightful discussions and comments.

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

# Appendix

## A   PROOFS

**Theorem 1.** *The fixed point of the model-free approach (Equation 4) and the solution of the model-based approach (Equation 5) are the same.*

*Proof.* Let $Z$ be a matrix containing state-action embeddings $\mathbf{z}_{sa}$ for each state-action pair $(s, a) \in S \times A$. Let $Z'$ be the corresponding matrix of next state-action embeddings $\mathbf{z}_{s'a'}$. Let $R$ be the vector of the corresponding rewards $r(s, a)$.

The linear semi-gradient TD update:

$$\mathbf{w}_{t+1} := \mathbf{w}_t - \alpha Z^\top (Z\mathbf{w}_t - (R + \gamma Z'\mathbf{w}_t)) \tag{21}$$

$$= \mathbf{w}_t - \alpha Z^\top Z\mathbf{w}_t + \alpha Z^\top R + \alpha \gamma Z^\top Z'\mathbf{w}_t \tag{22}$$

$$= (I - \alpha(Z^\top Z - \gamma Z^\top Z'))\mathbf{w}_t + \alpha Z^\top R \tag{23}$$

$$= (I - \alpha A)\mathbf{w}_t + \alpha B, \tag{24}$$

where $A := Z^\top Z - \gamma Z^\top Z'$ and $B := Z^\top R$.

The fixed point of the system:

$$\mathbf{w}_{\mathrm{mf}} = (I - \alpha A)\mathbf{w}_{\mathrm{mf}} + \alpha B \tag{25}$$

$$\mathbf{w}_{\mathrm{mf}} - (I - \alpha A)\mathbf{w}_{\mathrm{mf}} = \alpha B \tag{26}$$

$$\alpha A \mathbf{w}_{\mathrm{mf}} = \alpha B \tag{27}$$

$$\mathbf{w}_{\mathrm{mf}} = A^{-1} B. \tag{28}$$

The least squares solution to $W_p$ and $\mathbf{w}_r$

$$W_p := \left(Z^\top Z\right)^{-1} Z^\top Z' \tag{29}$$

$$\mathbf{w}_r := \left(Z^\top Z\right)^{-1} Z^\top R \tag{30}$$

By rolling out $W_p$ and $\mathbf{w}_r$, we arrive at a model-based solution:

$$Q := Z\mathbf{w}_{\text{mb}} = Z \sum_{t=0}^{\infty} \gamma^t W_p^t \mathbf{w}_r. \tag{31}$$

Simplify $\mathbf{w}_{\text{mb}}$:

$$\mathbf{w}_{\text{mb}} := \sum_{t=0}^{\infty} \gamma^t W_p^t \mathbf{w}_r \tag{32}$$

$$\mathbf{w}_{\text{mb}} = \left(I - \gamma W_p\right)^{-1} \mathbf{w}_r \tag{33}$$

$$\mathbf{w}_{\text{mb}} = \left(I - \gamma \left(Z^\top Z\right)^{-1} Z^\top Z'\right)^{-1} \left(Z^\top Z\right)^{-1} Z^\top R \tag{34}$$

$$Z^\top Z \left(I - \gamma \left(Z^\top Z\right)^{-1} Z^\top Z'\right) \mathbf{w}_{\text{mb}} = Z^\top R \tag{35}$$

$$\left(Z^\top Z - \gamma Z^\top Z'\right) \mathbf{w}_{\text{mb}} = Z^\top R \tag{36}$$

$$\mathbf{w}_{\text{mb}} = A^{-1} B \tag{37}$$

$$\mathbf{w}_{\text{mb}} = \mathbf{w}_{\text{mf}}. \tag{38}$$

∎

**Theorem 2.** *The value error of the solution described by Theorem 1 is bounded by the accuracy of the estimated dynamics and reward:*

$$|\text{VE}(s,a)| \le \frac{1}{1-\gamma} \max_{(s,a)\in S\times A} \left(|\mathbf{z}_{sa}^\top \mathbf{w}_r - \mathbb{E}_{r|s,a}[r]| + \max_i |\mathbf{w}_i| \left|\sum |\mathbf{z}_{sa}^\top W_p - \mathbb{E}_{s',a'|s,a}[\mathbf{z}_{s'a'}]|\right|\right). \tag{39}$$

*Proof.* Let $\mathbf{w}$ be the solution described in Theorem 1, i.e. $\mathbf{w} = \mathbf{w}_{\text{mb}} = \mathbf{w}_{\text{mf}}$. Let $p^\pi(s,a)$ be the discounted state-action visitation distribution according to the policy $\pi$ starting from the state-action pair $(s,a)$.

Firstly from Theorem 1, we can show that

$$\mathbf{w} = \left(I - \gamma W_p\right)^{-1} \mathbf{w}_r \tag{40}$$

$$\Rightarrow \left(I - \gamma W_p\right)\mathbf{w} = \mathbf{w}_r \tag{41}$$

$$\Rightarrow \mathbf{w} - \gamma W_p \mathbf{w} = \mathbf{w}_r. \tag{42}$$

Simplify $\text{VE}(s,a)$:

$$\text{VE}(s,a) := Q(s,a) - Q^\pi(s,a) \tag{43}$$

$$= Q(s,a) - Q^\pi(s,a) \tag{44}$$

$$= Q(s,a) - \mathbb{E}_{r,s',a'}\left[r + \gamma Q^\pi(s',a')\right] \tag{45}$$

$$= Q(s,a) - \mathbb{E}_{r,s',a'}\left[r + \gamma \left(Q(s',a') - \text{VE}(s',a')\right)\right] \tag{46}$$

$$= Q(s,a) - \mathbb{E}_{r,s',a'}\left[r + \gamma Q(s',a')\right] + \gamma \mathbb{E}_{s',a'}\left[\text{VE}(s',a')\right] \tag{47}$$

$$= Q(s,a) - \mathbb{E}_{r,s',a'}\left[r - \mathbf{z}_{sa}^\top \mathbf{w}_r + \mathbf{z}_{sa}^\top \mathbf{w}_r + \gamma Q(s',a')\right] + \gamma \mathbb{E}_{s',a'}\left[\text{VE}(s',a')\right] \tag{48}$$

$$= Q(s,a) - \mathbb{E}_{r,s',a'}\left[r - \mathbf{z}_{sa}^\top \mathbf{w}_r + \mathbf{z}_{sa}^\top \mathbf{w}_r + \gamma \left(\mathbf{z}_{s'a'}^\top \mathbf{w} - \mathbf{z}_{sa}^\top W_p \mathbf{w} + \mathbf{z}_{sa}^\top W_p \mathbf{w}\right)\right]$$
$$+ \gamma \mathbb{E}_{s',a'}\left[\text{VE}(s',a')\right] \tag{49}$$

$$= \mathbf{z}_{sa}^\top \mathbf{w} - \mathbb{E}_{r,s',a'}\left[r - \mathbf{z}_{sa}^\top \mathbf{w}_r + \mathbf{z}_{sa}^\top \mathbf{w}_r + \gamma \left(\mathbf{z}_{s'a'}^\top \mathbf{w} - \mathbf{z}_{sa}^\top W_p \mathbf{w} + \mathbf{z}_{sa}^\top W_p \mathbf{w}\right)\right]$$
$$+ \gamma \mathbb{E}_{s',a'}\left[\text{VE}(s',a')\right] \tag{50}$$

$$= \mathbf{z}_{sa}^\top \mathbf{w} - \mathbb{E}_r \left[ r - \mathbf{z}_{sa}^\top \mathbf{w}_r + \mathbf{z}_{sa}^\top \mathbf{w}_r \right] - \gamma \mathbb{E}_{s',a'} \left[ \mathbf{z}_{s'a'}^\top \mathbf{w} - \mathbf{z}_{sa}^\top W_p \mathbf{w} + \mathbf{z}_{sa}^\top W_p \mathbf{w} \right]$$
$$+ \gamma \mathbb{E}_{s',a'} \left[ \text{VE}(s', a') \right] \tag{51}$$

$$= \mathbf{z}_{sa}^\top \mathbf{w} - \mathbf{z}_{sa}^\top \mathbf{w}_r - \gamma \mathbf{z}_{sa}^\top W_p \mathbf{w} - \mathbb{E}_r \left[ r - \mathbf{z}_{sa}^\top \mathbf{w}_r \right] - \gamma \mathbb{E}_{s',a'} \left[ \mathbf{z}_{s'a'}^\top \mathbf{w} - \mathbf{z}_{sa}^\top W_p \mathbf{w} \right]$$
$$+ \gamma \mathbb{E}_{s',a'} \left[ \text{VE}(s', a') \right] \tag{52}$$

$$= \mathbf{z}_{sa}^\top \left( \mathbf{w} - \gamma W_p \mathbf{w} - \mathbf{w}_r \right) - \mathbb{E}_r \left[ r - \mathbf{z}_{sa}^\top \mathbf{w}_r \right] - \gamma \mathbb{E}_{s',a'} \left[ \mathbf{z}_{s'a'}^\top \mathbf{w} - \mathbf{z}_{sa}^\top W_p \mathbf{w} \right]$$
$$+ \gamma \mathbb{E}_{s',a'} \left[ \text{VE}(s', a') \right] \tag{53}$$

$$= - \mathbb{E}_r \left[ r - \mathbf{z}_{sa}^\top \mathbf{w}_r \right] - \gamma \mathbb{E}_{s',a'} \left[ \mathbf{z}_{s'a'}^\top \mathbf{w} - \mathbf{z}_{sa}^\top W_p \mathbf{w} \right] + \gamma \mathbb{E}_{s',a'} \left[ \text{VE}(s', a') \right] \tag{54}$$

$$= \left( \mathbf{z}_{sa}^\top \mathbf{w}_r - \mathbb{E}_r \left[ r \right] \right) + \gamma \left( \mathbf{z}_{sa}^\top W_p - \mathbb{E}_{s',a'} \left[ \mathbf{z}_{s'a'}^\top \right] \right) \mathbf{w} + \gamma \mathbb{E}_{s',a'} \left[ \text{VE}(s', a') \right]. \tag{55}$$

Then given the recursive relationship, akin to the Bellman equation (Sutton & Barto, 1998), the value error VE recursively expands to the discounted state-action visitation distribution $p^\pi$. For $(\hat{s}, \hat{a}) \in S \times A$:

$$\text{VE}(\hat{s}, \hat{a}) = \frac{1}{1 - \gamma} \mathbb{E}_{(s,a) \sim p^\pi(\hat{s}, \hat{a})} \left[ \left( \mathbf{z}_{sa}^\top \mathbf{w}_r - \mathbb{E}_{r|s,a} \left[ r \right] \right) + \gamma \left( \mathbf{z}_{sa}^\top W_p - \mathbb{E}_{s',a'|s,a} \left[ \mathbf{z}_{s'a'}^\top \right] \right) \mathbf{w} \right]. \tag{56}$$

Taking the absolute value:

$$|\text{VE}(\hat{s}, \hat{a})| = \left| \frac{1}{1 - \gamma} \mathbb{E}_{(s,a) \sim p^\pi(\hat{s}, \hat{a})} \left[ \left( \mathbf{z}_{sa}^\top \mathbf{w}_r - \mathbb{E}_{r|s,a} \left[ r \right] \right) + \gamma \left( \mathbf{z}_{sa}^\top W_p - \mathbb{E}_{s',a'|s,a} \left[ \mathbf{z}_{s'a'}^\top \right] \right) \mathbf{w} \right] \right| \tag{57}$$

$$|\text{VE}(\hat{s}, \hat{a})| \leq \frac{1}{1 - \gamma} \mathbb{E}_{(s,a) \sim p^\pi(\hat{s}, \hat{a})} \left[ \left| \mathbf{z}_{sa}^\top \mathbf{w}_r - \mathbb{E}_{r|s,a} \left[ r \right] \right| + \gamma \left| \left( \mathbf{z}_{sa}^\top W_p - \mathbb{E}_{s',a'|s,a} \left[ \mathbf{z}_{s'a'}^\top \right] \right) \mathbf{w} \right| \right] \tag{58}$$

$$= \frac{1}{1 - \gamma} \max_{(s,a) \in S \times A} \left( \left| \mathbf{z}_{sa}^\top \mathbf{w}_r - \mathbb{E}_{r|s,a} \left[ r \right] \right| + \gamma \left| \left( \mathbf{z}_{sa}^\top W_p - \mathbb{E}_{s',a'|s,a} \left[ \mathbf{z}_{s'a'}^\top \right] \right) \mathbf{w} \right| \right) \tag{59}$$

$$\leq \frac{1}{1 - \gamma} \max_{(s,a) \in S \times A} \left( \left| \mathbf{z}_{sa}^\top \mathbf{w}_r - \mathbb{E}_{r|s,a} \left[ r \right] \right| + \max_i |\mathbf{w}_i| \sum \left| \mathbf{z}_{sa}^\top W_p - \mathbb{E}_{s',a'|s,a} \left[ \mathbf{z}_{s'a'} \right] \right| \right). \tag{60}$$

∎

**Theorem 3.** *Given functions $f(s) = \mathbf{z}_s$ and $g(\mathbf{z}_s, a) = \mathbf{z}_{sa}$, then if there exists functions $\hat{p}$ and $\hat{R}$ such that for all $(s, a) \in S \times A$:*

$$\mathbb{E}_{\hat{R}}[\hat{R}(\mathbf{z}_{sa})] = \mathbb{E}_R \left[ R(s, a) \right], \qquad \hat{p}(\mathbf{z}_{s'} | \mathbf{z}_{sa}) = \sum_{\hat{s} : \mathbf{z}_{\hat{s}} = \mathbf{z}_{s'}} p(\hat{s} | s, a), \tag{61}$$

*then for any policy $\pi$ where there exists a corresponding policy $\hat{\pi}(a|\mathbf{z}_s) = \pi(a|s)$, there exists a function $\hat{Q}$ equal to the true value function $Q^\pi$ over all possible state-action pairs $(s, a) \in S \times A$:*

$$\hat{Q}(\mathbf{z}_{sa}) = Q^\pi(s, a). \tag{62}$$

*Furthermore, Equation 61 guarantees the existence of an optimal policy $\hat{\pi}^*(a|\mathbf{z}_s) = \pi^*(a|s)$.*

*Proof.* Let

$$Q_h^\pi(s, a) = \sum_{t=0}^h \gamma^t \mathbb{E}_\pi [R(s_t, a_t)|s_0 = s, a_0 = a] \tag{63}$$

$$\hat{Q}_h(\mathbf{z}_{sa}) = \sum_{t=0}^h \gamma^t \mathbb{E}_\pi [\hat{R}(\mathbf{z}_{s_t a_t})|s_0 = s, a_0 = a] \tag{64}$$

Then

$$Q_0^\pi(s, a) = \mathbb{E}_R[R(s, a)] \tag{65}$$

$$= \mathbb{E}_{\hat{R}}[\hat{R}(\mathbf{z}_{sa})] \tag{66}$$

$$= \hat{Q}_0(\mathbf{z}_{sa}). \tag{67}$$

Assuming $Q_{n-1}^{\pi}(s,a) = \hat{Q}_{n-1}(\mathbf{z}_{sa})$ then noting that $\hat{p}(\mathbf{z}|\mathbf{z}_{sa}) = 0$ if $\mathbf{z}$ that is not in the image of $f(s) = \mathbf{z}_s$.

$$Q_n^{\pi}(s,a) = \mathbb{E}_R[R(s,a)] + \gamma \mathbb{E}_{s',a'}[Q_{n-1}^{\pi}(s',a')] \tag{68}$$

$$= \mathbb{E}_{\hat{R}}[\hat{R}(s,a)] + \gamma \mathbb{E}_{s',a'}[\hat{Q}_{n-1}(\mathbf{z}_{s'a'})] \tag{69}$$

$$= \mathbb{E}_{\hat{R}}[\hat{R}(s,a)] + \gamma \sum_{s'} \sum_{a'} p(s'|s,a) \pi(a'|s') \hat{Q}_{n-1}(\mathbf{z}_{s'a'}) \tag{70}$$

$$= \mathbb{E}_{\hat{R}}[\hat{R}(s,a)] + \gamma \sum_{z_{s'}} \sum_{a'} \hat{p}(\mathbf{z}_{s'}|\mathbf{z}_{sa}) \hat{\pi}(a'|\mathbf{z}_{s'}) \hat{Q}_{n-1}(\mathbf{z}_{s'a'}) \tag{71}$$

$$= \hat{Q}_n(\mathbf{z}_{sa}). \tag{72}$$

Thus $\hat{Q}(\mathbf{z}_{sa}) = \lim_{n\to\infty} \hat{Q}_n(\mathbf{z}_{sa})$ exists, as $\hat{Q}_n$ can be defined as a function of $\hat{p}$, $\hat{R}$, and $\hat{\pi}$ for all $n$.

Similarly, let $\pi$ be an optimal policy. Repeating the same arguments we see that

$$Q_n^{\pi}(s,a) = \mathbb{E}_R[R(s,a)] + \gamma \mathbb{E}_{s',a'}[Q_{n-1}^{\pi}(s',a')] \tag{73}$$

$$= \mathbb{E}_R[R(s,a)] + \gamma \sum_{s'} p(s'|s,a) \max_{a'} Q_{n-1}^{\pi}(s',a') \tag{74}$$

$$= \mathbb{E}_{\hat{R}}[\hat{R}(s,a)] + \gamma \sum_{z_{s'}} \hat{p}(\mathbf{z}_{s'}|\mathbf{z}_{sa}) \max_{a'} \hat{Q}_{n-1}(\mathbf{z}_{s'a'}) \tag{75}$$

$$= \hat{Q}_n(\mathbf{z}_{sa}). \tag{76}$$

Thus there exists a function $\hat{Q}(g(\mathbf{z}_s,a)) = Q^*(s,a)$, consequently, there exists an optimal policy $\hat{\pi}^*(a|\mathbf{z}_s) = \operatorname{argmax}_a \hat{Q}(s,a)$.

∎

# B  EXPERIMENTAL DETAILS

## B.1  HYPERPARAMETERS

Table 3: **MR.Q Hyperparameters.** Hyperparameters values are kept fixed across all benchmarks.

| | Hyperparameter | Value |
|---|---|---|
| | Dynamics loss weight $\lambda_{\text{Dynamics}}$ | 1 |
| | Reward loss weight $\lambda_{\text{Reward}}$ | 0.1 |
| | Terminal loss weight $\lambda_{\text{Terminal}}$ | 0.1 |
| | Pre-activation loss weight $\lambda_{\text{pre-activ}}$ | $1e-5$ |
| | Encoder horizon $H_{\text{Enc}}$ | 5 |
| | Multi-step returns horizon $H_Q$ | 3 |
| TD3 (Fujimoto et al., 2018) | Target policy noise $\sigma$ | $\mathcal{N}(0, 0.2^2)$ |
| | Target policy noise clipping $c$ | $(-0.3, 0.3)$ |
| LAP (Fujimoto et al., 2020) | Probability smoothing $\alpha$ | 0.4 |
| | Minimum priority | 1 |
| Exploration | Initial random exploration time steps | 10k |
| | Exploration noise | $\mathcal{N}(0, 0.2^2)$ |
| Common | Discount factor $\gamma$ | 0.99 |
| | Replay buffer capacity | 1M |
| | Mini-batch size | 256 |
| | Target update frequency $T_{\text{target}}$ | 250 |
| | Replay ratio | 1 |
| Encoder Network | Optimizer | AdamW (Loshchilov & Hutter, 2019) |
| | Learning rate | $1e-4$ |
| | Weight decay | $1e-4$ |
| | $\mathbf{z}_s$ dim | 512 |
| | $\mathbf{z}_{sa}$ dim | 512 |
| | $\mathbf{z}_a$ dim (only used within architecture) | 256 |
| | Hidden dim | 512 |
| | Activation function | ELU (Clevert et al., 2015) |
| | Weight initialization | Xavier uniform (Glorot & Bengio, 2010) |
| | Bias initialization | 0 |
| | Reward bins | 65 |
| | Reward range | $[-10, 10]$ (effective: $[-22k, 22k]$) |
| Value Network | Optimizer | AdamW |
| | Learning rate | $3e-4$ |
| | Hidden dim | 512 |
| | Activation function | ELU |
| | Weight initialization | Xavier uniform |
| | Bias initialization | 0 |
| | Gradient clip norm | 20 |
| Policy Network | Optimizer | AdamW |
| | Learning rate | $3e-4$ |
| | Hidden dim | 512 |
| | Activation function | ReLU |
| | Weight initialization | Xavier uniform |
| | Bias initialization | 0 |
| | Gumbel-Softmax $\tau$ (Jang et al., 2017) | 10 |

### B.2 NETWORK ARCHITECTURE

This section describes the networks used in our method using PyTorch code blocks (Paszke et al., 2019). The state encoder and state-action encoder are described as separate networks for clarity but are trained end-to-end as a single network. The value and policy networks are trained independently from the encoders.

**Preamble**

```python
import torch
import torch.nn as nn
import torch.nn.functional as F
from functools import partial

zs_dim = 512
za_dim = 256
zsa_dim = 512

def ln_activ(self, x):
    x = F.layer_norm(x, (x.shape[-1],))
    return self.activ(x)
```

**State Encoder $f$ Network**

For image inputs, four convolutional layers are used, each with 32 output channels, kernel size of 3, strides of $(2, 2, 2, 1)$, and ELU activations (Clevert et al., 2015). The convolutional layers are followed by a linear layer taking in the flattened output followed by LayerNorm (Ba et al., 2016) and a final ELU activation.

For vector inputs, a three layer multilayer perceptron (MLP) is used, with hidden dimension $512$ and LayerNorm followed by ELU activations after each layer.

The resulting state embedding $\mathbf{z}_s$ is trained end-to-end with the state-action encoder. It is also used downstream by the policy network (without propagating gradients).

```python
if image_observation_space:
    self.zs_cnn1 = nn.Conv2d(state_channels, 32, 3, stride=2)
    self.zs_cnn2 = nn.Conv2d(32, 32, 3, stride=2)
    self.zs_cnn3 = nn.Conv2d(32, 32, 3, stride=2)
    self.zs_cnn4 = nn.Conv2d(32, 32, 3, stride=1)
    # Assumes 84 x 84 input
    self.zs_lin = nn.Linear(1568, zs_dim)
else:
    self.zs_mlp1 = nn.Linear(state_dim, 512)
    self.zs_mlp2 = nn.Linear(512, 512)
    self.zs_mlp3 = nn.Linear(512, zs_dim)

self.activ = F.elu

def cnn_forward(self, state):
    state = state/255. - 0.5
    zs = self.activ(self.zs_cnn1(state))
    zs = self.activ(self.zs_cnn2(zs))
    zs = self.activ(self.zs_cnn3(zs))
    zs = self.activ(self.zs_cnn4(zs))
    zs = zs.reshape(batch_size, 1568)
    return ln_activ(self.zs_lin(zs))

def mlp_forward(self, state):
    zs = self.ln_activ(self.zs_mlp1(state))
    zs = self.ln_activ(self.zs_mlp2(zs))
    return self.ln_activ(self.zs_mlp3(zs))
```

**State-Action Encoder $g$ Network**

Action input is processed by a linear layer followed by an ELU activation. Afterwards, the processed action is concatenated with the state embedding and processed by a three layer MLP with hidden dimension 512, and LayerNorm followed by ELU activations after the first two layers.

The resulting state-action embedding $\mathbf{z}_{sa}$ is used by a linear layer to make predictions about reward, the next state embedding, and the terminal signal. It is also used downstream by the value network (without propagating gradients).

```python
1  self.za = nn.Linear(action_dim, za_dim)
2  self.zsa1 = nn.Linear(zs_dim + za_dim, 512)
3  self.zsa2 = nn.Linear(512, 512)
4  self.zsa3 = nn.Linear(512, zsa_dim)
5  self.model = nn.Linear(zsa_dim, output_dim)
6  self.activ = F.elu
7
8  def forward(self, zs, action):
9      za = self.activ(self.za(action))
10     zsa = torch.cat([zs, za], 1)
11     zsa = self.ln_activ(self.zsa1(zsa))
12     zsa = self.ln_activ(self.zsa2(zsa))
13     zsa = self.zsa3(zsa)
14     return self.model(zsa), zsa
```

**Value $Q$ Networks**

The value network is a four layer MLP with hidden dimension 512, and LayerNorm followed by ELU activations after the first three layers.

Two value networks are used with the same network and forward pass.

```python
1  self.l1 = nn.Linear(zsa_dim, 512)
2  self.l2 = nn.Linear(512, 512)
3  self.l3 = nn.Linear(512, 512)
4  self.l4 = nn.Linear(512, 1)
5  self.activ = F.elu
6
7  def forward(self, zsa):
8      q = self.ln_activ(self.l1(zsa))
9      q = self.ln_activ(self.l2(q))
10     q = self.ln_activ(self.l3(q))
11     return self.l4(q)
```

**Policy $\pi$ Network**

The policy network is a three layer MLP with hidden dimension 512, and LayerNorm followed by ReLU activations after the first two layers.

For discrete actions, the final activation is the Gumbel Softmax with $\tau = 10$. For continous actions, the final activation is a tanh function.

```python
1  self.l1 = nn.Linear(zs_dim, 512)
2  self.l2 = nn.Linear(hdim, 512)
3  self.l3 = nn.Linear(512, action_dim)
4  self.activ = F.relu
5
6  if discrete_action_space:
7      self.final_activ = partial(F.gumbel_softmax, tau=10)
8  else:
9      self.final_activ = torch.tanh
10
11 def forward(self, zs):
12     a = self.ln_activ(self.l1(zs))
13     a = self.ln_activ(self.l2(a))
14     return self.final_activ(self.l3(a))
```

### B.3 ENVIRONMENTS

All main experiments were run for 10 seeds (the design study is based on 5 seeds). Evaluations are based on the average performance over 10 episodes, measured every 5k time steps for Gym and DM control and every 100k time steps for Atari.

**Gym - Locomotion.** For the gym locomotion tasks (Todorov et al., 2012; Brockman et al., 2016; Towers et al., 2024), we choose the five most common environments that appear in prior work (Fujimoto et al., 2018; 2024; Haarnoja et al., 2018; Kuznetsov et al., 2020). We use the -v4 version. No preprocessing is applied. When aggregating scores, we use normalize with the TD3 scores obtained from TD7 (Fujimoto et al., 2024):

$$\text{TD3-Normalized}(x) \coloneqq \frac{x - \text{random score}}{\text{TD3 score} - \text{random score}}. \tag{77}$$

|  | Random | TD3 |
|---|---|---|
| Ant-v4 | -70.288 | 3942 |
| HalfCheetah-v4 | -289.415 | 10574 |
| Hopper-v4 | 18.791 | 3226 |
| Humanoid-v4 | 120.423 | 5165 |
| Walker2d-v4 | 2.791 | 3946 |

**DM Control Suite.** For the DM control suite (Tassa et al., 2018), we choose the 28 default environments that appear either in the evaluation of TD-MPC2 or DreamerV3. We omit any custom environments included by the TD-MPC2 authors. The same subset of tasks are used in the evaluation of proprioceptive and visual control. Like prior work, for both observation spaces, we use an action repeat of 2 (Hansen et al., 2024). For visual control, the state (network input) is composed of the previous 3 observations which are resized to $84 \times 84$ pixels in RGB format (Tassa et al., 2018).

**Atari.** For the Atari games (Bellemare et al., 2013; Brockman et al., 2016; Towers et al., 2024), we use the 57 games in the Atari-57 benchmark that appears in prior work (Hessel et al., 2018; Schrittwieser et al., 2020; Badia et al., 2020; Hafner et al., 2023). For DQN and Rainbow, two games (Defender and Surround) are missing from the Dopamine framework (Castro et al., 2018) and are omitted. We use the -v5 version. For MR.Q, we use the common preprocessing steps (Mnih et al., 2015; Machado et al., 2018; Castro et al., 2018), where an action repeat of 4 is used and the observations are grayscaled, resized to $84 \times 84$ pixels and set to the max between the 3rd and 4th frame. The state (network input) is composed of the previous 4 observations.

Consider the 16 frame sequence used by a single state, where $f_i$ is the $i$th grayscaled and resized frame and $o_j$ is the $j$th observation set to the max of two frames

$$\overbrace{f_0, f_1, \underbrace{f_2, f_3}_{o_0 = \max(f_2, f_3)}}^{\text{action } a_0}, \overbrace{f_4, f_5, \underbrace{f_6, f_7}_{o_1 = \max(f_6, f_7)}}^{\text{action } a_1}, \overbrace{f_8, f_9, \underbrace{f_{10}, f_{11}}_{o_2 = \max(f_{10}, f_{11})}}^{\text{action } a_2}, \overbrace{f_{12}, f_{13}, \underbrace{f_{14}, f_{15}}_{o_3 = \max(f_{14}, f_{15})}}^{\text{action } a_3}, \tag{78}$$

then the state is defined as follows:

$$s = \begin{bmatrix} o_0 = \max(f_2, f_3) \\ o_1 = \max(f_6, f_7) \\ o_2 = \max(f_{10}, f_{11}) \\ o_3 = \max(f_{14}, f_{15}) \end{bmatrix}. \tag{79}$$

When aggregating scores, we normalize with Human scores obtained from (Wang et al., 2016):

$$\text{Human-Normalized}(x) \coloneqq \frac{x - \text{random score}}{\text{Human score} - \text{random score}}. \tag{80}$$

|  | Random | Human |
|---|---|---|
| Alien | 227.8 | 7127.7 |
| Amidar | 5.8 | 1719.5 |
| Assault | 222.4 | 742.0 |
| Asterix | 210.0 | 8503.3 |
| Asteroids | 719.1 | 47388.7 |
| Atlantis | 12850.0 | 29028.1 |
| BankHeist | 14.2 | 753.1 |
| BattleZone | 2360.0 | 37187.5 |
| BeamRider | 363.9 | 16926.5 |
| Berzerk | 123.7 | 2630.4 |
| Bowling | 23.1 | 160.7 |
| Boxing | 0.1 | 12.1 |
| Breakout | 1.7 | 30.5 |
| Centipede | 2090.9 | 12017.0 |
| ChopperCommand | 811.0 | 7387.8 |
| CrazyClimber | 10780.5 | 35829.4 |
| Defender (not used) | 2874.5 | 18688.9 |
| DemonAttack | 152.1 | 1971.0 |
| DoubleDunk | -18.6 | -16.4 |
| Enduro | 0.0 | 860.5 |
| FishingDerby | -91.7 | -38.7 |
| Freeway | 0.0 | 29.6 |
| Frostbite | 65.2 | 4334.7 |
| Gopher | 257.6 | 2412.5 |
| Gravitar | 173.0 | 3351.4 |
| Hero | 1027.0 | 30826.4 |
| IceHockey | -11.2 | 0.9 |
| Jamesbond | 29.0 | 302.8 |
| Kangaroo | 52.0 | 3035.0 |
| Krull | 1598.0 | 2665.5 |
| KungFuMaster | 258.5 | 22736.3 |
| MontezumaRevenge | 0.0 | 4753.3 |
| MsPacman | 307.3 | 6951.6 |
| NameThisGame | 2292.3 | 8049.0 |
| Phoenix | 761.4 | 7242.6 |
| Pitfall | -229.4 | 6463.7 |
| Pong | -20.7 | 14.6 |
| PrivateEye | 24.9 | 69571.3 |
| Qbert | 163.9 | 13455.0 |
| Riverraid | 1338.5 | 17118.0 |
| RoadRunner | 11.5 | 7845.0 |
| Robotank | 2.2 | 11.9 |
| Seaquest | 68.4 | 42054.7 |
| Skiing | -17098.1 | -4336.9 |
| Solaris | 1236.3 | 12326.7 |
| SpaceInvaders | 148.0 | 1668.7 |
| StarGunner | 664.0 | 10250.0 |
| Surround (not used) | -10.0 | 6.5 |
| Tennis | -23.8 | -8.3 |
| TimePilot | 3568.0 | 5229.2 |
| Tutankham | 11.4 | 167.6 |
| UpNDown | 533.4 | 11693.2 |
| Venture | 0.0 | 1187.5 |
| VideoPinball | 16256.9 | 17667.9 |
| WizardOfWor | 563.5 | 4756.5 |
| YarsRevenge | 3092.9 | 54576.9 |
| Zaxxon | 32.5 | 9173.3 |

### B.4 BASELINES

**DreamerV3.** (Hafner et al., 2023). Results for Gym and DMC were obtained by re-running the authors' code (https://github.com/danijar/dreamerv3 - Commit 251910d04c9f38dd9dc385775bb0d6-efa0e57a95) over 10 seeds, using the author-suggested hyperparameters from the DMC benchmark. Code was modified slightly to match our evaluation protocol. Atari results are based on the authors' reported results.

**DrQ-v2.** (Yarats et al., 2022). We use the authors' reported results whenever possible. For missing any results, we re-ran the authors' code (https://github.com/facebookresearch/drqv2 - Commit c0c650b76c6e5d22a7eb5f2edffd1440fe94f8ef) for 10 seeds.

**DQN.** (Mnih et al., 2015). Results were obtained from the Dopamine framework (Castro et al., 2018).

**PPO.** (Schulman et al., 2017). Results were gathered using Stable Baselines 3 (Raffin et al., 2021) and default hyperparameters. The default MLP policy was used for Gym and DMC-proprioceptive and the default CNN policy was used for DMC-visual and Atari.

**Rainbow.** (Hessel et al., 2018). Results were obtained from the Dopamine framework (Castro et al., 2018).

**TD-MPC2.** (Hansen et al., 2024). Results for DMC were obtained by re-running the authors' code on their main branch (https://github.com/nicklashansen/tdmpc2 - Commit 5f6fadec0fec78304b4b53e8171d348b58cac486). As the Gym environments include a termination signal, results for Gym were obtained by running their episodic branch (https://github.com/nicklashansen/tdmpc2/tree/episodic-rl - Commit 3789fcd5b872079ad610fa3299ff47c3a427a04a). All experiments were run for 10 seeds and use the default author-suggested hyperparameters for all tasks.

**TD7.** (Fujimoto et al., 2024). Results for Gym were obtained from the authors. Results for DMC were obtained by re-running the authors' code (https://github.com/sfujim/TD7 - Commit c1c280de1513f474488061b4cf39642b75dd84bd) using our setup for DMC. All experiments use 10 seeds and use the default author-suggested hyperparameters from the Gym benchmark.

### B.5 SOFTWARE VERSIONS

- Gymnasium 0.29.1 (Towers et al., 2024)
- MuJoCo 3.2.2 (Todorov et al., 2012)
- NumPy 2.1.1 (Harris et al., 2020)
- Python 3.11.8 (Van Rossum & Drake Jr, 1995)
- PyTorch 2.4.1 (Paszke et al., 2019)

# C Complete Main Results

## C.1 Gym

Table 4: **Gym - Locomotion final results.** Final average performance at 1M time steps over 10 seeds. The [bracketed values] represent a 95% bootstrap confidence interval. The aggregate mean, median and interquartile mean (IQM) are computed over the TD3-normalized score (see Appendix B.3).

| Task | TD7 | PPO | TD-MPC2 | DreamerV3 | MR.Q |
|------|-----|-----|---------|-----------|------|
| Ant | 8509 [8164, 8852] | 1584 [1355, 1802] | 4751 [3012, 6261] | 1947 [1121, 2751] | 6901 [6261, 7482] |
| HalfCheetah | 17433 [17284, 17550] | 1744 [1525, 2120] | 15078 [14050, 16012] | 5502 [3887, 7117] | 12939 [11663, 13762] |
| Hopper | 3511 [3245, 3746] | 3022 [2587, 3356] | 2081 [1233, 2916] | 2666 [2071, 3201] | 2692 [2131, 3309] |
| Humanoid | 7428 [7300, 7555] | 477 [431, 522] | 6071 [5767, 6327] | 4217 [2791, 5481] | 10223 [9929, 10498] |
| Walker2d | 6096 [5535, 6521] | 2487 [1875, 3067] | 3008 [1659, 4220] | 4519 [3746, 5190] | 6039 [5644, 6386] |
| Mean | 1.57 [1.54, 1.60] | 0.45 [0.41, 0.48] | 1.04 [0.90, 1.16] | 0.76 [0.67, 0.85] | 1.46 [1.41, 1.52] |
| Median | 1.55 [1.45, 1.63] | 0.41 [0.36, 0.47] | 1.18 [0.80, 1.23] | 0.81 [0.56, 0.90] | 1.53 [1.43, 1.61] |
| IQM | 1.54 [1.49, 1.58] | 0.41 [0.35, 0.46] | 1.05 [0.87, 1.19] | 0.72 [0.62, 0.85] | 1.50 [1.44, 1.55] |

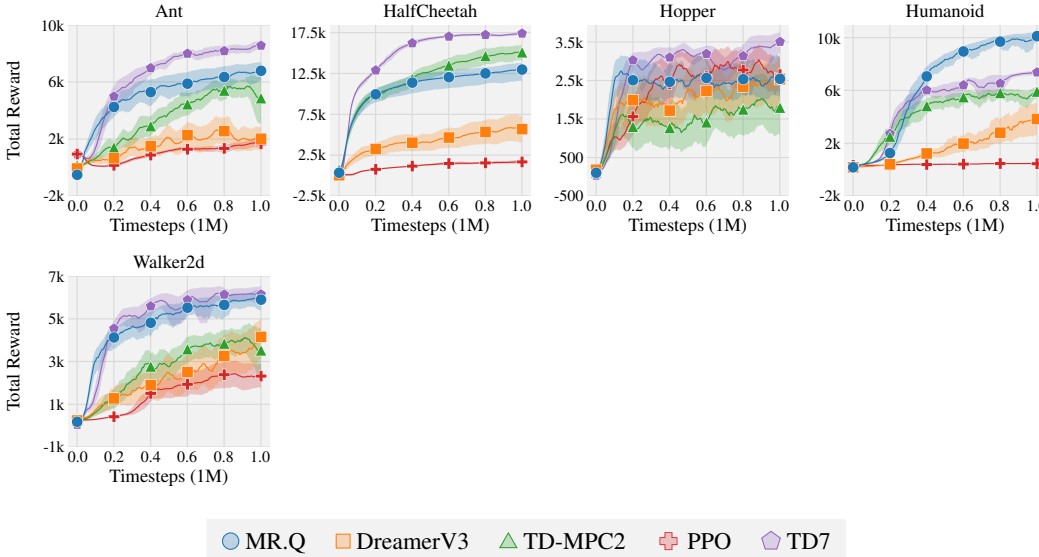

Figure 3: **Gym - Locomotion learning curves.** Results are over 10 seeds. The shaded area captures a 95% boostrap confidence interval.

## C.2 DMC - PROPRIOCEPTIVE

Table 5: **DMC - Proprioceptive final results.** Final average performance at 500k time steps (1M time steps in the original environment due to action repeat) over 10 seeds. The [bracketed values] represent a 95% bootstrap confidence interval. The aggregate mean, median and interquartile mean (IQM) are computed over the default reward.

| Task | TD7 | PPO | TD-MPC2 | DreamerV3 | MR.Q |
|---|---|---|---|---|---|
| acrobot-swingup | 58 [38, 75] | 39 [33, 45] | 584 [551, 615] | 230 [193, 266] | 567 [523, 616] |
| ball_in_cup-catch | 983 [981, 985] | 769 [689, 841] | 984 [982, 986] | 968 [965, 973] | 981 [979, 984] |
| cartpole-balance | 999 [998, 1000] | 999 [1000, 1000] | 996 [995, 998] | 998 [997, 1000] | 999 [999, 1000] |
| cartpole-balance_sparse | 1000 [1000, 1000] | 1000 [1000, 1000] | 1000 [1000, 1000] | 999 [1000, 1000] | 1000 [1000, 1000] |
| cartpole-swingup | 869 [866, 873] | 776 [661, 853] | 875 [870, 880] | 736 [591, 838] | 866 [866, 866] |
| cartpole-swingup_sparse | 573 [333, 806] | 391 [159, 625] | 845 [839, 849] | 702 [560, 792] | 798 [780, 818] |
| cheetah-run | 821 [642, 913] | 269 [247, 295] | 917 [915, 920] | 699 [655, 744] | 914 [911, 917] |
| dog-run | 69 [36, 101] | 26 [26, 28] | 265 [166, 342] | 4 [4, 5] | 569 [547, 595] |
| dog-stand | 582 [432, 741] | 129 [122, 139] | 506 [266, 715] | 22 [20, 27] | 967 [960, 975] |
| dog-trot | 21 [13, 30] | 31 [30, 34] | 407 [265, 530] | 10 [6, 17] | 877 [845, 898] |
| dog-walk | 52 [19, 116] | 40 [37, 43] | 486 [240, 704] | 17 [15, 21] | 916 [908, 924] |
| finger-spin | 335 [99, 596] | 459 [420, 497] | 986 [986, 988] | 666 [577, 763] | 937 [917, 956] |
| finger-turn_easy | 912 [774, 983] | 182 [153, 211] | 979 [975, 983] | 906 [883, 927] | 953 [931, 974] |
| finger-turn_hard | 470 [199, 727] | 58 [35, 79] | 947 [916, 977] | 864 [812, 900] | 950 [910, 974] |
| fish-swim | 86 [64, 120] | 103 [84, 128] | 659 [615, 706] | 813 [808, 819] | 792 [773, 810] |
| hopper-hop | 87 [25, 160] | 10 [0, 23] | 425 [368, 500] | 116 [66, 165] | 251 [195, 301] |
| hopper-stand | 670 [466, 829] | 128 [56, 216] | 952 [944, 958] | 747 [669, 806] | 951 [948, 955] |
| humanoid-run | 57 [23, 92] | 0 [1, 1] | 181 [121, 231] | 0 [1, 1] | 200 [170, 236] |
| humanoid-stand | 317 [117, 516] | 5 [5, 6] | 658 [506, 745] | 5 [5, 6] | 868 [822, 903] |
| humanoid-walk | 176 [42, 320] | 1 [1, 2] | 754 [725, 791] | 1 [1, 2] | 662 [610, 724] |
| pendulum-swingup | 500 [251, 743] | 115 [70, 164] | 846 [830, 862] | 774 [740, 802] | 748 [597, 829] |
| quadruped-run | 645 [567, 713] | 144 [122, 170] | 942 [938, 947] | 130 [92, 169] | 947 [940, 954] |
| quadruped-walk | 949 [939, 957] | 122 [103, 142] | 963 [959, 967] | 193 [137, 243] | 963 [959, 967] |
| reacher-easy | 970 [951, 982] | 367 [188, 558] | 983 [980, 986] | 966 [964, 970] | 983 [983, 985] |
| reacher-hard | 898 [861, 936] | 125 [40, 234] | 960 [936, 979] | 919 [864, 955] | 977 [975, 980] |
| walker-run | 804 [783, 825] | 97 [91, 104] | 854 [851, 859] | 510 [430, 588] | 793 [765, 815] |
| walker-stand | 983 [974, 989] | 431 [363, 495] | 991 [990, 994] | 941 [934, 948] | 988 [987, 990] |
| walker-walk | 977 [975, 980] | 283 [253, 312] | 981 [979, 984] | 898 [875, 919] | 978 [978, 980] |
| Mean | 566 [544, 590] | 254 [241, 267] | 783 [769, 797] | 530 [520, 539] | 835 [829, 842] |
| Median | 613 [548, 718] | 127 [112, 145] | 896 [893, 899] | 700 [644, 741] | 927 [914, 934] |
| IQM | 612 [569, 657] | 154 [135, 167] | 868 [860, 880] | 577 [557, 594] | 907 [903, 914] |

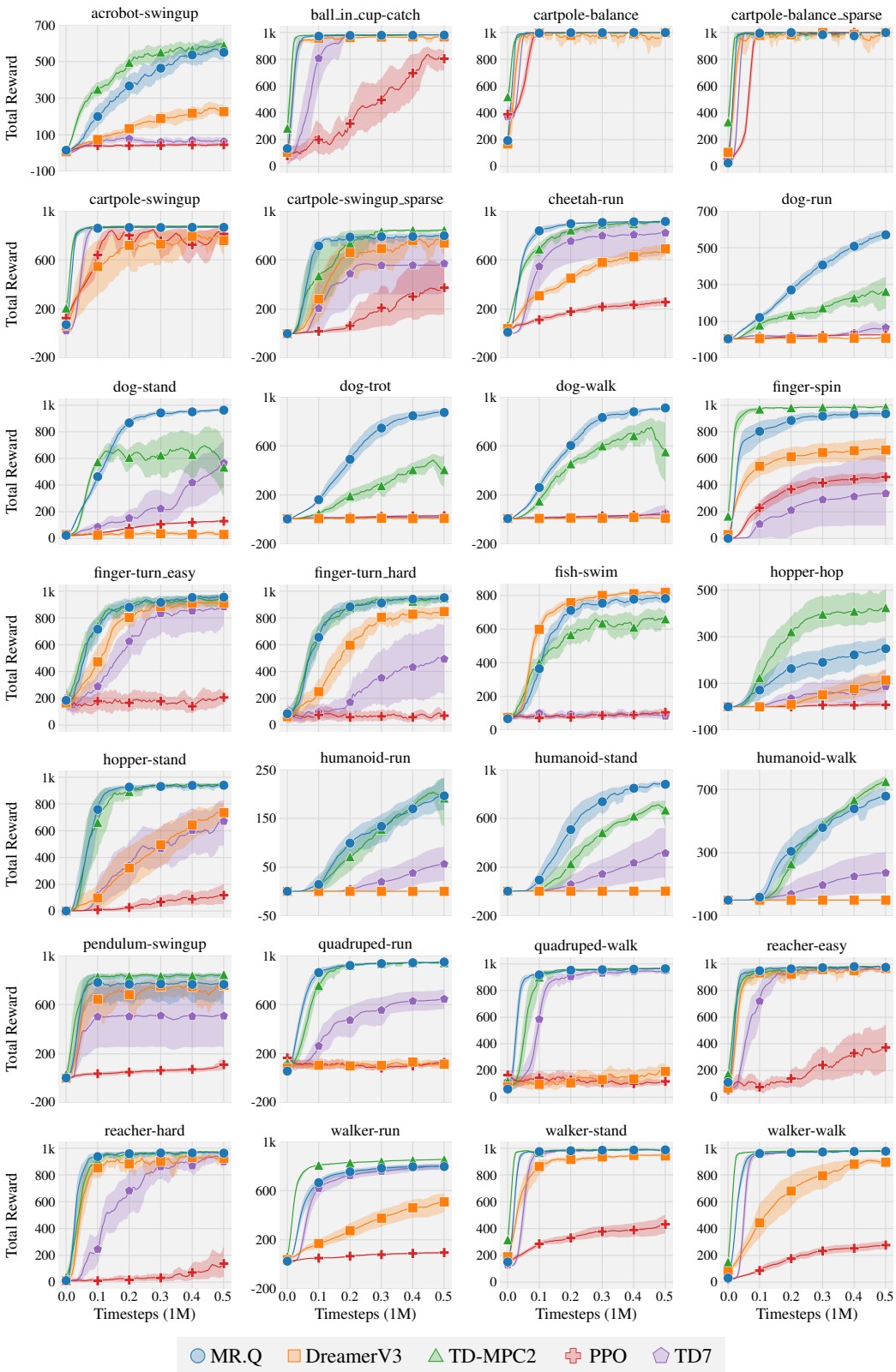

Figure 4: **DMC - Proprioceptive learning curves.** Time steps consider the number of environment interactions, where 500k time steps equals 1M frames in the original environment. Results are over 10 seeds. The shaded area captures a 95% boostrap confidence interval.

## C.3 DMC - VISUAL

Table 6: **DMC - Visual final results.** Final average performance at 500k time steps (1M time steps in the original environment due to action repeat) over 10 seeds. The [bracketed values] represent a 95% bootstrap confidence interval. The aggregate mean, median and interquartile mean (IQM) are computed over the default reward.

| Task | DrQ-v2 | PPO | TD-MPC2 | DreamerV3 | MR.Q |
|---|---|---|---|---|---|
| acrobot-swingup | 168 [127, 219] | 2 [1, 4] | 197 [179, 217] | 121 [106, 145] | 287 [254, 316] |
| ball_in_cup-catch | 909 [821, 973] | 105 [5, 282] | 932 [899, 961] | 971 [969, 973] | 977 [975, 980] |
| cartpole-balance | 993 [990, 996] | 353 [231, 485] | 972 [948, 991] | 998 [997, 1000] | 999 [999, 999] |
| cartpole-balance_sparse | 962 [887, 1000] | 487 [233, 751] | 1000 [1000, 1000] | 999 [999, 1000] | 1000 [1000, 1000] |
| cartpole-swingup | 864 [854, 873] | 596 [437, 723] | 690 [521, 813] | 725 [603, 807] | 868 [860, 875] |
| cartpole-swingup_sparse | 774 [741, 805] | 0 [0, 0] | 636 [404, 804] | 547 [351, 726] | 797 [777, 816] |
| cheetah-run | 728 [701, 753] | 155 [110, 210] | 431 [267, 556] | 618 [576, 661] | 775 [752, 807] |
| dog-run | 10 [9, 12] | 11 [9, 14] | 14 [10, 18] | 9 [6, 14] | 60 [44, 80] |
| dog-stand | 43 [37, 49] | 51 [48, 56] | 117 [72, 148] | 61 [30, 92] | 216 [201, 232] |
| dog-trot | 14 [11, 18] | 13 [12, 15] | 20 [14, 25] | 14 [13, 16] | 65 [55, 79] |
| dog-walk | 22 [18, 29] | 16 [14, 18] | 22 [17, 28] | 11 [11, 12] | 77 [71, 83] |
| finger-spin | 860 [787, 922] | 241 [107, 377] | 786 [492, 984] | 656 [544, 765] | 965 [938, 982] |
| finger-turn_easy | 503 [399, 615] | 189 [144, 233] | 562 [317, 779] | 491 [447, 542] | 953 [927, 974] |
| finger-turn_hard | 223 [121, 340] | 60 [1, 120] | 903 [870, 940] | 494 [401, 571] | 932 [905, 957] |
| fish-swim | 84 [65, 107] | 77 [64, 92] | 43 [21, 64] | 90 [84, 96] | 79 [68, 93] |
| hopper-hop | 224 [170, 278] | 0 [0, 0] | 187 [119, 238] | 205 [125, 287] | 270 [230, 315] |
| hopper-stand | 917 [903, 931] | 1 [0, 2] | 582 [321, 794] | 888 [875, 900] | 852 [703, 930] |
| humanoid-run | 1 [1, 1] | 1 [1, 1] | 0 [1, 1] | 1 [1, 1] | 1 [1, 2] |
| humanoid-stand | 6 [7, 7] | 6 [6, 7] | 5 [5, 7] | 5 [5, 7] | 7 [7, 8] |
| humanoid-walk | 2 [2, 2] | 1 [1, 1] | 1 [1, 2] | 1 [2, 2] | 2 [2, 3] |
| pendulum-swingup | 838 [813, 861] | 0 [0, 1] | 748 [574, 850] | 761 [709, 807] | 829 [816, 842] |
| quadruped-run | 459 [412, 507] | 118 [98, 139] | 262 [184, 330] | 328 [255, 397] | 498 [476, 522] |
| quadruped-walk | 750 [699, 796] | 149 [113, 184] | 246 [179, 310] | 316 [260, 379] | 833 [797, 867] |
| reacher-easy | 938 [903, 973] | 113 [55, 192] | 956 [932, 978] | 735 [678, 796] | 979 [978, 982] |
| reacher-hard | 705 [580, 831] | 10 [0, 30] | 911 [867, 946] | 338 [227, 461] | 965 [945, 977] |
| walker-run | 546 [475, 612] | 39 [35, 44] | 665 [566, 719] | 669 [615, 708] | 615 [571, 655] |
| walker-stand | 980 [977, 984] | 253 [210, 310] | 937 [907, 962] | 969 [966, 973] | 980 [977, 985] |
| walker-walk | 766 [489, 957] | 47 [40, 56] | 958 [952, 965] | 942 [936, 949] | 970 [968, 973] |
| Mean | 510 [497, 523] | 110 [98, 125] | 492 [471, 512] | 463 [452, 475] | 602 [595, 608] |
| Median | 626 [528, 665] | 49 [32, 53] | 572 [419, 654] | 493 [420, 532] | 813 [779, 822] |
| IQM | 545 [519, 564] | 58 [46, 67] | 501 [458, 537] | 452 [430, 473] | 692 [678, 703] |

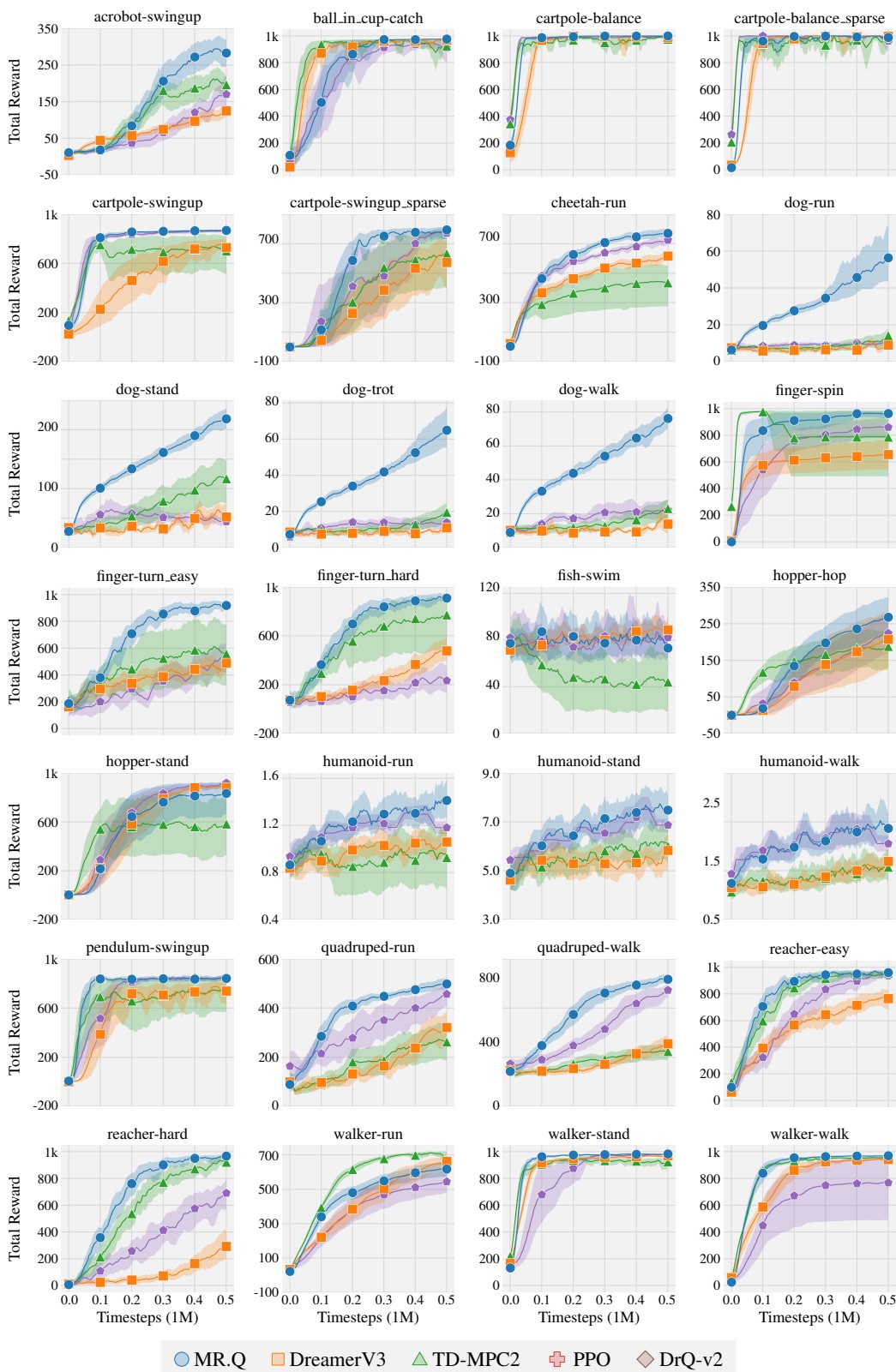

Figure 5: **DMC - Visual learning curves.** Time steps consider the number of environment interactions, where 500k time steps equals 1M frames in the original environment. Results are over 10 seeds. The shaded area captures a 95% boostrap confidence interval.

## C.4 ATARI

Table 7: **Atari final results.** Final average performance at 2.5M time steps (10M time steps in the original environment due to action repeat) over 10 seeds. The [bracketed values] represent a 95% bootstrap confidence interval. The aggregate mean, median and interquartile mean (IQM) are computed over the human-normalized score.

| Task | DQN | Rainbow | PPO | DreamerV3 | MR.Q |
|---|---|---|---|---|---|
| Alien | 925 [879, 968] | 1220 [1191, 1268] | 320 [251, 383] | 4838 [3863, 5813] | 2834 [2241, 3388] |
| Amidar | 178 [169, 186] | 301 [280, 330] | 126 [90, 167] | 470 [419, 524] | 595 [525, 657] |
| Assault | 988 [957, 1011] | 1430 [1392, 1475] | 423 [271, 581] | 3518 [2969, 4179] | 1296 [1254, 1343] |
| Asterix | 2381 [2313, 2469] | 2699 [2598, 2783] | 296 [216, 403] | 7319 [6251, 8354] | 3358 [3004, 3797] |
| Asteroids | 423 [408, 436] | 754 [711, 816] | 206 [180, 232] | 1359 [1243, 1482] | 715 [638, 796] |
| Atlantis | 7365 [6893, 7742] | 80837 [51139, 126780] | 2000 [2000, 2000] | 664529 [197588, 973362] | 556845 [469425, 660043] |
| BankHeist | 474 [448, 493] | 895 [889, 901] | 187 [41, 421] | 801 [691, 1002] | 809 [639, 960] |
| BattleZone | 3598 [3235, 3878] | 20209 [17157, 22375] | 2200 [1460, 3100] | 22599 [21055, 24669] | 19880 [13450, 26060] |
| BeamRider | 869 [728, 1065] | 5982 [5664, 6268] | 479 [348, 581] | 5635 [3161, 7962] | 2299 [1921, 2813] |
| Berzerk | 488 [466, 508] | 443 [413, 484] | 384 [310, 469] | 758 [681, 823] | 523 [456, 588] |
| Bowling | 29 [27, 32] | 44 [36, 52] | 51 [38, 60] | 101 [69, 138] | 59 [45, 72] |
| Boxing | 37 [31, 44] | 68 [66, 71] | -3 [-6, 0] | 97 [97, 99] | 96 [95, 97] |
| Breakout | 21 [19, 25] | 41 [40, 44] | 9 [8, 11] | 137 [110, 162] | 34 [28, 42] |
| Centipede | 2832 [2418, 3215] | 4992 [4784, 5138] | 4239 [2222, 6622] | 20067 [17410, 22758] | 17835 [16161, 19817] |
| ChopperCommand | 997 [971, 1022] | 2265 [2160, 2357] | 688 [501, 878] | 15172 [12940, 17219] | 5748 [4822, 6651] |
| CrazyClimber | 64611 [46203, 78709] | 103539 [99749, 106850] | 896 [174, 1727] | 132811 [128446, 135930] | 116954 [111371, 122032] |
| Defender | 116954 [111371, 122032] | 116954 [111371, 122032] | 1333 [705, 2094] | 34187 [29814, 39261] | 40457 [36892, 43638] |
| DemonAttack | 1503 [1282, 1690] | 2477 [2269, 2678] | 139 [116, 165] | 4836 [3443, 6231] | 5924 [4491, 7289] |
| DoubleDunk | -18 [-20, -18] | -18 [-19, -19] | -1 [-3, 0] | 21 [20, 22] | -10 [-15, -9] |
| Enduro | 589 [567, 617] | 1601 [1555, 1635] | 13 [9, 17] | 476 [175, 782] | 1845 [1758, 1938] |
| FishingDerby | -42 [-62, -17] | 10 [5, 15] | -89 [-91, -87] | 40 [32, 47] | 10 [2, 18] |
| Freeway | 8 [0, 19] | 32 [32, 32] | 15 [11, 18] | 19 [6, 32] | 32 [32, 32] |
| Frostbite | 269 [238, 294] | 2510 [2040, 2823] | 245 [231, 259] | 5183 [2151, 8291] | 4561 [3299, 5740] |
| Gopher | 1470 [1316, 1590] | 4279 [4139, 4425] | 126 [80, 174] | 38711 [26066, 48187] | 19174 [14932, 23587] |
| Gravitar | 167 [153, 183] | 202 [184, 218] | 63 [31, 98] | 831 [768, 900] | 397 [320, 490] |
| Hero | 2679 [2404, 2945] | 9323 [7914, 10863] | 1741 [1062, 2302] | 20582 [19845, 21583] | 13450 [11915, 14781] |
| IceHockey | -9 [-10, -9] | -5 [-6, -5] | -8 [-10, -8] | 14 [13, 16] | 0 [-1, 2] |
| Jamesbond | 47 [42, 52] | 514 [509, 520] | 85 [62, 106] | 836 [568, 1119] | 624 [588, 662] |
| Kangaroo | 539 [525, 553] | 5501 [3853, 7151] | 402 [280, 520] | 8825 [5234, 12418] | 9807 [7851, 11591] |
| Krull | 4229 [3942, 4490] | 5972 [5903, 6047] | 421 [136, 735] | 23092 [14679, 28172] | 9309 [8646, 9953] |
| KungFuMaster | 15997 [13182, 18813] | 18074 [16041, 20864] | 52 [18, 95] | 70703 [50114, 94578] | 29369 [26954, 31595] |
| MontezumaRevenge | 0 [0, 0] | 0 [0, 0] | 0 [0, 0] | 1310 [900, 2180] | 50 [0, 140] |
| MsPacman | 2187 [2121, 2247] | 2347 [2292, 2403] | 457 [352, 578] | 4484 [3539, 5511] | 4922 [4191, 5843] |
| NameThisGame | 4000 [3814, 4187] | 8604 [8252, 8931] | 1084 [663, 1501] | 15742 [14542, 17103] | 8693 [8071, 9199] |
| Phoenix | 4948 [4236, 5627] | 4830 [4707, 4968] | 101 [81, 120] | 15827 [14903, 16429] | 5173 [5025, 5322] |
| Pitfall | -60 [-89, -35] | -14 [-29, -6] | -16 [-38, -2] | 0 [0, 0] | -20 [-60, 0] |
| Pong | -4 [-14, 3] | 15 [14, 16] | -5 [-8, -3] | 16 [16, 17] | 17 [16, 19] |
| PrivateEye | 118 [78, 181] | 111 [78, 166] | -17 [-592, 762] | 3046 [975, 5118] | 100 [100, 100] |
| Qbert | 1658 [1246, 2139] | 5353 [4363, 6783] | 484 [393, 570] | 16807 [16073, 17564] | 3938 [3210, 4327] |
| Riverraid | 3198 [3167, 3222] | 4272 [4060, 4440] | 1045 [833, 1241] | 9160 [8177, 10077] | 10791 [9307, 12511] |
| RoadRunner | 27980 [27269, 28692] | 33412 [32459, 34435] | 723 [454, 940] | 66453 [40606, 104163] | 49579 [47425, 51426] |
| Robotank | 4 [4, 5] | 19 [18, 20] | 4 [2, 6] | 51 [47, 55] | 13 [12, 15] |
| Seaquest | 299 [277, 318] | 1641 [1621, 1661] | 250 [214, 282] | 3416 [2665, 4426] | 3522 [2401, 4850] |
| Skiing | -19568 [-19793, -19362] | -24070 [-25305, -22667] | -27901 [-30000, -23704] | -30043 [-30394, -29764] | -30000 [-30000, -30000] |
| Solaris | 1645 [1480, 1804] | 1289 [1143, 1451] | 0 [0, 2] | 2340 [1882, 2799] | 1103 [799, 1430] |
| SpaceInvaders | 663 [651, 675] | 743 [721, 764] | 294 [235, 354] | 1433 [1039, 1943] | 701 [626, 768] |
| StarGunner | 692 [662, 719] | 1488 [1470, 1506] | 415 [316, 499] | 2090 [1678, 2649] | 3488 [1032, 8241] |
| Surround | 3488 [1032, 8241] | 3488 [1032, 8241] | -9 [-10, -10] | 5 [4, 7] | -2 [-4, -2] |
| Tennis | -21 [-24, -19] | -1 [-2, 0] | -20 [-22, -19] | -3 [-11, 0] | 0 [0, 0] |
| TimePilot | 1539 [1479, 1613] | 2703 [2627, 2787] | 548 [450, 690] | 7779 [3128, 13016] | 4382 [4208, 4528] |
| Tutankham | 112 [97, 123] | 179 [165, 191] | 29 [17, 43] | 253 [240, 269] | 164 [145, 185] |
| UpNDown | 7669 [7116, 8147] | 12397 [11489, 13312] | 595 [428, 737] | 284807 [178615, 391388] | 73095 [40836, 108810] |
| Venture | 25 [6, 45] | 19 [14, 25] | 2 [0, 6] | 0 [0, 0] | 112 [0, 304] |
| VideoPinball | 5129 [4611, 5649] | 26245 [23075, 29067] | 1005 [0, 2485] | 22345 [20669, 23955] | 53826 [40600, 67972] |
| WizardOfWor | 481 [396, 542] | 2213 [1827, 2617] | 225 [185, 264] | 7086 [6518, 7730] | 2599 [2259, 2942] |
| YarsRevenge | 9426 [9177, 9656] | 10708 [10405, 11071] | 1891 [925, 2964] | 62209 [57783, 67113] | 34861 [29734, 40020] |
| Zaxxon | 112 [15, 230] | 3661 [3131, 4192] | 0 [0, 0] | 17347 [15320, 19385] | 8850 [8045, 9740] |
| Mean | 0.25 [0.24, 0.26] | 1.08 [1.02, 1.14] | -0.09 [-0.10, -0.07] | 3.74 [3.29, 4.13] | 2.54 [2.34, 2.75] |
| Median | 0.12 [0.10, 0.12] | 0.40 [0.40, 0.47] | 0.01 [0.00, 0.01] | 1.25 [1.11, 1.47] | 0.96 [0.78, 0.98] |
| IQM | 0.17 [0.16, 0.17] | 0.61 [0.60, 0.62] | 0.02 [0.01, 0.02] | 1.46 [1.34, 1.51] | 0.90 [0.88, 0.94] |

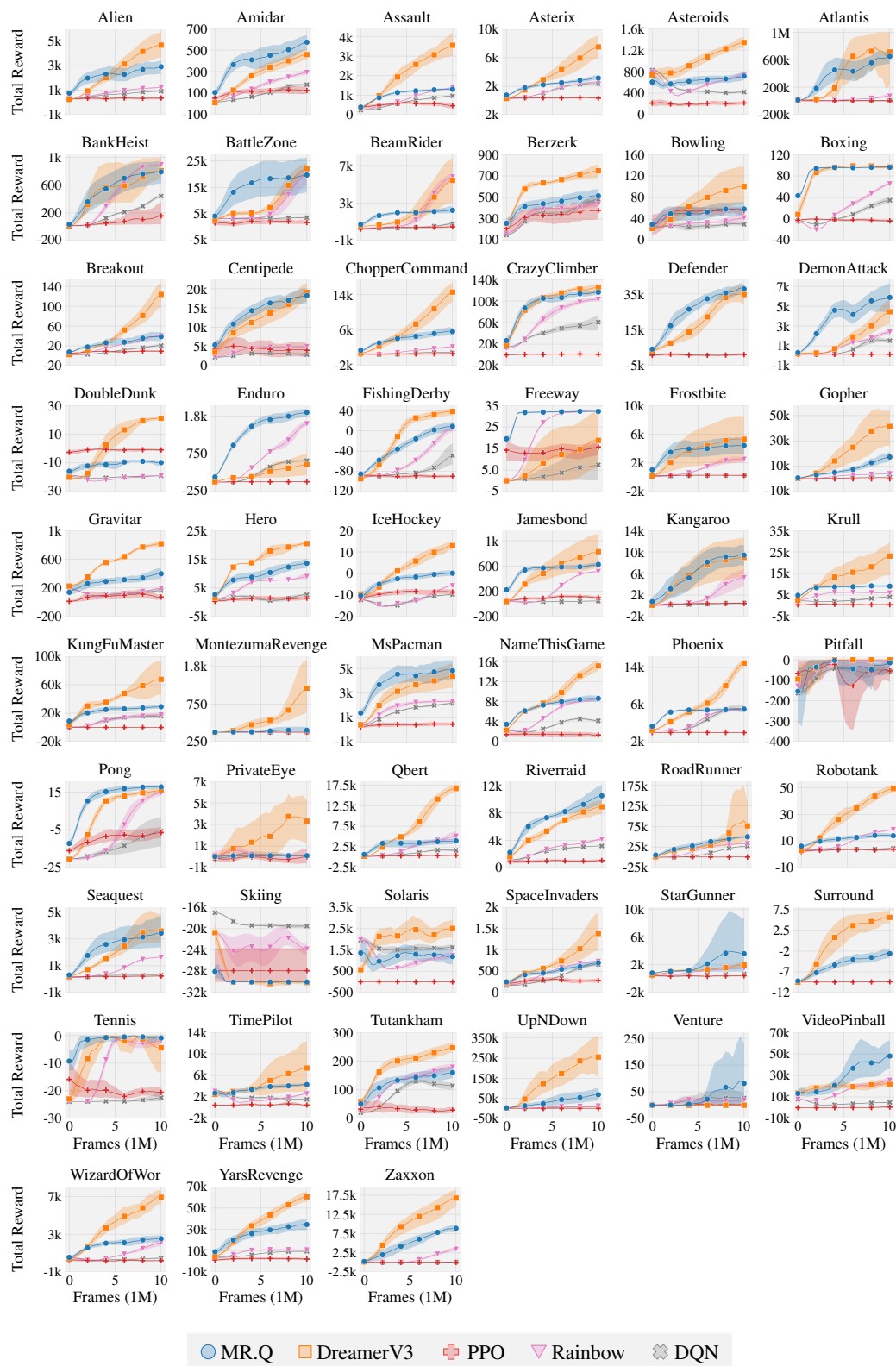

Figure 6: **Atari learning curves.** Time steps consider the number of environment interactions, where 2.5M time steps equals 10M frames in the original environment. Results are over 10 seeds. The shaded area captures a 95% boostrap confidence interval.

# D   COMPLETE ABLATION RESULTS

In this section, we show a per-environment breakdown of each variation in the design study in Section 5.2. Each table reports the raw score for each environment. The [bracketed values] represent a 95% bootstrap confidence interval. The aggregate mean, median and interquartile mean (IQM) are computed over the the difference in the normalized score. We use TD3 to normalize for Gym, raw scores divided by 1000 for DMC and human scores to normalize for Atari (see Appendix B.3). Highlighting is used to designate the scale of the difference in normalized score:

- $(\leq -0.5)$
- $[-0.2, -0.5)$
- $[-0.01, -0.2)$
- $[0.01, 0.2)$
- $[0.2, 0.5)$
- $(\geq 0.5)$

## D.1   GYM

| Task | MR.Q | Linear value function | Dynamics target | No target encoder |
|---|---|---|---|---|
| Ant | 6901 [6261, 7482] | 1844 [1663, 2018] | 5867 [5543, 6289] | 3970 [2468, 5509] |
| HalfCheetah | 12939 [11663, 13762] | 3383 [3054, 3732] | 14019 [13746, 14285] | 12838 [12459, 13266] |
| Hopper | 2692 [2131, 3309] | 968 [720, 1210] | 2890 [2030, 3747] | 3007 [2164, 3852] |
| Humanoid | 10223 [9929, 10498] | 461 [395, 532] | 8370 [7651, 8988] | 305 [272, 356] |
| Walker2d | 6039 [5644, 6386] | 1117 [999, 1238] | 5844 [5146, 6477] | 5944 [5570, 6323] |
| Mean | - | -1.17 [-1.19, -1.15] | -0.10 [-0.17, -0.04] | -0.53 [-0.60, -0.46] |
| Median | - | -1.25 [-1.28, -1.21] | -0.05 [-0.23, 0.09] | -0.02 [-0.16, 0.02] |
| IQM | - | -1.13 [-1.15, -1.11] | -0.08 [-0.19, 0.01] | -0.25 [-0.37, -0.16] |

| Task | MR.Q | Revert | Non-linear model | MSE reward loss |
|---|---|---|---|---|
| Ant | 6901 [6261, 7482] | -422 [-1770, 846] | 7215 [6971, 7466] | 7153 [5991, 7815] |
| HalfCheetah | 12939 [11663, 13762] | -658 [-750, -604] | 13370 [12649, 14053] | 14413 [14096, 14710] |
| Hopper | 2692 [2131, 3309] | 103 [39, 189] | 2492 [1835, 3424] | 2869 [2090, 3689] |
| Humanoid | 10223 [9929, 10498] | 189 [104, 277] | 10257 [9612, 10688] | 10592 [10017, 10983] |
| Walker2d | 6039 [5644, 6386] | 260 [-5, 638] | 5548 [4980, 6117] | 6626 [5256, 7984] |
| Mean | - | -1.47 [-1.54, -1.39] | -0.01 [-0.07, 0.03] | 0.10 [-0.02, 0.19] |
| Median | - | -1.47 [-1.53, -1.37] | 0.01 [-0.03, 0.08] | 0.07 [0.00, 0.17] |
| IQM | - | -1.51 [-1.58, -1.39] | -0.01 [-0.05, 0.04] | 0.09 [-0.01, 0.18] |

| Task | MR.Q | No reward scaling | No min | No LAP |
|---|---|---|---|---|
| Ant | 6901 [6261, 7482] | 6866 [6227, 7547] | 6936 [6582, 7329] | 6817 [6616, 7039] |
| HalfCheetah | 12939 [11663, 13762] | 13502 [13333, 13673] | 14143 [13819, 14515] | 13185 [13085, 13299] |
| Hopper | 2692 [2131, 3309] | 2551 [2090, 3064] | 2113 [1728, 2626] | 2681 [1883, 3465] |
| Humanoid | 10223 [9929, 10498] | 9515 [8520, 10245] | 10528 [10202, 10837] | 8441 [6206, 9738] |
| Walker2d | 6039 [5644, 6386] | 5743 [5362, 6102] | 4293 [3547, 5107] | 5463 [4134, 6376] |
| Mean | - | -0.04 [-0.09, 0.02] | -0.09 [-0.16, -0.01] | -0.10 [-0.24, -0.00] |
| Median | - | -0.04 [-0.11, 0.04] | 0.01 [-0.08, 0.07] | -0.02 [-0.12, 0.02] |
| IQM | - | -0.04 [-0.10, 0.03] | -0.04 [-0.10, 0.02] | -0.06 [-0.18, 0.00] |

| Task | MR.Q | No MR | 1-step return | No unroll |
|---|---|---|---|---|
| Ant | 6901 [6261, 7482] | 4195 [2573, 5819] | 7757 [7729, 7799] | 7528 [7224, 7830] |
| HalfCheetah | 12939 [11663, 13762] | 11249 [9238, 12495] | 13123 [10691, 14653] | 14409 [13817, 15002] |
| Hopper | 2692 [2131, 3309] | 1877 [1524, 2153] | 2737 [2131, 3343] | 2578 [1857, 3414] |
| Humanoid | 10223 [9929, 10498] | 3942 [3262, 4624] | 2328 [1491, 3337] | 10617 [10504, 10731] |
| Walker2d | 6039 [5644, 6386] | 4155 [3251, 4897] | 4747 [3197, 6229] | 6077 [5752, 6355] |
| Mean | - | -0.56 [-0.69, -0.43] | -0.33 [-0.46, -0.21] | 0.07 [0.01, 0.14] |
| Median | - | -0.48 [-0.71, -0.27] | 0.01 [-0.21, 0.16] | 0.08 [0.06, 0.16] |
| IQM | - | -0.47 [-0.66, -0.28] | -0.10 [-0.32, 0.12] | 0.07 [0.04, 0.15] |

## D.2 DMC - PROPRIOCEPTIVE

| Task | MR.Q | Linear value function | Dynamics target | No target encoder |
|---|---|---|---|---|
| acrobot-swingup | 567 [517, 621] | 30 [15, 46] | 626 [578, 684] | 16 [9, 25] |
| ball_in_cup-catch | 981 [979, 983] | 820 [658, 922] | 980 [978, 983] | 569 [436, 719] |
| cartpole-balance | 999 [999, 1000] | 449 [380, 520] | 999 [999, 1000] | 992 [986, 997] |
| cartpole-balance_sparse | 1000 [1000, 1000] | 183 [142, 224] | 1000 [1000, 1000] | 1000 [1000, 1000] |
| cartpole-swingup | 866 [866, 866] | 267 [225, 310] | 869 [866, 876] | 852 [840, 861] |
| cartpole-swingup_sparse | 798 [779, 818] | 12 [5, 22] | 817 [809, 832] | 0 [0, 0] |
| cheetah-run | 914 [911, 917] | 394 [376, 411] | 919 [919, 921] | 904 [899, 910] |
| dog-run | 569 [546, 595] | 11 [5, 18] | 254 [202, 305] | 11 [8, 16] |
| dog-stand | 967 [959, 975] | 22 [17, 29] | 672 [520, 830] | 36 [27, 51] |
| dog-trot | 877 [845, 898] | 15 [11, 20] | 319 [279, 365] | 12 [8, 19] |
| dog-walk | 916 [908, 924] | 13 [11, 18] | 312 [247, 396] | 10 [7, 15] |
| finger-spin | 937 [917, 958] | 736 [670, 825] | 942 [916, 971] | 869 [698, 963] |
| finger-turn_easy | 953 [928, 975] | 238 [157, 319] | 947 [900, 979] | 624 [509, 756] |
| finger-turn_hard | 950 [908, 974] | 23 [1, 63] | 923 [878, 966] | 431 [301, 563] |
| fish-swim | 792 [772, 811] | 83 [65, 102] | 410 [307, 493] | 97 [65, 129] |
| hopper-hop | 251 [201, 295] | 10 [4, 17] | 199 [131, 265] | 0 [0, 1] |
| hopper-stand | 951 [948, 955] | 66 [30, 102] | 639 [413, 866] | 4 [3, 7] |
| humanoid-run | 200 [169, 236] | 1 [1, 1] | 1 [1, 1] | 1 [1, 1] |
| humanoid-stand | 868 [823, 907] | 6 [5, 7] | 7 [6, 8] | 7 [6, 9] |
| humanoid-walk | 662 [609, 721] | 1 [1, 2] | 2 [2, 3] | 2 [2, 3] |
| pendulum-swingup | 748 [594, 830] | 357 [114, 617] | 826 [812, 840] | 784 [706, 834] |
| quadruped-run | 947 [940, 954] | 172 [93, 252] | 942 [930, 951] | 829 [757, 895] |
| quadruped-walk | 963 [959, 968] | 91 [52, 141] | 939 [935, 943] | 952 [946, 960] |
| reacher-easy | 983 [983, 985] | 802 [722, 877] | 984 [983, 986] | 983 [981, 986] |
| reacher-hard | 977 [975, 979] | 853 [778, 914] | 970 [965, 974] | 975 [972, 979] |
| walker-run | 793 [766, 816] | 238 [207, 274] | 730 [585, 814] | 776 [757, 792] |
| walker-stand | 988 [987, 990] | 859 [780, 921] | 988 [985, 991] | 988 [986, 991] |
| walker-walk | 978 [978, 980] | 504 [397, 613] | 975 [975, 977] | 974 [973, 976] |
| Mean | - | -0.58 [-0.59, -0.56] | -0.15 [-0.15, -0.15] | -0.35 [-0.35, -0.34] |
| Median | - | -0.58 [-0.64, -0.57] | -0.01 [-0.02, -0.00] | -0.22 [-0.23, -0.21] |
| IQM | - | -0.62 [-0.64, -0.60] | -0.05 [-0.06, -0.03] | -0.27 [-0.29, -0.25] |

| Task | MR.Q | Revert | Non-linear model | MSE reward loss |
|---|---|---|---|---|
| acrobot-swingup | 567 [517, 621] | 11 [8, 18] | 553 [478, 629] | 577 [547, 612] |
| ball_in_cup-catch | 981 [979, 983] | 301 [233, 365] | 982 [982, 984] | 983 [982, 985] |
| cartpole-balance | 999 [999, 1000] | 272 [206, 332] | 999 [999, 1000] | 999 [998, 1000] |
| cartpole-balance_sparse | 1000 [1000, 1000] | 197 [177, 214] | 1000 [1000, 1000] | 1000 [1000, 1000] |
| cartpole-swingup | 866 [866, 866] | 191 [99, 263] | 866 [866, 867] | 865 [865, 866] |
| cartpole-swingup_sparse | 798 [779, 818] | 0 [0, 0] | 824 [809, 839] | 812 [786, 834] |
| cheetah-run | 914 [911, 917] | 74 [31, 123] | 909 [906, 913] | 910 [907, 915] |
| dog-run | 569 [546, 595] | 3 [4, 4] | 588 [516, 646] | 527 [513, 545] |
| dog-stand | 967 [959, 975] | 22 [18, 29] | 962 [938, 982] | 964 [958, 971] |
| dog-trot | 877 [845, 898] | 4 [3, 5] | 868 [816, 914] | 861 [831, 886] |
| dog-walk | 916 [908, 924] | 4 [4, 6] | 920 [915, 925] | 724 [473, 882] |
| finger-spin | 937 [917, 958] | 0 [0, 1] | 868 [767, 951] | 907 [875, 947] |
| finger-turn_easy | 953 [928, 975] | 159 [60, 280] | 972 [968, 978] | 935 [894, 977] |
| finger-turn_hard | 950 [908, 974] | 60 [20, 100] | 931 [887, 975] | 947 [910, 969] |
| fish-swim | 792 [772, 811] | 71 [53, 90] | 790 [754, 824] | 793 [766, 821] |
| hopper-hop | 251 [201, 295] | 0 [0, 1] | 288 [222, 332] | 174 [119, 230] |
| hopper-stand | 951 [948, 955] | 5 [3, 8] | 848 [664, 978] | 854 [707, 936] |
| humanoid-run | 200 [169, 236] | 1 [1, 1] | 205 [191, 221] | 91 [45, 121] |
| humanoid-stand | 868 [823, 907] | 7 [7, 8] | 811 [712, 878] | 214 [14, 566] |
| humanoid-walk | 662 [609, 721] | 1 [2, 2] | 668 [590, 734] | 77 [3, 224] |
| pendulum-swingup | 748 [594, 830] | 61 [20, 103] | 819 [802, 836] | 827 [811, 843] |
| quadruped-run | 947 [940, 954] | 87 [40, 145] | 944 [934, 954] | 949 [941, 956] |
| quadruped-walk | 963 [959, 968] | 81 [36, 130] | 963 [961, 967] | 966 [961, 971] |
| reacher-easy | 983 [983, 985] | 789 [727, 856] | 983 [982, 984] | 964 [927, 984] |
| reacher-hard | 977 [975, 979] | 526 [361, 695] | 953 [914, 976] | 976 [974, 978] |
| walker-run | 793 [766, 816] | 25 [22, 31] | 795 [784, 811] | 778 [709, 821] |
| walker-stand | 988 [987, 990] | 221 [179, 264] | 983 [971, 991] | 988 [987, 990] |
| walker-walk | 978 [978, 980] | 33 [23, 46] | 974 [972, 978] | 967 [948, 979] |
| Mean | - | -0.72 [-0.73, -0.72] | -0.00 [-0.02, 0.01] | -0.06 [-0.08, -0.05] |
| Median | - | -0.78 [-0.80, -0.76] | -0.00 [-0.00, 0.00] | -0.00 [-0.01, 0.00] |
| IQM | - | -0.78 [-0.78, -0.76] | -0.00 [-0.01, 0.00] | -0.01 [-0.02, -0.00] |

| Task | MR.Q | No reward scaling | No min | No LAP |
|------|------|-------------------|--------|--------|
| acrobot-swingup | 567 [517, 621] | 593 [532, 672] | 623 [572, 673] | 566 [520, 612] |
| ball_in_cup-catch | 981 [979, 983] | 983 [982, 984] | 982 [980, 984] | 981 [980, 984] |
| cartpole-balance | 999 [999, 1000] | 998 [999, 999] | 999 [1000, 1000] | 999 [998, 1000] |
| cartpole-balance_sparse | 1000 [1000, 1000] | 1000 [1000, 1000] | 1000 [1000, 1000] | 992 [982, 1000] |
| cartpole-swingup | 866 [866, 866] | 865 [864, 866] | 868 [866, 874] | 865 [865, 866] |
| cartpole-swingup_sparse | 798 [779, 818] | 647 [318, 822] | 799 [780, 812] | 796 [779, 810] |
| cheetah-run | 914 [911, 917] | 911 [909, 914] | 910 [893, 921] | 908 [905, 913] |
| dog-run | 569 [546, 595] | 586 [546, 613] | 577 [540, 610] | 536 [499, 573] |
| dog-stand | 967 [959, 975] | 959 [940, 979] | 946 [917, 969] | 971 [966, 976] |
| dog-trot | 877 [845, 898] | 817 [713, 903] | 846 [767, 906] | 842 [764, 897] |
| dog-walk | 916 [908, 924] | 901 [890, 917] | 747 [447, 908] | 899 [886, 914] |
| finger-spin | 937 [917, 958] | 873 [768, 947] | 926 [907, 950] | 915 [892, 948] |
| finger-turn_easy | 953 [928, 975] | 977 [973, 982] | 976 [972, 980] | 975 [967, 983] |
| finger-turn_hard | 950 [908, 974] | 946 [905, 969] | 894 [833, 953] | 949 [909, 972] |
| fish-swim | 792 [772, 811] | 745 [663, 809] | 785 [763, 810] | 788 [754, 826] |
| hopper-hop | 251 [201, 295] | 343 [263, 477] | 336 [322, 352] | 347 [265, 431] |
| hopper-stand | 951 [948, 955] | 934 [912, 948] | 935 [926, 947] | 941 [935, 948] |
| humanoid-run | 200 [169, 236] | 184 [149, 214] | 198 [175, 225] | 202 [191, 212] |
| humanoid-stand | 868 [823, 907] | 810 [655, 899] | 833 [793, 871] | 880 [856, 900] |
| humanoid-walk | 662 [609, 721] | 665 [589, 765] | 597 [292, 808] | 697 [561, 828] |
| pendulum-swingup | 748 [594, 830] | 816 [790, 838] | 825 [811, 839] | 815 [792, 836] |
| quadruped-run | 947 [940, 954] | 951 [944, 958] | 946 [941, 951] | 937 [927, 947] |
| quadruped-walk | 963 [959, 968] | 966 [961, 971] | 959 [942, 972] | 955 [942, 967] |
| reacher-easy | 983 [983, 985] | 964 [926, 984] | 983 [981, 986] | 983 [982, 986] |
| reacher-hard | 977 [975, 979] | 971 [968, 975] | 978 [974, 982] | 974 [969, 981] |
| walker-run | 793 [766, 816] | 804 [783, 820] | 806 [779, 821] | 812 [803, 822] |
| walker-stand | 988 [987, 990] | 989 [988, 990] | 989 [986, 992] | 986 [985, 987] |
| walker-walk | 978 [978, 980] | 979 [978, 980] | 978 [976, 980] | 977 [974, 980] |
| Mean | - | -0.01 [-0.02, 0.00] | -0.01 [-0.02, 0.01] | 0.00 [-0.00, 0.01] |
| Median | - | -0.00 [-0.00, 0.00] | -0.00 [-0.00, 0.00] | -0.00 [-0.00, 0.00] |
| IQM | - | -0.00 [-0.00, 0.00] | -0.00 [-0.00, 0.00] | -0.00 [-0.00, 0.00] |

| Task | MR.Q | No MR | 1-step return | No unroll |
|------|------|-------|---------------|-----------|
| acrobot-swingup | 567 [517, 621] | 576 [483, 665] | 440 [360, 528] | 515 [455, 598] |
| ball_in_cup-catch | 981 [979, 983] | 981 [980, 984] | 984 [983, 985] | 982 [981, 984] |
| cartpole-balance | 999 [999, 1000] | 994 [991, 999] | 999 [1000, 1000] | 999 [999, 1000] |
| cartpole-balance_sparse | 1000 [1000, 1000] | 1000 [1000, 1000] | 961 [886, 1000] | 1000 [1000, 1000] |
| cartpole-swingup | 866 [866, 866] | 870 [864, 878] | 881 [879, 882] | 864 [861, 867] |
| cartpole-swingup_sparse | 798 [779, 818] | 684 [528, 814] | 845 [845, 847] | 818 [811, 831] |
| cheetah-run | 914 [911, 917] | 871 [823, 907] | 922 [921, 924] | 909 [908, 911] |
| dog-run | 569 [546, 595] | 68 [63, 75] | 299 [196, 360] | 514 [473, 554] |
| dog-stand | 967 [959, 975] | 494 [452, 530] | 606 [344, 865] | 955 [944, 971] |
| dog-trot | 877 [845, 898] | 65 [49, 80] | 725 [679, 756] | 857 [833, 883] |
| dog-walk | 916 [908, 924] | 102 [81, 122] | 788 [739, 832] | 920 [905, 934] |
| finger-spin | 937 [917, 958] | 888 [731, 975] | 983 [977, 988] | 880 [781, 940] |
| finger-turn_easy | 953 [928, 975] | 947 [913, 974] | 980 [979, 982] | 950 [917, 976] |
| finger-turn_hard | 950 [908, 974] | 846 [756, 926] | 968 [958, 976] | 947 [907, 971] |
| fish-swim | 792 [772, 811] | 706 [683, 727] | 498 [323, 651] | 709 [618, 783] |
| hopper-hop | 251 [201, 295] | 85 [33, 142] | 364 [336, 394] | 297 [169, 442] |
| hopper-stand | 951 [948, 955] | 365 [233, 491] | 952 [947, 959] | 949 [944, 955] |
| humanoid-run | 200 [169, 236] | 1 [1, 2] | 190 [124, 241] | 192 [172, 214] |
| humanoid-stand | 868 [823, 907] | 201 [9, 517] | 753 [665, 838] | 858 [806, 913] |
| humanoid-walk | 662 [609, 721] | 84 [3, 247] | 761 [689, 827] | 675 [593, 772] |
| pendulum-swingup | 748 [594, 830] | 827 [812, 842] | 823 [807, 841] | 819 [793, 843] |
| quadruped-run | 947 [940, 954] | 871 [793, 933] | 945 [940, 950] | 950 [944, 955] |
| quadruped-walk | 963 [959, 968] | 951 [943, 962] | 962 [958, 968] | 962 [955, 969] |
| reacher-easy | 983 [983, 985] | 980 [979, 983] | 984 [984, 986] | 981 [976, 986] |
| reacher-hard | 977 [975, 979] | 949 [909, 974] | 980 [979, 983] | 954 [913, 978] |
| walker-run | 793 [766, 816] | 780 [769, 790] | 835 [827, 843] | 780 [702, 825] |
| walker-stand | 988 [987, 990] | 983 [980, 988] | 990 [990, 992] | 988 [988, 990] |
| walker-walk | 978 [978, 980] | 976 [975, 977] | 979 [977, 982] | 975 [969, 981] |
| Mean | - | -0.19 [-0.19, -0.18] | -0.04 [-0.05, -0.02] | -0.01 [-0.01, -0.00] |
| Median | - | -0.05 [-0.08, -0.01] | 0.00 [0.00, 0.00] | -0.00 [-0.00, 0.00] |
| IQM | - | -0.06 [-0.08, -0.05] | 0.00 [-0.00, 0.01] | -0.00 [-0.01, -0.00] |

## D.3 DMC - VISUAL

| Task | MR.Q | Linear value function | Dynamics target | No target encoder |
|---|---|---|---|---|
| acrobot-swingup | 287 [253, 317] | 15 [5, 22] | 296 [281, 323] | 16 [0, 38] |
| ball_in_cup-catch | 977 [975, 980] | 644 [328, 904] | 972 [965, 976] | 605 [496, 726] |
| cartpole-balance | 999 [999, 999] | 306 [254, 349] | 998 [998, 999] | 978 [947, 997] |
| cartpole-balance_sparse | 1000 [1000, 1000] | 243 [183, 327] | 1000 [1000, 1000] | 1000 [1000, 1000] |
| cartpole-swingup | 868 [861, 875] | 229 [181, 294] | 861 [859, 865] | 689 [487, 808] |
| cartpole-swingup_sparse | 797 [777, 818] | 4 [0, 14] | 267 [0, 801] | 0 [0, 0] |
| cheetah-run | 775 [752, 805] | 230 [159, 294] | 831 [761, 875] | 745 [723, 784] |
| dog-run | 60 [44, 80] | 19 [17, 22] | 36 [34, 39] | 10 [6, 20] |
| dog-stand | 216 [201, 233] | 76 [70, 82] | 191 [155, 247] | 60 [44, 89] |
| dog-trot | 65 [54, 79] | 19 [16, 24] | 46 [42, 53] | 9 [9, 10] |
| dog-walk | 77 [70, 83] | 30 [26, 33] | 62 [57, 72] | 16 [10, 21] |
| finger-spin | 965 [938, 982] | 789 [598, 923] | 786 [672, 931] | 929 [893, 981] |
| finger-turn_easy | 953 [925, 974] | 132 [98, 200] | 876 [691, 969] | 898 [855, 969] |
| finger-turn_hard | 932 [905, 957] | 66 [0, 100] | 859 [777, 963] | 492 [385, 577] |
| fish-swim | 79 [67, 93] | 69 [45, 109] | 71 [38, 106] | 65 [49, 84] |
| hopper-hop | 270 [229, 317] | 1 [0, 2] | 184 [165, 204] | 2 [0, 6] |
| hopper-stand | 852 [705, 932] | 7 [3, 11] | 911 [900, 922] | 5 [3, 9] |
| humanoid-run | 1 [1, 2] | 1 [1, 1] | 1 [1, 2] | 1 [1, 1] |
| humanoid-stand | 7 [7, 8] | 5 [4, 8] | 6 [5, 8] | 7 [6, 8] |
| humanoid-walk | 2 [2, 3] | 1 [1, 2] | 2 [1, 3] | 1 [1, 2] |
| pendulum-swingup | 829 [815, 842] | 97 [0, 192] | 749 [632, 840] | 191 [93, 287] |
| quadruped-run | 498 [474, 523] | 131 [77, 187] | 488 [468, 517] | 575 [566, 594] |
| quadruped-walk | 833 [796, 868] | 105 [57, 155] | 717 [445, 895] | 817 [790, 868] |
| reacher-easy | 979 [978, 982] | 605 [398, 868] | 977 [973, 981] | 979 [970, 986] |
| reacher-hard | 965 [945, 977] | 288 [195, 473] | 975 [971, 978] | 970 [963, 975] |
| walker-run | 615 [571, 655] | 158 [133, 193] | 531 [463, 577] | 611 [590, 631] |
| walker-stand | 980 [977, 984] | 707 [601, 881] | 982 [981, 984] | 984 [979, 988] |
| walker-walk | 970 [968, 973] | 350 [228, 529] | 904 [850, 951] | 965 [957, 972] |
| Mean | - | -0.41 [-0.42, -0.39] | -0.05 [-0.05, -0.04] | -0.15 [-0.15, -0.15] |
| Median | - | -0.37 [-0.44, -0.37] | -0.01 [-0.01, -0.00] | -0.03 [-0.05, -0.03] |
| IQM | - | -0.42 [-0.43, -0.38] | -0.02 [-0.02, -0.01] | -0.05 [-0.06, -0.04] |

| Task | MR.Q | Revert | Non-linear model | MSE reward loss |
|---|---|---|---|---|
| acrobot-swingup | 287 [253, 317] | 19 [13, 23] | 279 [235, 314] | 265 [242, 294] |
| ball_in_cup-catch | 977 [975, 980] | 195 [91, 297] | 973 [970, 977] | 974 [969, 978] |
| cartpole-balance | 999 [999, 999] | 190 [163, 212] | 999 [999, 999] | 998 [998, 1000] |
| cartpole-balance_sparse | 1000 [1000, 1000] | 346 [193, 639] | 1000 [1000, 1000] | 1000 [1000, 1000] |
| cartpole-swingup | 868 [861, 875] | 115 [82, 175] | 849 [819, 873] | 876 [875, 878] |
| cartpole-swingup_sparse | 797 [777, 818] | 0 [0, 0] | 768 [684, 824] | 33 [0, 60] |
| cheetah-run | 775 [752, 805] | 69 [36, 129] | 763 [757, 770] | 732 [712, 757] |
| dog-run | 60 [44, 80] | 3 [3, 4] | 37 [36, 40] | 36 [33, 38] |
| dog-stand | 216 [201, 233] | 17 [16, 19] | 200 [193, 207] | 195 [187, 208] |
| dog-trot | 65 [54, 79] | 5 [4, 6] | 51 [47, 56] | 47 [44, 50] |
| dog-walk | 77 [70, 83] | 6 [6, 8] | 69 [62, 78] | 63 [58, 71] |
| finger-spin | 965 [938, 982] | 1 [0, 2] | 907 [841, 971] | 924 [884, 980] |
| finger-turn_easy | 953 [925, 974] | 133 [99, 200] | 932 [889, 977] | 844 [786, 881] |
| finger-turn_hard | 932 [905, 957] | 66 [0, 100] | 938 [903, 967] | 900 [867, 964] |
| fish-swim | 79 [67, 93] | 53 [47, 60] | 67 [55, 79] | 75 [59, 105] |
| hopper-hop | 270 [229, 317] | 0 [0, 2] | 308 [257, 368] | 102 [14, 151] |
| hopper-stand | 852 [705, 932] | 4 [3, 8] | 935 [929, 942] | 919 [914, 925] |
| humanoid-run | 1 [1, 2] | 1 [1, 1] | 1 [1, 1] | 1 [1, 1] |
| humanoid-stand | 7 [7, 8] | 6 [5, 7] | 6 [6, 8] | 7 [7, 8] |
| humanoid-walk | 2 [2, 3] | 1 [1, 2] | 2 [2, 3] | 1 [1, 2] |
| pendulum-swingup | 829 [815, 842] | 66 [0, 100] | 820 [795, 844] | 581 [94, 842] |
| quadruped-run | 498 [474, 523] | 86 [69, 120] | 555 [514, 578] | 516 [478, 544] |
| quadruped-walk | 833 [796, 868] | 76 [34, 100] | 762 [727, 788] | 835 [751, 880] |
| reacher-easy | 979 [978, 982] | 583 [395, 684] | 939 [900, 979] | 979 [976, 984] |
| reacher-hard | 965 [945, 977] | 33 [0, 100] | 868 [753, 956] | 941 [879, 976] |
| walker-run | 615 [571, 655] | 35 [29, 39] | 612 [593, 633] | 596 [505, 643] |
| walker-stand | 980 [977, 984] | 280 [269, 294] | 982 [977, 987] | 983 [984, 984] |
| walker-walk | 970 [968, 973] | 22 [19, 25] | 951 [917, 972] | 969 [961, 976] |
| Mean | - | -0.52 [-0.52, -0.51] | -0.01 [-0.02, -0.00] | -0.05 [-0.07, -0.04] |
| Median | - | -0.68 [-0.72, -0.62] | -0.01 [-0.01, -0.00] | -0.01 [-0.01, 0.00] |
| IQM | - | -0.57 [-0.58, -0.56] | -0.01 [-0.01, -0.00] | -0.01 [-0.01, -0.00] |

| Task | MR.Q | No reward scaling | No min | No LAP |
|---|---|---|---|---|
| acrobot-swingup | 287 [253, 317] | 323 [275, 368] | 332 [301, 391] | 280 [235, 354] |
| ball_in_cup-catch | 977 [975, 980] | 973 [971, 977] | 975 [974, 976] | 971 [966, 978] |
| cartpole-balance | 999 [999, 999] | 999 [999, 999] | 998 [998, 999] | 998 [997, 999] |
| cartpole-balance_sparse | 1000 [1000, 1000] | 1000 [1000, 1000] | 1000 [1000, 1000] | 1000 [1000, 1000] |
| cartpole-swingup | 868 [861, 875] | 860 [829, 877] | 879 [879, 880] | 798 [785, 821] |
| cartpole-swingup_sparse | 797 [777, 818] | 813 [805, 823] | 805 [763, 829] | 764 [736, 799] |
| cheetah-run | 775 [752, 805] | 720 [678, 758] | 751 [734, 762] | 706 [670, 741] |
| dog-run | 60 [44, 80] | 61 [49, 73] | 42 [37, 51] | 62 [45, 91] |
| dog-stand | 216 [201, 233] | 317 [239, 387] | 228 [224, 232] | 279 [229, 315] |
| dog-trot | 65 [54, 79] | 65 [55, 79] | 50 [48, 53] | 58 [56, 61] |
| dog-walk | 77 [70, 83] | 89 [83, 96] | 86 [69, 106] | 91 [85, 101] |
| finger-spin | 965 [938, 982] | 903 [776, 975] | 870 [709, 982] | 940 [864, 979] |
| finger-turn_easy | 953 [925, 974] | 873 [775, 954] | 963 [952, 975] | 844 [785, 879] |
| finger-turn_hard | 932 [905, 957] | 923 [885, 962] | 933 [874, 976] | 932 [858, 974] |
| fish-swim | 79 [67, 93] | 73 [59, 87] | 54 [49, 61] | 63 [49, 89] |
| hopper-hop | 270 [229, 317] | 244 [204, 298] | 255 [244, 275] | 186 [152, 204] |
| hopper-stand | 852 [705, 932] | 911 [888, 926] | 923 [902, 945] | 884 [877, 896] |
| humanoid-run | 1 [1, 2] | 1 [1, 1] | 1 [1, 1] | 1 [1, 2] |
| humanoid-stand | 7 [7, 8] | 7 [6, 8] | 6 [5, 8] | 10 [6, 18] |
| humanoid-walk | 2 [2, 3] | 2 [2, 4] | 2 [2, 4] | 2 [2, 3] |
| pendulum-swingup | 829 [815, 842] | 823 [798, 846] | 831 [809, 843] | 829 [808, 841] |
| quadruped-run | 498 [474, 523] | 505 [471, 545] | 539 [500, 578] | 463 [428, 485] |
| quadruped-walk | 833 [796, 868] | 823 [781, 867] | 849 [745, 909] | 799 [713, 905] |
| reacher-easy | 979 [978, 982] | 962 [924, 983] | 953 [897, 981] | 948 [885, 980] |
| reacher-hard | 965 [945, 977] | 972 [970, 975] | 936 [872, 975] | 973 [973, 974] |
| walker-run | 615 [571, 655] | 600 [544, 632] | 666 [643, 682] | 662 [629, 725] |
| walker-stand | 980 [977, 984] | 986 [984, 989] | 982 [975, 988] | 984 [984, 985] |
| walker-walk | 970 [968, 973] | 970 [968, 974] | 970 [968, 972] | 972 [964, 979] |
| Mean | - | -0.00 [-0.01, 0.01] | 0.00 [-0.01, 0.01] | -0.01 [-0.02, -0.01] |
| Median | - | -0.00 [-0.00, 0.00] | 0.00 [-0.00, 0.00] | -0.00 [-0.01, 0.00] |
| IQM | - | -0.00 [-0.00, 0.00] | 0.00 [-0.00, 0.00] | -0.01 [-0.02, 0.00] |

| Task | MR.Q | No MR | 1-step return | No unroll |
|---|---|---|---|---|
| acrobot-swingup | 287 [253, 317] | 362 [305, 421] | 91 [76, 112] | 126 [77, 159] |
| ball_in_cup-catch | 977 [975, 980] | 898 [746, 977] | 980 [979, 982] | 976 [973, 980] |
| cartpole-balance | 999 [999, 999] | 998 [998, 999] | 998 [998, 1000] | 998 [999, 999] |
| cartpole-balance_sparse | 1000 [1000, 1000] | 1000 [1000, 1000] | 1000 [1000, 1000] | 1000 [1000, 1000] |
| cartpole-swingup | 868 [861, 875] | 871 [864, 878] | 858 [835, 875] | 872 [863, 879] |
| cartpole-swingup_sparse | 797 [777, 818] | 459 [139, 780] | 712 [685, 759] | 529 [0, 816] |
| cheetah-run | 775 [752, 805] | 782 [765, 805] | 675 [674, 677] | 753 [679, 845] |
| dog-run | 60 [44, 80] | 22 [21, 24] | 30 [21, 43] | 45 [36, 56] |
| dog-stand | 216 [201, 233] | 137 [127, 148] | 160 [143, 191] | 209 [195, 217] |
| dog-trot | 65 [54, 79] | 32 [29, 36] | 29 [25, 32] | 47 [46, 49] |
| dog-walk | 77 [70, 83] | 42 [36, 50] | 55 [33, 67] | 67 [63, 76] |
| finger-spin | 965 [938, 982] | 887 [757, 965] | 984 [978, 989] | 738 [588, 960] |
| finger-turn_easy | 953 [925, 974] | 694 [539, 805] | 942 [874, 979] | 869 [766, 962] |
| finger-turn_hard | 932 [905, 957] | 622 [436, 825] | 908 [872, 974] | 902 [780, 973] |
| fish-swim | 79 [67, 93] | 72 [60, 93] | 64 [58, 72] | 67 [55, 90] |
| hopper-hop | 270 [229, 317] | 192 [166, 216] | 248 [231, 280] | 242 [219, 270] |
| hopper-stand | 852 [705, 932] | 918 [897, 935] | 877 [820, 915] | 925 [907, 940] |
| humanoid-run | 1 [1, 2] | 1 [1, 2] | 1 [1, 2] | 1 [1, 1] |
| humanoid-stand | 7 [7, 8] | 7 [7, 8] | 7 [6, 9] | 7 [5, 9] |
| humanoid-walk | 2 [2, 3] | 2 [2, 3] | 2 [2, 3] | 2 [2, 3] |
| pendulum-swingup | 829 [815, 842] | 819 [787, 844] | 828 [811, 839] | 665 [382, 811] |
| quadruped-run | 498 [474, 523] | 478 [432, 515] | 456 [424, 476] | 398 [326, 465] |
| quadruped-walk | 833 [796, 868] | 701 [663, 731] | 666 [627, 720] | 730 [663, 769] |
| reacher-easy | 979 [978, 982] | 978 [976, 981] | 972 [948, 985] | 978 [977, 980] |
| reacher-hard | 965 [945, 977] | 545 [214, 858] | 978 [972, 982] | 893 [776, 967] |
| walker-run | 615 [571, 655] | 568 [538, 599] | 672 [639, 730] | 656 [619, 696] |
| walker-stand | 980 [977, 984] | 974 [962, 986] | 986 [982, 991] | 983 [981, 987] |
| walker-walk | 970 [968, 973] | 955 [948, 964] | 971 [967, 978] | 971 [971, 972] |
| Mean | - | -0.07 [-0.09, -0.03] | -0.03 [-0.03, -0.02] | -0.04 [-0.06, -0.01] |
| Median | - | -0.02 [-0.03, -0.01] | -0.01 [-0.01, -0.00] | -0.01 [-0.01, -0.00] |
| IQM | - | -0.03 [-0.03, -0.01] | -0.01 [-0.02, -0.01] | -0.02 [-0.02, -0.01] |

## D.4 ATARI

| Task | MR.Q | Linear value function | Dynamics target | No target encoder |
|---|---|---|---|---|
| Alien | 2471 [1848, 3155] | 596 [561, 631] | 1176 [1138, 1215] | 2040 [1585, 2495] |
| Amidar | 443 [376, 499] | 48 [38, 61] | 214 [182, 247] | 249 [232, 268] |
| Assault | 1125 [1094, 1160] | 366 [359, 374] | 911 [906, 917] | 1057 [880, 1234] |
| Asterix | 2216 [2081, 2346] | 810 [585, 1035] | 1940 [1865, 2015] | 2407 [1860, 2955] |
| Asteroids | 602 [493, 689] | 609 [595, 623] | 776 [716, 837] | 765 [591, 939] |
| Atlantis | 445022 [282338, 630730] | 17683 [13080, 20330] | 529745 [145750, 913740] | 76930 [76090, 77770] |
| BankHeist | 542 [348, 749] | 93 [76, 111] | 646 [326, 966] | 40 [38, 43] |
| BattleZone | 16520 [10560, 22760] | 11300 [11100, 11500] | 8250 [2100, 14400] | 4600 [2500, 6700] |
| BeamRider | 2007 [1855, 2194] | 584 [528, 642] | 1201 [1015, 1387] | 1468 [1298, 1639] |
| Berzerk | 430 [383, 472] | 315 [216, 415] | 381 [359, 403] | 359 [275, 444] |
| Bowling | 50 [37, 65] | 31 [31, 33] | 81 [81, 82] | 40 [33, 48] |
| Boxing | 95 [94, 97] | 39 [37, 42] | 90 [88, 93] | 92 [89, 96] |
| Breakout | 25 [21, 32] | 3 [2, 4] | 9 [8, 11] | 1 [1, 2] |
| Centipede | 14954 [13541, 16508] | 6709 [6207, 7213] | 7853 [7680, 8026] | 5167 [3661, 6674] |
| ChopperCommand | 4348 [3756, 5002] | 890 [780, 1000] | 3055 [2870, 3240] | 1385 [1060, 1710] |
| CrazyClimber | 104766 [99290, 109629] | 23016 [21610, 25010] | 92455 [84000, 100910] | 40240 [29150, 51330] |
| Defender | 25962 [23406, 29182] | 7825 [5640, 10010] | 17592 [9290, 25895] | 11627 [5745, 17510] |
| DemonAttack | 4660 [4072, 5241] | 1608 [1365, 1852] | 281 [242, 322] | 278 [204, 352] |
| DoubleDunk | -9 [-11, -9] | -18 [-24, -13] | -15 [-22, -10] | -20 [-21, -19] |
| Enduro | 1480 [1378, 1592] | 343 [319, 368] | 622 [621, 623] | 690 [667, 713] |
| FishingDerby | -34 [-43, -27] | -90 [-96, -86] | -64 [-66, -62] | -81 [-87, -75] |
| Freeway | 31 [31, 32] | 20 [19, 23] | 32 [33, 33] | 11 [0, 22] |
| Frostbite | 4003 [2871, 5163] | 198 [198, 198] | 268 [267, 269] | 824 [258, 1390] |
| Gopher | 4936 [3923, 5730] | 598 [524, 672] | 853 [832, 874] | 4371 [3794, 4948] |
| Gravitar | 275 [232, 322] | 190 [130, 250] | 352 [250, 455] | 60 [40, 80] |
| Hero | 8391 [6845, 10060] | 615 [112, 1118] | 7560 [7560, 7560] | 2200 [1377, 3024] |
| IceHockey | -2 [-3, -1] | -15 [-16, -15] | -6 [-10, -4] | -8 [-10, -6] |
| Jamesbond | 551 [534, 573] | 46 [40, 50] | 412 [310, 515] | 102 [80, 125] |
| Kangaroo | 4833 [2716, 7064] | 555 [520, 590] | 6830 [600, 13060] | 685 [590, 780] |
| Krull | 8660 [8198, 9147] | 6078 [5777, 6379] | 7460 [6961, 7959] | 9088 [8624, 9552] |
| KungFuMaster | 26150 [21973, 30490] | 10400 [9320, 11480] | 17020 [7520, 26520] | 12130 [11280, 12980] |
| MontezumaRevenge | 12 [0, 34] | 0 [0, 0] | 0 [0, 0] | 0 [0, 0] |
| MsPacman | 4395 [3799, 5002] | 826 [757, 896] | 2950 [2490, 3410] | 3171 [1873, 4469] |
| NameThisGame | 7511 [7085, 7911] | 2339 [2258, 2436] | 6660 [6568, 6752] | 7015 [6396, 7634] |
| Phoenix | 4843 [4635, 5033] | 570 [286, 854] | 3996 [3843, 4150] | 4260 [3944, 4577] |
| Pitfall | -8 [-19, -1] | -25 [-50, 0] | 0 [0, 0] | -66 [-122, -12] |
| Pong | 15 [13, 17] | -20 [-21, -20] | 14 [14, 16] | -10 [-19, -2] |
| PrivateEye | 100 [100, 100] | 45 [0, 90] | 100 [100, 100] | 90 [80, 100] |
| Qbert | 3600 [2554, 4366] | 256 [228, 285] | 493 [488, 500] | 747 [615, 880] |
| Riverraid | 7362 [7062, 7630] | 1997 [1829, 2165] | 6860 [6391, 7330] | 6342 [5450, 7234] |
| RoadRunner | 27152 [19731, 34480] | 3245 [2410, 4080] | 21835 [20960, 22710] | 35120 [33050, 37190] |
| Robotank | 10 [9, 13] | 7 [3, 11] | 7 [4, 10] | 6 [3, 9] |
| Seaquest | 2660 [2055, 3579] | 305 [214, 400] | 895 [834, 956] | 1227 [378, 2076] |
| Skiing | -30000 [-30000, -30000] | -30000 [-30000, -30000] | -30000 [-30000, -30000] | -30000 [-30000, -30000] |
| Solaris | 1262 [863, 1686] | 480 [0, 960] | 710 [378, 1042] | 1219 [1198, 1240] |
| SpaceInvaders | 478 [429, 524] | 242 [230, 255] | 303 [263, 344] | 296 [269, 323] |
| StarGunner | 1146 [996, 1437] | 1060 [880, 1240] | 960 [920, 1000] | 970 [960, 980] |
| Surround | -6 [-7, -5] | -9 [-9, -9] | -7 [-8, -7] | -7 [-9, -6] |
| Tennis | 0 [-1, 0] | -24 [-24, -24] | -5 [-10, 0] | 0 [0, 0] |
| TimePilot | 3101 [2772, 3482] | 2525 [1430, 3620] | 2525 [2040, 3010] | 1710 [1020, 2400] |
| Tutankham | 130 [124, 139] | 90 [86, 95] | 164 [150, 179] | 78 [2, 155] |
| UpNDown | 26477 [11956, 43260] | 3180 [2321, 4040] | 3045 [2667, 3424] | 4523 [4422, 4625] |
| Venture | 0 [0, 0] | 65 [0, 130] | 0 [0, 0] | 0 [0, 0] |
| VideoPinball | 18826 [15048, 23233] | 12994 [10570, 15419] | 9170 [7880, 10460] | 19734 [14479, 24989] |
| WizardOfWor | 1918 [1706, 2154] | 635 [480, 790] | 1260 [670, 1850] | 1440 [930, 1950] |
| YarsRevenge | 27299 [23434, 30493] | 6404 [6391, 6417] | 23613 [16984, 30244] | 21163 [21047, 21280] |
| Zaxxon | 3820 [2577, 4854] | 0 [0, 0] | 690 [0, 1380] | 0 [0, 0] |
| Mean | - | -1.35 [-1.41, -1.29] | -0.38 [-0.81, 0.05] | -0.86 [-0.89, -0.83] |
| Median | - | -0.42 [-0.55, -0.42] | -0.16 [-0.16, -0.11] | -0.18 [-0.19, -0.12] |
| IQM | - | -0.56 [-0.60, -0.55] | -0.20 [-0.22, -0.15] | -0.26 [-0.27, -0.25] |

| Task | MR.Q | Revert | Non-linear model | MSE reward loss |
|---|---|---|---|---|
| Alien | 2471 [1848, 3155] | 66 [32, 100] | 2167 [1426, 3169] | 734 [617, 856] |
| Amidar | 443 [376, 499] | 39 [19, 60] | 466 [364, 570] | 157 [140, 177] |
| Assault | 1125 [1094, 1160] | 366 [359, 374] | 1033 [998, 1068] | 923 [873, 987] |
| Asterix | 2216 [2081, 2346] | 492 [425, 560] | 1987 [1560, 2414] | 2503 [2050, 3040] |
| Asteroids | 602 [493, 689] | 249 [239, 259] | 563 [424, 740] | 765 [624, 952] |
| Atlantis | 445022 [282338, 630730] | 7310 [3140, 11480] | 444370 [241216, 647524] | 87410 [31610, 153510] |
| BankHeist | 542 [348, 749] | 14 [0, 28] | 1006 [961, 1042] | 245 [195, 309] |
| BattleZone | 16520 [10560, 22760] | 3550 [3300, 3800] | 23820 [20300, 26200] | 4566 [3900, 5200] |
| BeamRider | 2007 [1855, 2194] | 546 [510, 582] | 1904 [1777, 2046] | 1489 [1446, 1543] |
| Berzerk | 430 [383, 472] | 315 [275, 355] | 527 [498, 579] | 334 [295, 398] |
| Bowling | 50 [37, 65] | 41 [27, 55] | 69 [58, 82] | 30 [27, 35] |
| Boxing | 95 [94, 97] | 29 [21, 38] | 95 [91, 98] | 93 [89, 98] |
| Breakout | 25 [21, 32] | 2 [1, 3] | 17 [17, 18] | 11 [7, 18] |
| Centipede | 14954 [13541, 16508] | 2877 [2821, 2933] | 10053 [6514, 13594] | 11624 [8061, 16184] |
| ChopperCommand | 4348 [3756, 5002] | 530 [420, 640] | 2918 [2006, 3830] | 2806 [1610, 3570] |
| CrazyClimber | 104766 [99290, 109629] | 1700 [0, 3400] | 103950 [98066, 110282] | 107220 [104990, 109150] |
| Defender | 25962 [23406, 29182] | 1917 [1780, 2055] | 24283 [22936, 25425] | 15231 [7875, 21485] |
| DemonAttack | 4660 [4072, 5241] | 149 [147, 151] | 2467 [1370, 3548] | 311 [288, 331] |
| DoubleDunk | -9 [-11, -9] | -23 [-24, -23] | -10 [-14, -8] | -10 [-14, -9] |
| Enduro | 1480 [1378, 1592] | 3 [0, 6] | 1117 [1059, 1173] | 800 [694, 899] |
| FishingDerby | -34 [-43, -27] | -86 [-91, -82] | -33 [-36, -30] | -71 [-75, -66] |
| Freeway | 31 [31, 32] | 0 [0, 0] | 31 [31, 32] | 30 [30, 31] |
| Frostbite | 4003 [2871, 5163] | 170 [151, 190] | 2693 [834, 4491] | 3954 [266, 7285] |
| Gopher | 4936 [3923, 5730] | 385 [280, 490] | 7216 [3645, 11049] | 2484 [886, 3514] |
| Gravitar | 275 [232, 322] | 82 [80, 85] | 309 [156, 419] | 140 [50, 310] |
| Hero | 8391 [6845, 10060] | 0 [0, 0] | 7635 [7577, 7693] | 933 [0, 2799] |
| IceHockey | -2 [-3, -1] | -14 [-17, -11] | -1 [-2, -1] | -8 [-9, -7] |
| Jamesbond | 551 [534, 573] | 60 [45, 75] | 495 [462, 538] | 355 [280, 455] |
| Kangaroo | 4833 [2716, 7064] | 80 [0, 160] | 5732 [3148, 7954] | 2793 [540, 7060] |
| Krull | 8660 [8198, 9147] | 10 [0, 20] | 8396 [8178, 8593] | 8886 [7771, 9458] |
| KungFuMaster | 26150 [21973, 30490] | 1815 [130, 3500] | 24644 [19158, 30310] | 19536 [12710, 31840] |
| MontezumaRevenge | 12 [0, 34] | 0 [0, 0] | 240 [80, 400] | 0 [0, 0] |
| MsPacman | 4395 [3799, 5002] | 283 [182, 385] | 3721 [3169, 4499] | 1457 [1382, 1549] |
| NameThisGame | 7511 [7085, 7911] | 2675 [2609, 2742] | 6162 [5941, 6450] | 6091 [5656, 6475] |
| Phoenix | 4843 [4635, 5033] | 728 [517, 939] | 4611 [4300, 4800] | 3638 [3317, 3959] |
| Pitfall | -8 [-19, -1] | -1012 [-2000, -24] | -3 [-19, 0] | -22 [-68, 0] |
| Pong | 15 [13, 17] | -20 [-21, -21] | 16 [15, 19] | 13 [8, 17] |
| PrivateEye | 100 [100, 100] | 50 [20, 80] | 100 [100, 100] | 33 [0, 100] |
| Qbert | 3600 [2554, 4366] | 137 [125, 150] | 4295 [4006, 4586] | 861 [785, 968] |
| Riverraid | 7362 [7062, 7630] | 1660 [1131, 2190] | 6679 [5053, 7770] | 3418 [482, 6556] |
| RoadRunner | 27152 [19731, 34480] | 0 [0, 0] | 26678 [19418, 32016] | 24583 [15870, 35950] |
| Robotank | 10 [9, 13] | 2 [3, 3] | 10 [8, 13] | 8 [2, 13] |
| Seaquest | 2660 [2055, 3579] | 134 [20, 248] | 2344 [1541, 3450] | 1676 [368, 2336] |
| Skiing | -30000 [-30000, -30000] | -30000 [-30000, -30000] | -30000 [-30000, -30000] | -30000 [-30000, -30000] |
| Solaris | 1262 [863, 1686] | 878 [8, 1748] | 1280 [498, 2062] | 870 [370, 1736] |
| SpaceInvaders | 478 [429, 524] | 207 [198, 216] | 536 [405, 667] | 253 [235, 278] |
| StarGunner | 1146 [996, 1437] | 930 [710, 1150] | 1014 [994, 1048] | 976 [960, 1000] |
| Surround | -6 [-7, -5] | -9 [-9, -9] | -5 [-7, -5] | -8 [-9, -8] |
| Tennis | 0 [-1, 0] | -12 [-24, -2] | 0 [-1, 0] | -2 [-5, 0] |
| TimePilot | 3101 [2772, 3482] | 2195 [2040, 2350] | 3822 [3282, 4418] | 2376 [2070, 2880] |
| Tutankham | 130 [124, 139] | 14 [0, 30] | 138 [127, 151] | 37 [0, 112] |
| UpNDown | 26477 [11956, 43260] | 1692 [1526, 1859] | 34574 [9812, 71080] | 4568 [4174, 4972] |
| Venture | 0 [0, 0] | 0 [0, 0] | 74 [0, 210] | 0 [0, 0] |
| VideoPinball | 18826 [15048, 23233] | 6961 [6299, 7623] | 14689 [10497, 17911] | 11244 [9717, 12780] |
| WizardOfWor | 1918 [1706, 2154] | 575 [550, 600] | 1852 [1650, 2062] | 1190 [1000, 1430] |
| YarsRevenge | 27299 [23434, 30493] | 148 [0, 297] | 29495 [25242, 32299] | 14267 [10884, 16770] |
| Zaxxon | 3820 [2577, 4854] | 0 [0, 0] | 3144 [1128, 5118] | 0 [0, 0] |
| Mean | - | -1.69 [-1.70, -1.67] | -0.07 [-0.32, 0.18] | -0.79 [-0.86, -0.73] |
| Median | - | -0.63 [-0.64, -0.63] | -0.01 [-0.02, 0.00] | -0.24 [-0.24, -0.17] |
| IQM | - | -0.67 [-0.69, -0.65] | -0.02 [-0.05, 0.00] | -0.23 [-0.24, -0.20] |

| Task | MR.Q | No reward scaling | No min | No LAP |
|---|---|---|---|---|
| Alien | 2471 [1848, 3155] | 2074 [1402, 2821] | 2375 [1921, 3052] | 2265 [1784, 2768] |
| Amidar | 443 [376, 499] | 402 [345, 454] | 567 [465, 686] | 361 [291, 428] |
| Assault | 1125 [1094, 1160] | 1235 [1140, 1330] | 1154 [967, 1254] | 1095 [1027, 1170] |
| Asterix | 2216 [2081, 2346] | 2476 [2051, 2901] | 2881 [2600, 3045] | 2503 [2250, 2800] |
| Asteroids | 602 [493, 689] | 614 [520, 711] | 769 [713, 878] | 509 [447, 568] |
| Atlantis | 445022 [282338, 630730] | 608658 [262742, 946168] | 386213 [77480, 941750] | 329941 [196522, 506576] |
| BankHeist | 542 [348, 749] | 490 [176, 806] | 304 [205, 436] | 235 [136, 388] |
| BattleZone | 16520 [10560, 22760] | 13660 [8040, 18780] | 21300 [20100, 22200] | 13650 [9270, 17290] |
| BeamRider | 2007 [1855, 2194] | 1989 [1947, 2031] | 1989 [1948, 2016] | 1543 [1194, 1840] |
| Berzerk | 430 [383, 472] | 427 [361, 524] | 601 [398, 708] | 531 [464, 593] |
| Bowling | 50 [37, 65] | 78 [66, 88] | 38 [28, 50] | 68 [58, 78] |
| Boxing | 95 [94, 97] | 91 [87, 96] | 95 [92, 97] | 95 [94, 97] |
| Breakout | 25 [21, 32] | 22 [20, 26] | 27 [24, 30] | 21 [18, 25] |
| Centipede | 14954 [13541, 16508] | 15952 [12806, 19014] | 11288 [8365, 16843] | 12236 [9602, 14654] |
| ChopperCommand | 4348 [3756, 5002] | 2796 [2228, 3378] | 3283 [2330, 4770] | 3490 [2724, 4341] |
| CrazyClimber | 104766 [99290, 109629] | 105014 [94550, 112412] | 110046 [107390, 113610] | 88805 [77424, 100499] |
| Defender | 25962 [23406, 29182] | 30912 [27871, 33695] | 32336 [27280, 38695] | 32385 [29809, 34752] |
| DemonAttack | 4660 [4072, 5241] | 4893 [4282, 5345] | 2468 [712, 3513] | 4280 [2797, 6098] |
| DoubleDunk | -9 [-11, -9] | -11 [-14, -8] | -9 [-13, -7] | -11 [-14, -9] |
| Enduro | 1480 [1378, 1592] | 1450 [1311, 1593] | 967 [0, 1461] | 1117 [883, 1287] |
| FishingDerby | -34 [-43, -27] | -22 [-25, -20] | -17 [-28, -13] | -35 [-46, -27] |
| Freeway | 31 [31, 32] | 25 [13, 32] | 32 [31, 34] | 32 [32, 32] |
| Frostbite | 4003 [2871, 5163] | 3247 [1532, 4771] | 2401 [267, 4457] | 1595 [644, 2511] |
| Gopher | 4936 [3923, 5730] | 5802 [1467, 13322] | 11774 [7552, 18634] | 11483 [4836, 23229] |
| Gravitar | 275 [232, 322] | 256 [215, 305] | 393 [185, 570] | 329 [286, 382] |
| Hero | 8391 [6845, 10060] | 8775 [7575, 11125] | 7594 [7525, 7670] | 7519 [7379, 7640] |
| IceHockey | -2 [-3, -1] | -3 [-5, -2] | -2 [-3, -1] | -5 [-7, -4] |
| Jamesbond | 551 [534, 573] | 543 [460, 610] | 546 [470, 650] | 471 [406, 528] |
| Kangaroo | 4833 [2716, 7064] | 6148 [3412, 8600] | 8033 [5540, 9390] | 4616 [2392, 6998] |
| Krull | 8660 [8198, 9147] | 8878 [7898, 9491] | 8430 [6464, 9987] | 8535 [8189, 8924] |
| KungFuMaster | 26150 [21973, 30490] | 24292 [18954, 29568] | 25553 [23960, 27040] | 23422 [21084, 25671] |
| MontezumaRevenge | 12 [0, 34] | 0 [0, 0] | 190 [100, 370] | 20 [0, 50] |
| MsPacman | 4395 [3799, 5002] | 4086 [3847, 4413] | 3860 [3394, 4470] | 3602 [3051, 4116] |
| NameThisGame | 7511 [7085, 7911] | 8323 [7308, 9338] | 7992 [7194, 8458] | 7681 [6795, 8351] |
| Phoenix | 4843 [4635, 5033] | 4940 [4541, 5255] | 4780 [4605, 5008] | 4717 [4388, 5016] |
| Pitfall | -8 [-19, -1] | 0 [0, 0] | 0 [0, 0] | -1 [-3, 0] |
| Pong | 15 [13, 17] | 15 [14, 18] | 18 [19, 19] | 13 [10, 17] |
| PrivateEye | 100 [100, 100] | 40 [0, 80] | 125 [100, 177] | 50 [20, 80] |
| Qbert | 3600 [2554, 4366] | 2848 [1410, 4287] | 6567 [4638, 7722] | 3365 [2590, 4045] |
| Riverraid | 7362 [7062, 7630] | 6669 [5104, 7779] | 7464 [7213, 7688] | 6191 [5522, 6889] |
| RoadRunner | 27152 [19731, 34480] | 37306 [32906, 40582] | 38703 [35530, 45050] | 33950 [27807, 39559] |
| Robotank | 10 [9, 13] | 15 [13, 17] | 15 [10, 19] | 10 [7, 13] |
| Seaquest | 2660 [2055, 3579] | 1998 [1141, 2626] | 2005 [1596, 2348] | 1421 [986, 1782] |
| Skiing | -30000 [-30000, -30000] | -30000 [-30000, -30000] | -30000 [-30000, -30000] | -30000 [-30000, -30000] |
| Solaris | 1262 [863, 1686] | 1175 [660, 1770] | 772 [216, 1606] | 886 [637, 1106] |
| SpaceInvaders | 478 [429, 524] | 458 [431, 493] | 515 [488, 532] | 494 [459, 530] |
| StarGunner | 1146 [996, 1437] | 1074 [964, 1266] | 11993 [1240, 22010] | 991 [976, 1005] |
| Surround | -6 [-7, -5] | -5 [-7, -5] | -5 [-6, -4] | -7 [-9, -5] |
| Tennis | 0 [-1, 0] | 0 [0, 0] | 6 [-4, 20] | 0 [0, 0] |
| TimePilot | 3101 [2772, 3482] | 3118 [2620, 3618] | 4433 [4190, 4830] | 3616 [2880, 4259] |
| Tutankham | 130 [124, 139] | 144 [87, 184] | 156 [154, 160] | 125 [99, 150] |
| UpNDown | 26477 [11956, 43260] | 13859 [6662, 21316] | 37177 [17986, 54020] | 17527 [8653, 28247] |
| Venture | 0 [0, 0] | 0 [0, 0] | 0 [0, 0] | 0 [0, 0] |
| VideoPinball | 18826 [15048, 23233] | 18345 [15176, 21920] | 22721 [18131, 28516] | 20490 [14091, 27791] |
| WizardOfWor | 1918 [1706, 2154] | 1906 [1070, 2984] | 1790 [1560, 2010] | 1549 [1375, 1708] |
| YarsRevenge | 27299 [23434, 30493] | 27358 [21924, 31727] | 26211 [21118, 33567] | 24249 [20860, 27018] |
| Zaxxon | 3820 [2577, 4854] | 1336 [0, 3300] | 6746 [4940, 7760] | 1488 [369, 2733] |
| Mean | - | 0.18 [-0.25, 0.56] | 0.13 [-0.10, 0.58] | -0.13 [-0.38, 0.14] |
| Median | - | -0.00 [-0.01, 0.00] | 0.01 [0.00, 0.04] | -0.03 [-0.07, -0.00] |
| IQM | - | -0.01 [-0.02, 0.03] | 0.03 [0.03, 0.06] | -0.04 [-0.08, -0.01] |

| Task | MR.Q | No MR | 1-step return | No unroll |
|---|---|---|---|---|
| Alien | 2471 [1848, 3155] | 2886 [2458, 3315] | 1342 [1226, 1454] | 3056 [2300, 3614] |
| Amidar | 443 [376, 499] | 312 [204, 421] | 240 [213, 287] | 385 [335, 471] |
| Assault | 1125 [1094, 1160] | 1105 [1075, 1142] | 1045 [904, 1134] | 1079 [1003, 1140] |
| Asterix | 2216 [2081, 2346] | 2298 [1920, 2731] | 2708 [2110, 3710] | 2388 [2190, 2740] |
| Asteroids | 602 [493, 689] | 485 [387, 584] | 723 [689, 783] | 581 [406, 766] |
| Atlantis | 445022 [282338, 630730] | 12834 [9038, 17186] | 60986 [25200, 87460] | 57426 [40850, 88730] |
| BankHeist | 542 [348, 749] | 404 [73, 726] | 123 [77, 160] | 783 [231, 1080] |
| BattleZone | 16520 [10560, 22760] | 22000 [12100, 29080] | 4700 [3600, 5800] | 23733 [19600, 26100] |
| BeamRider | 2007 [1855, 2194] | 1849 [1726, 2004] | 1438 [1213, 1604] | 1389 [1274, 1535] |
| Berzerk | 430 [383, 472] | 437 [333, 537] | 361 [290, 400] | 443 [399, 489] |
| Bowling | 50 [37, 65] | 84 [73, 95] | 44 [30, 67] | 72 [60, 88] |
| Boxing | 95 [94, 97] | 92 [89, 95] | 93 [92, 95] | 95 [94, 96] |
| Breakout | 25 [21, 32] | 15 [14, 18] | 14 [7, 23] | 27 [17, 44] |
| Centipede | 14954 [13541, 16508] | 10517 [9157, 11949] | 8927 [7098, 10732] | 11984 [11005, 13691] |
| ChopperCommand | 4348 [3756, 5002] | 3394 [3140, 3764] | 2520 [1980, 2900] | 2866 [1670, 3990] |
| CrazyClimber | 104766 [99290, 109629] | 83734 [71466, 93070] | 66980 [65140, 70490] | 104076 [82650, 114820] |
| Defender | 25962 [23406, 29182] | 14469 [9247, 18763] | 9658 [3245, 17870] | 34718 [29120, 44180] |
| DemonAttack | 4660 [4072, 5241] | 746 [527, 994] | 1861 [1675, 1990] | 2840 [2064, 3618] |
| DoubleDunk | -9 [-11, -9] | -14 [-18, -11] | -7 [-10, -6] | -11 [-14, -10] |
| Enduro | 1480 [1378, 1592] | 897 [784, 1075] | 1064 [1055, 1069] | 1075 [1018, 1131] |
| FishingDerby | -34 [-43, -27] | -21 [-30, -8] | -82 [-95, -64] | -54 [-63, -45] |
| Freeway | 31 [31, 32] | 26 [13, 33] | 32 [32, 33] | 32 [32, 33] |
| Frostbite | 4003 [2871, 5163] | 3098 [1645, 4358] | 1231 [254, 3182] | 4182 [3473, 5451] |
| Gopher | 4936 [3923, 5730] | 1833 [1552, 2114] | 5597 [4242, 6512] | 4746 [2378, 6822] |
| Gravitar | 275 [232, 322] | 89 [0, 267] | 166 [70, 310] | 270 [145, 415] |
| Hero | 8391 [6845, 10060] | 7584 [7548, 7644] | 7594 [7544, 7676] | 6879 [5384, 7694] |
| IceHockey | -2 [-3, -1] | -5 [-8, -4] | -6 [-8, -5] | -2 [-4, -1] |
| Jamesbond | 551 [534, 573] | 433 [406, 462] | 518 [495, 545] | 546 [460, 625] |
| Kangaroo | 4833 [2716, 7064] | 7508 [5608, 9500] | 6940 [1520, 10460] | 9166 [8520, 9500] |
| Krull | 8660 [8198, 9147] | 8403 [7433, 9087] | 7386 [6611, 7881] | 7785 [7176, 8271] |
| KungFuMaster | 26150 [21973, 30490] | 19066 [15450, 22028] | 15300 [14760, 16350] | 29020 [27150, 30700] |
| MontezumaRevenge | 12 [0, 34] | 0 [0, 0] | 0 [0, 0] | 0 [0, 0] |
| MsPacman | 4395 [3799, 5002] | 3297 [2679, 3908] | 2282 [1956, 2592] | 2839 [2612, 2954] |
| NameThisGame | 7511 [7085, 7911] | 3638 [3207, 3999] | 5590 [5026, 5941] | 6529 [6069, 6983] |
| Phoenix | 4843 [4635, 5033] | 4101 [3752, 4503] | 3825 [3435, 4306] | 4941 [4613, 5193] |
| Pitfall | -8 [-19, -1] | -27 [-67, -5] | 0 [0, 0] | 0 [0, 0] |
| Pong | 15 [13, 17] | 12 [8, 16] | 14 [9, 20] | 12 [8, 16] |
| PrivateEye | 100 [100, 100] | 3068 [24, 9080] | 100 [100, 100] | 100 [100, 100] |
| Qbert | 3600 [2554, 4366] | 4491 [3362, 5870] | 2517 [2032, 3105] | 4220 [3918, 4508] |
| Riverraid | 7362 [7062, 7630] | 7479 [6818, 8239] | 6733 [5928, 7759] | 5856 [3607, 7855] |
| RoadRunner | 27152 [19731, 34480] | 29182 [23250, 35114] | 36145 [33590, 38700] | 37636 [33610, 40160] |
| Robotank | 10 [9, 13] | 11 [6, 16] | 6 [6, 7] | 12 [11, 17] |
| Seaquest | 2660 [2055, 3579] | 1556 [1248, 1866] | 2166 [2154, 2178] | 2194 [2112, 2284] |
| Skiing | -30000 [-30000, -30000] | -30000 [-30000, -30000] | -30000 [-30000, -30000] | -30000 [-30000, -30000] |
| Solaris | 1262 [863, 1686] | 552 [319, 881] | 1107 [662, 1552] | 1088 [638, 1386] |
| SpaceInvaders | 478 [429, 524] | 550 [476, 632] | 389 [382, 396] | 551 [502, 596] |
| StarGunner | 1146 [996, 1437] | 1272 [1020, 1730] | 1290 [1000, 1580] | 1423 [990, 2220] |
| Surround | -6 [-7, -5] | -8 [-9, -7] | -5 [-6, -5] | -5 [-10, 1] |
| Tennis | 0 [-1, 0] | 0 [-2, 0] | 0 [-1, 0] | 0 [-2, 0] |
| TimePilot | 3101 [2772, 3482] | 2440 [1540, 3178] | 2535 [2030, 3040] | 2236 [1170, 3810] |
| Tutankham | 130 [124, 139] | 112 [65, 148] | 151 [120, 182] | 148 [127, 181] |
| UpNDown | 26477 [11956, 43260] | 25451 [17297, 34036] | 4477 [2993, 5962] | 31342 [3170, 84771] |
| Venture | 0 [0, 0] | 0 [0, 0] | 0 [0, 0] | 0 [0, 0] |
| VideoPinball | 18826 [15048, 23233] | 8524 [3899, 12519] | 13506 [7955, 19058] | 27525 [22242, 35815] |
| WizardOfWor | 1918 [1706, 2154] | 2058 [1632, 2640] | 1545 [1490, 1600] | 1573 [1430, 1700] |
| YarsRevenge | 27299 [23434, 30493] | 30666 [28659, 33225] | 18513 [16940, 20088] | 19082 [11442, 24565] |
| Zaxxon | 3820 [2577, 4854] | 5758 [4712, 6962] | 330 [0, 660] | 0 [0, 0] |
| Mean | - | -0.78 [-0.88, -0.69] | -0.70 [-0.81, -0.59] | -0.33 [-0.41, -0.28] |
| Median | - | -0.06 [-0.10, -0.01] | -0.12 [-0.12, -0.07] | -0.00 [-0.02, 0.00] |
| IQM | - | -0.09 [-0.14, -0.05] | -0.15 [-0.16, -0.13] | -0.01 [-0.04, -0.00] |

