# OpenReview forum: "Towards General-Purpose Model-Free Reinforcement Learning"
_ICLR.cc/2025/Conference — ICLR 2025 Spotlight_

### Official Review · Reviewer_fYrs · 2024-10-16

**Soundness:** 3
**Presentation:** 4
**Contribution:** 3
**Rating:** 8
**Confidence:** 3

**Summary:**

This paper introduces MR.Q, a model-free RL algorithm that uses model-based representations to generalize across a variety of domains with the same set of hyper-parameters. MR.Q leverages the fact that the fixed-point between linear reconstructions of dynamics and value are the same. MR.Q relaxes the theory to show that non-linear value functions using this learned representation improve performance. The algorithm trains a policy by using a deterministic policy gradient similar to DDPG. To prevent trivial solutions, the dynamics reconstruction loss uses the next state as a target, as opposed to the next state-action encoding. A number of other tricks are used, including a two-hot representation of reward, target policy noise from TD3, pre-activation regularization, Huber loss, and scaling by the average reward. These components are tested in a series of ablations, and evaluations are conducted over Gym, DMC, and Atari, comparing to DreamerV3, TD-MPC2, TD7, Rainbow, and DQN.

**Strengths:**

This paper is well-written and address the import question of robustness of RL algorithms across various domains. The paper is theoretically motivated, and performs many ablations. The proposed algorithm is clear and if the evaluations hold, could prove to be of great interest to the community.

**Weaknesses:**

The primary weaknesses are evaluation and novelty. A few points are listed here, and some clarifying questions in the questions section:
1) While it is not possible to evaluate every benchmark, the Minecraft benchmark from Dreamer, which I believe tests long-term credit assignment, is notably missing. It could be great to see how this method holds up when reward is sparse, exploration is difficult, and policy gradient variance may be high. Procgen is also missing, which may be one of the harder domains in the Dreamer paper.
2) The model-based representations are not particularly novel. For example, consider VariBAD (Zintgraf et al., 2020), which reconstructs reward and dynamics functions.
3) One piece of novelty is that the reconstruction of reward and transitions use a linear function of the same latent encoding. However, experiments show that a non-linear value reconstruction is useful, and it may be the same for the model-based objectives as well. Are there experiments testing this?
4) It would be great to see comparisons to other baselines here that use contrastive learning as well, e.g., Neural Predictive Belief Representations (Guo et al., 2018).
5) I am also concerned that the method cannot handle general POMDPs. Is it true that only the state and action are passed the policy and not history? If you are only using a latent that reconstructs the reward and dynamics, this statistic must condition on history, and is not general enough for any POMDP, but only a distribution of MDPs, as in meta-RL (Zintgraf et al., 2020).
6) It would be great to see a comparison to your method, but with additional synthetic trajectories simulated by your learned model. Without this experiment, it is hard to know if the compromise between model-based RL and model-free RL is the right trade-off.

Minor - Clarity:
A) Could you further clarify and motivate the two-hot classification and the symexp so the reader does not have to look externally?
B) Could you give more motivation and explanation of the two equalities in Equation (5)? Also, how is Wp defined for when given a specific time-step, t?

**Questions:**

Some questions:
1) Is there a reason that TD-MPC2 and TD7 were chosen over Truncated Quantile Critics (TQC) and Randomized Ensembled Double Q-Learning (REDQ)? Are there any papers showing your baselines to be superior?
2) Are there any experiments evaluating whether a non-linear reward and transition reconstruction might be superior?
3) Have you evaluated you algorithm with sequence models than can handle POMDPs?
4) Have you compared to other self-supervised model-based algorithms (esp. those using contrastive learning or those designed for POMDPs?)
5) You mention that the number of frames is not equal to the number of time-steps due to sticky actions. If you plot the number of frames on the x-axis, are the results still the same? Does your algorithm require more frames than others?
6) It looks like the curves for Atari are not converged. Could you run longer?

---

> ### Author Response · Authors · 2024-11-19
> **Changes: Added limitations section + related work + additional experiment + writing improvements**
>
> Thank you for the review! We respond to each of your comments below.
>
> > While it is not possible to evaluate every benchmark, the Minecraft benchmark from Dreamer, which I believe tests long-term credit assignment, is notably missing. It could be great to see how this method holds up when reward is sparse, exploration is difficult, and policy gradient variance may be high. Procgen is also missing, which may be one of the harder domains in the Dreamer paper.
>
> We agree that analyzing every benchmark provides real value in understanding the performance of an algorithm and its strengths and weaknesses. Since we can only evaluate on some subset, we chose the most common benchmarks in the RL literature as a starting point. Admittedly, as mentioned in Section 6 (discussion and conclusion), we would not expect MR.Q to perform well on sparse reward environments such as the Minecraft benchmark, as it does not consider any explicit exploration mechanism, nor does it contain any history-based component.
>
> We have added some discussion on the limitations of MR.Q to the conclusion to expand on this (Section 6 - Limitations). Despite this, we hope that our 118 environment evaluation setup can convince you of the strength of MR.Q across a varied set of challenging domains.
>
> > The model-based representations are not particularly novel. For example, consider VariBAD (Zintgraf et al., 2020), which reconstructs reward and dynamics functions.
>
> We agree that model-based representations are a well considered area of research, our related work attempts to cover as many closely related papers as possible but it is not surprising that some slipped past us. We have included VariBAD as a reference in the related work.
>
> To comment on novelty: we believe that model-based representations are an important research area in RL, and therefore this direction is worth contributing to. Even if our approach is not significantly different from existing approaches, we believe that our key contribution to this area is demonstrating their effectiveness as a general tool across many benchmarks.
>
> > One piece of novelty is that the reconstruction of reward and transitions use a linear function of the same latent encoding. However, experiments show that a non-linear value reconstruction is useful, and it may be the same for the model-based objectives as well. Are there experiments testing this?
>
> We agree that this is an interesting design choice. We have tried similar experiments during development. An interesting choice here is balancing model size, for example, additional layers to improve reward or dynamics estimation could also be spent on improving capacity elsewhere in the model. We have added an design experiment that replaces the linear MDP prediction layer with 3 MLPs (Non-linear model in Section 5.2 - Design study). We find that this variant overall performs slightly worse than the linear version, suggesting that the benefit from being approximately linear outweighs the benefit of additional capacity.
>
> > It would be great to see comparisons to other baselines here that use contrastive learning as well, e.g., Neural Predictive Belief Representations (Guo et al., 2018).
>
> While more baselines will always provide value, considering our number of environments, we had to be selective about which baselines to use. Since our emphasis was on raw performance, we tried to include only highly competitive baselines (SOTA or near SOTA). We believe there are many new settings to tackle in future work, and contrasting against other representation learning methods may provide interesting insight or unlock new approaches in these settings, but we have to leave something for the future!
>
> > I am also concerned that the method cannot handle general POMDPs. Is it true that only the state and action are passed the policy and not history? If you are only using a latent that reconstructs the reward and dynamics, this statistic must condition on history, and is not general enough for any POMDP, but only a distribution of MDPs, as in meta-RL (Zintgraf et al., 2020).
>
> The method is indeed intended for the MDP setting rather than POMDPs. All of the benchmarks we evaluate with only consider the MDP setting (or are approximated with short histories). The representation learning scheme can be extended to consider the history by using recurrent or transformer architectures, but again we leave this important addition to future work.

---

> > ### Author Response · Authors · 2024-11-19
> > **Part 2**
> >
> > > It would be great to see a comparison to your method, but with additional synthetic trajectories simulated by your learned model. Without this experiment, it is hard to know if the compromise between model-based RL and model-free RL is the right trade-off.
> >
> > This is an interesting idea! One issue we foresaw with using the model for actual model-based RL is the lack of established (or community accepted) ways to do model-based RL. For example, adding synthetic trajectories opens up a huge number of design choices: what percentage of data should be synthetic? How many timesteps should synthetic trajectories be? How should we handle terminal transitions? Should we plan over synthetic trajectories and if so, how? The problem this creates is that there’s no guarantee that we’d be doing it “right”. This is the main reason we have omitted these experiments.
> >
> > One of the advantages of our approach is that these design choices can be skipped. That’s not to say that a maximally performing algorithm wouldn’t consider or include these components, but our results do show that these model-based components are not necessary on common RL benchmarks to achieve SOTA or near SOTA performance.
> >
> > So maybe a nice perspective is that we’re not trying to decide the winner between model-free and model-based methods, but instead, just trying to show the model-free part of the story. Hopefully you find that as interesting as we do.
> >
> > > A) Could you further clarify and motivate the two-hot classification and the symexp so the reader does not have to look externally?
> >
> > Of course. We have adapted the writing in this section to improve clarity and simplify some of the necessary details. Let us know if you feel that it can be improved further.
> >
> > > B) Could you give more motivation and explanation of the two equalities in Equation (5)?
> >
> > The second equality follows from the geometric series of matrices. But on second look, this relationship is only necessary for the proof of Theorem 1—so we have removed it from the main body to minimize confusion. We have also modified equation 6 to more clearly state these are definitions of wr and Wp rather than equalities.
> >
> > > Also, how is Wp defined for when given a specific time-step, t?
> >
> > We are assuming MDP structure so the dynamics (and Wp) are unaffected by the time step.
> >
> > > Is there a reason that TD-MPC2 and TD7 were chosen over Truncated Quantile Critics (TQC) and Randomized Ensembled Double Q-Learning (REDQ)? Are there any papers showing your baselines to be superior?
> >
> > TD-MPC2 was chosen because it is a general-purpose continuous control algorithm, with strong results across a wide range of benchmarks including state-of-the-art (or near) results in DMC. TD7 was chosen because it is state-of-the-art (or near) for the Gym benchmark, and is conceptually similar to MR.Q. The TD7 paper establishes a clear performance benefit over TQC in Gym. We are not aware of any work that shows REDQ is state-of-the-art beyond the early learning stage but regardless, we believe our choices of baselines to be competitive with state-of-the-art results in these benchmarks.
> >
> > > You mention that the number of frames is not equal to the number of time-steps due to sticky actions. If you plot the number of frames on the x-axis, are the results still the same? Does your algorithm require more frames than others?
> >
> > The difference between the number of frames and timesteps is not due to sticky actions, but instead due to the action repeat. As such, the number of frames is 4 times the number of time steps for all algorithms. This is standard practice for these benchmarks and holds across many existing prior work (including the original DQN paper for Atari), so this is not a quirk of our algorithm or environment setup and also holds true for all of our baselines.
> >
> > Hope this helps address some of your primary concerns!

---

> > > ### Comment · Reviewer_fYrs · 2024-11-20
> > >
> > > Thank you for addressing these concerns. I have increased my score accordingly.

---

### Official Review · Reviewer_dWbu · 2024-10-26

**Soundness:** 3
**Presentation:** 3
**Contribution:** 3
**Rating:** 8
**Confidence:** 4

**Summary:**

The paper proposes to leverage model-based representations for model-free reinforcement learning. Model-based approaches are only used for learning the representation, but no planning or simulation is employed. The authors integrate model-based representations in the proposed method MR.Q. They evaluate MR.Q on 3 benchmarks (Gym, DMC, Atari) and find that it leads to faster learning speed and improved performance compared to competitors, albeit not consistently across domains.

**Strengths:**

The authors propose an algorithm named, MR.Q that performs well across domains. They conduct well-selected ablations on MR.Q to highlight what effect their design decisions have on the algorithm. The empirical results are broad and seem sound. Generally, a useful contribution to the field.

**Weaknesses:**

The authors argue that the benefits of model-based approaches may stem from their learned representations, rather than from their planning abilities. While this may be true to some degree, we assume that this is highly specific to the domains they conduct experiments on. In particular, Gym and DMC environments and most Atari games do not require much planning to be solved. However, we know that in other domains planning plays a critical role. Therefore, we believe it is important for the authors to clarify that the benefits of model-based representations alone depend on the domain at hand, to avoid any misconceptions.

The authors refer to their method as “generalist”, which is a term commonly used for multi-task agents in the literature. However, all conducted experiments are in single task settings. Instead, “generalist” in the context of this paper refers to being robust to different environments (reward scales, dynamics, with the same hyperparameters). We therefore strongly encourage the authors to rethink this terminology to avoid confusions.

The results for Dreamerv3 are surprisingly bad on continuous control tasks. This suggests that this baseline is purely tuned for these environments.

The proposed approach comes with a large list of hyperparameters, which may limits its practical applicability.

**Questions:**

- Could you clarify your use of “generalist”?
- There seems to be a clear trend of Dreamerv3 underperforming on continuous control environments. Any intuitions why?
- How do the Training FPS of MRQ. in Figure 1 compare to other non-planning baselines on DMC-Visual (TD7) and Atari (Rainbow, DQN)?

---

> ### Author Response · Authors · 2024-11-19
> **Changes: Remove generalist terminology + added limitations section**
>
> Thank you for the review! We respond to each of your comments below.
>
> > The authors argue that the benefits of model-based approaches may stem from their learned representations, rather than from their planning abilities. While this may be true to some degree, we assume that this is highly specific to the domains they conduct experiments on. In particular, Gym and DMC environments and most Atari games do not require much planning to be solved. However, we know that in other domains planning plays a critical role. Therefore, we believe it is important for the authors to clarify that the benefits of model-based representations alone depend on the domain at hand, to avoid any misconceptions.
>
> We certainly agree. We tried to be careful with our wording (e.g., “suggesting”) in our comments in Section 6, but you are correct that this is highly dependent on the set of benchmarks. It is also worth noting that our results don’t necessarily show any certain performance downside to planning, rather we aim to highlight the effectiveness of the representation itself. We will adjust the writing to reflect the importance of the environment when considering the benefits of different methods (Section 6 - Model-based and model-free RL).
>
> > The authors refer to their method as “generalist”, which is a term commonly used for multi-task agents in the literature. However, all conducted experiments are in single task settings. Instead, “generalist” in the context of this paper refers to being robust to different environments (reward scales, dynamics, with the same hyperparameters). We therefore strongly encourage the authors to rethink this terminology to avoid confusions.
>
> Thank you for pointing out this oversight. We have adjusted “generalist” to “general” (as used by DreamerV3) or “general-purpose” algorithm. Let us know if you think there is a better term which captures this sentiment.
>
> > The results for Dreamerv3 are surprisingly bad on continuous control tasks. This suggests that this baseline is purely tuned for these environments.
>
> We were also surprised by the poor performance of DreamerV3 at these tasks. We use the suggested default hyperparameters for DreamerV3, and compare against DreamerV3 on environments contained in their original evaluation. Our results are consistent with the TD-MPC2 paper, which also shows the weakness of DreamerV3 in continuous control tasks. Ultimately, we believe our evaluation is fair and highlights the overwhelming challenge of developing algorithms which are effective across a diverse range of benchmarks. As far as possible reasons, we suspect DreamerV3 was over-tuned for the Atari benchmark (compared to continuous domains), again a consequence of the difficulty of maintaining a high performance across varied benchmarks.
>
> > The proposed approach comes with a large list of hyperparameters, which may limits its practical applicability.
>
> We aim to take a transparent approach to reporting all of our hyperparameters, to provide high clarity and maximize reproducibility. Many hyperparameters do not require tuning or are based on default values defined in prior work. Regardless, our results demonstrate that it is possible for a single set of hyperparameters to work across a wide range of environments, so the number of hyperparameters may not be such an explicit downside of the method.
>
> > How do the Training FPS of MRQ. in Figure 1 compare to other non-planning baselines on DMC-Visual (TD7) and Atari (Rainbow, DQN)?
>
> MR.Q is more expensive than TD7 and Rainbow, due to the larger model size (for TD7) and the additional training of the model (for Rainbow). These costs can often depend on the implementation details, but we estimate MR.Q runs at about half the speed.

---

> > ### Comment · Reviewer_dWbu · 2024-11-25
> >
> > Thank you for the response and the added changes. I am happy to keep my original score.

---

### Official Review · Reviewer_b3JB · 2024-11-01

**Soundness:** 3
**Presentation:** 4
**Contribution:** 3
**Rating:** 8
**Confidence:** 4

**Summary:**

This paper presents a model-free deep RL algorithm, designed to achieve consistently high performance across RL settings. The method (MR.Q) is based on the widely accepted principle that self-supervision (world models) can provide an effective state encoding for policy and value learning. Additional components are added to ensure hyperparameter robustness and the method is evaluated with a single set of hyperparameters in each setting. A theoretical analysis is provided, suggesting an approximately linear relationship between model-based and value-based representations. MR.Q is evaluated on MuJoCo, DMC, and Atari benchmarks, with comparisons to a handful of algorithms and an extensive number of ablations.

As detailed below, I was unable to find some key details regarding the author's experiments, so my current score is provisional. I am highly open to raising or lowering my score following discussion with the authors.

**Strengths:**

* The paper is very well presented and has a clear structure.
* The core approach - using world model representations to support policy learning - is sound and well-motivated in prior literature.
* Strong efforts are made to make the model hyperparameter-insensitive, which is particularly valuable in RL where hyperparameter tuning is often ignored or exploited to give the appearance of improved performance.
* The evaluation is extensive in the number of benchmarks and ablations. A consistent set of hyperparameters is used by the method, strengthening the claim of robustness.

**Weaknesses:**

## Hyperparameters
One of the primary aims of the paper is hyperparameter robustness, with multiple components being designed to improve robustness and a single set of hyperparameters being used to evaluate MR.Q. However, aiming for robustness does not allow hyperparameters to be ignored in evaluation. In fact, *more* attention should be paid to hyperparameters, to demonstrate that MR.Q is less sensitive hyperparameters than baseline methods, or performs better when all algorithms are constrained to a single hyperparameter setting.

Unfortunately, the paper does the opposite and provides no detail (that I could find) regarding the hyperparameter settings of the baseline algorithms, nor how or if they were tuned. This is a critical weakness of the paper, as the possibility of untuned baselines undermines the claimed performance improvements.

One example of this is that DreamerV3 outperforms MR.Q on only 1/4 benchmarks (Atari). Notably, Atari is also the only benchmark for which the results come from the reference work. For the remaining 3 benchmarks, the authors run DreamerV3 themselves. This does not imply that the authors failed to tune DreamerV3 or that the comparison was unfair, however, given the lack of detail regarding their tuning procedure or the hyperparameter sensitivity of the methods, the result is seriously undermined.

Another example is that the paper seems to maintain model sizes from the original implementations of each algorithm. However, the size of the model is not defined by the algorithm, so it would be much more convincing to normalize the parameter count between algorithms and show that MR.Q achieves superior performance at the *same* size, or has a better scaling curve.

## Baselines
Some key algorithms are missing, most notably PPO. It is not expected that the authors compare to every popular RL algorithm, but PPO is undoubtedly the closest thing to a "general-purpose" RL algorithm in current literature.* A number of recent methods have also claimed to serve as "general-purpose" RL algorithms (e.g. PQN, Gallici et al., 2025). These would make for valuable comparisons in future work but are not required for acceptance.

*As of 31st Oct, PPO has 5460 citations in 2024, compared to 269 (Dreamer v3), 32 (TD7), 39 (TD-MPC2), 1531 (TD3). Not a perfect proxy for the methods' generality, but certainly indicative.

## Ablation study
The ablation study fails to provide the statistical significance of the results and lacks analysis. Furthermore, the "reverting to theory" ablations are highly unsurprising so provide little contribution, and many of the remaining ablations show minimal performance gains. Some of these components are designed to handle edge cases (e.g. reward scaling handling environments without normalized reward), meaning the benefit is unclear from mean performance across a suite, but may be apparent by examining the performance distribution. For these in particular, further analysis of the results would strengthen the contribution of each component.

## Theory
The theory is based on highly constrained linear assumptions and provides little justification for the proposed method. The idea of using world model representations for policy learning is intuitive and can be explained much more clearly without sacrificing correctness. I understand there is a pressure on authors to provide "theoretical rigour", but in this case it adds little to the paper and would be better replaced with an extended analysis of the empirical results.

**Questions:**

Were the baselines tuned, and how was this done if so?

Do you have evidence that A) MR.Q is less hyperparameter sensitive than baselines, or B) the best single set of hyperparameters for MR.Q achieves higher performance than the equivalent for a baseline?

As far as I can tell, the proposed symexp loss will have higher resolution at higher reward values. This is counterintuitive, as one would expect precision to increase for values near 0. Can the authors explain this?

---

> ### Author Response · Authors · 2024-11-19
> **Changes: Clarify hyperparameters + added PPO as a baseline + statistical significance and environment breakdown**
>
> Thank you for the review! We respond to each of your comments below.
>
> > Unfortunately, the paper does the opposite and provides no detail (that I could find) regarding the hyperparameter settings of the baseline algorithms, nor how or if they were tuned. This is a critical weakness of the paper, as the possibility of untuned baselines undermines the claimed performance improvements.
>
> > Were the baselines tuned, and how was this done if so?
>
> > Do you have evidence that A) MR.Q is less hyperparameter sensitive than baselines, or B) the best single set of hyperparameters for MR.Q achieves higher performance than the equivalent for a baseline?
>
> Thank you for highlighting this oversight, as it is important to state clearly how the baseline hyperparameters were chosen. In our work, we stick to author-suggested defaults without tuning and keep them fixed across all benchmarks. We will add clarification on this point in the paper to avoid any ambiguity (Section 5 and Appendix B.4).
>
> This is consistent with the claims of TD-MPC2 and DreamerV3, which both use a fixed single set of hyperparameters across their respective benchmarks. Using new hyperparameters would contradict the claims made by the authors, and we believe that our setup is as fair as possible, given that it matches the authors’ original experimental setup, and we mainly test on environments included in their respective papers.
>
> To be specific:
> - For TD-MPC2, we use the hyperparameters used by the authors for DMC and DMC-visual. For Gym we use the episodic branch of the authors’ repo (as discussed in Appendix B.4), which is intended for the Gym environments.
> - For DreamerV3, we use the hyperparameters tuned by the authors’ for DMC, DMC-visual, and the author’s results for Atari. For Gym we use the same setup as DMC.
> - For all the other baselines, we use the specific hyperparameters designated for their respective benchmark. The sole exception is TD7, which we keep constant between Gym and DMC.
>
> Even if we omit the benchmarks that TD-MPC2 and DreamerV3 do not tune for (Gym), we believe that our results demonstrate that MR.Q’s best single set of hyperparameters is competitive or outperforms the best single set of hyperparameters of TD-MPC2 and DreamerV3 with lower computational overhead.
>
> > Another example is that the paper seems to maintain model sizes from the original implementations of each algorithm. However, the size of the model is not defined by the algorithm, so it would be much more convincing to normalize the parameter count between algorithms and show that MR.Q achieves superior performance at the same size, or has a better scaling curve.
>
> We believe that architecture falls under the single set of hyperparameters of the baseline models, and as such we do not modify them. To your point however, MR.Q already uses a smaller model size than both TD-MPC2 and DreamerV3 across all of our 118 environments. We believe that our results demonstrate MR.Q’s superior parameter efficiency than the baselines.
>
> > Some key algorithms are missing, most notably PPO. It is not expected that the authors compare to every popular RL algorithm, but PPO is undoubtedly the closest thing to a "general-purpose" RL algorithm in current literature.* A number of recent methods have also claimed to serve as "general-purpose" RL algorithms (e.g. PQN, Gallici et al., 2025). These would make for valuable comparisons in future work but are not required for acceptance.
>
> We agree that PPO is a highly general-purpose algorithm. We chose to exclude PPO from our experimental results because it has not been shown to be a competitive baseline in any of the benchmarks we considered. Our primary objective was to evaluate MR.Q against well-established near-SOTA baselines to demonstrate the effectiveness of MR.Q. The DreamerV3 paper demonstrates PPO underperforms DreamerV3 in DMC, DMC-visual, and Atari. TD7 is shown to outperform TD3 in Gym. TD3 in turn, was demonstrated to outperform PPO in the original TD3 paper.
>
> However, we acknowledge that demonstrating an explicit performance gap by showing PPO results would strengthen the paper. So we have included PPO in our results.
>
> PQN (Gallici et al., 2025) is an interesting comparison, but unfortunately, it appears to be a concurrent submission to this ICLR, which explains why we were not aware of it (but thank you for highlighting it regardless). PQN also does not extend naively to continuous actions, which make up three out of our four benchmarks. However, this does highlight the community’s interest in powerful general-purpose algorithms.

---

> > ### Author Response · Authors · 2024-11-19
> > **Part 2**
> >
> > > The ablation study fails to provide the statistical significance of the results and lacks analysis. Furthermore, the "reverting to theory" ablations are highly unsurprising so provide little contribution, and many of the remaining ablations show minimal performance gains. Some of these components are designed to handle edge cases (e.g. reward scaling handling environments without normalized reward), meaning the benefit is unclear from mean performance across a suite, but may be apparent by examining the performance distribution. For these in particular, further analysis of the results would strengthen the contribution of each component.
> >
> > The reverting to theory experiments aim to show the importance of practical approximations to our theoretically-guided loss and also highlights the overall effectiveness of our complete model-based representation.
> >
> > We were trying to reduce clutter in the paper by compressing the ablation analysis to a single table, but we agree with your comment about components handling edge cases. We have included tables showing the performance breakdown of each ablation on every environment (Appendix D) and included statistical significance in the original table (Section 5.2, Table 2). We have also changed from percent difference to change in normalized score (since percent difference is not stable for some Atari games where scores are 0).
> >
> > The theory is based on highly constrained linear assumptions and provides little justification for the proposed method. The idea of using world model representations for policy learning is intuitive and can be explained much more clearly without sacrificing correctness. I understand there is a pressure on authors to provide "theoretical rigour", but in this case it adds little to the paper and would be better replaced with an extended analysis of the empirical results.
> >
> > As highlighted in our related work, the use of model-based objectives is common practice for representation learning in RL, driven by intuitive ideas, with many different approaches and practical details. However, we believe that the value of our theoretical results is that it provides guidance in terms of how to approach this representation learning problem and what components are necessary.
> >
> > > As far as I can tell, the proposed symexp loss will have higher resolution at higher reward values. This is counterintuitive, as one would expect precision to increase for values near 0. Can the authors explain this?
> >
> > We have adjusted the writing of the loss function to try to avoid this misunderstanding (under Equation 15). Let us know if you feel that it could be improved further.
> >
> > To be clear, we use a cross-entropy loss with a two-hot encoding of the reward. The unique element of our approach is that the buckets are not uniformly spread out and instead use buckets spaced with symexp intervals.
> > To be explicit: there are 65 buckets from symexp(-10) to symexp(10).
> > - The middle bucket (33rd bucket) will have value exp(0) - 1 = 0.
> > - The 34rd bucket has the value exp(10/32) -1 = 0.367.
> > - The last bucket (65th) has a value of exp(10) - 1 = 22025.
> > - The 64th bucket has a value of exp(10 - 10/32) - 1 = 16114.
> > So, in line with your expectations, there is higher precision at values near 0.
> >
> > Hope this addresses your main concerns!

---

> > > ### Comment · Reviewer_b3JB · 2024-11-19
> > > **Rebuttal response**
> > >
> > > Thank you for taking the time to address my comments thoroughly!
> > >
> > > I appreciate the clarification on hyperparameters here and in the paper, I understand your approach. I still believe that model scale should be orthogonal to the choice of algorithm and hyperparameters, but since MR.Q outperforms these baselines with fewer parameters I am confident in its robustness.
> > >
> > > The new ablation breakdown on each task is also an interesting addition and significantly strengths work. In particular, the Atari results demonstrate the inconsistency of some components, particularly the loss function. This is a valuable result, so it would be good to see it referenced and discussed further in the main body.
> > >
> > > Overall, this paper is rigorous and a valuable contribution to the literature, so I've increased my score (5 -> 8). Nice work!

---

### Official Review · Reviewer_qhCX · 2024-11-04

**Soundness:** 3
**Presentation:** 2
**Contribution:** 3
**Rating:** 6
**Confidence:** 3

**Summary:**

The authors propose a generalist, model-free deep reinforcement learning algorithm named MR.Q, which integrates several advanced algorithmic techniques to achieve performance improvements and reduce training costs across multiple benchmark datasets.

**Strengths:**

1. MR.Q combines various state-of-the-art techniques and demonstrates promising performance gains with a single set of hyperparameters across multiple benchmarks, showing both efficacy and resource efficiency.
2. The paper provides detailed ablation studies, offering insights that may benefit future research in this field.

**Weaknesses:**

The theoretical motivation presented by the authors is somewhat confusing and challenging to interpret. Please refer to Question 3 for details.

**Questions:**

1. What exactly is the objective of the model-based elements mentioned by the authors? It seems that all techniques described are already present in existing model-free reinforcement learning frameworks.
2. The authors only conducted experiments on the Gym, DMC, and Atari benchmarks, which are classic but relatively homogeneous environments. Claiming generality might be overstated given these limited evaluation. For example, the widely accepted generalist algorithm PPO has demonstrated its effectiveness across a broad range of tasks. To establish the generality of this method, additional testing on tasks with some heterogeneity from these benchmarks, such as fine-tuning on large models, would strengthen the claim.
3. Equation (7) seems to be derivable directly from Equation (3). Could you clarify the phrase "from this insight"?
4. The authors describe the linear decomposition of the value function as the motivation, yet the final algorithm design and ablation studies prefer using common nonlinear function approximations. The authors claim that Equations (5-6) are inspired by model-based representations, but related ideas have already appeared in existing model-free algorithms. While this work is innovative and effective from an algorithmic design perspective, the theoretical sections seem somewhat retrofitted to fit the empirical results.

---

> ### Author Response · Authors · 2024-11-19
> **Changes: Limitations section + additional ablation + improvements to writing**
>
> Thank you for the review! We respond to each of your comments below.
>
> > What exactly is the objective of the model-based elements mentioned by the authors? It seems that all techniques described are already present in existing model-free reinforcement learning frameworks.
>
> We outline the purpose and benefits of a model-based representation in Section 4, line 171-177. To summarize, the model-based objective provides a richer learning signal than value-based learning, which may help learning stability or generalization and also helps abstract away uninformative information in the original observation space. The actual learning objective is described in Equations 10 and 14.
>
> While it is true that some of the techniques we describe are inspired by existing model-free algorithms, we demonstrate that the correct combination of these concepts leads to a powerful final algorithm. Let us know if we misunderstood your question, we are happy to clarify further.
>
> > The authors only conducted experiments on the Gym, DMC, and Atari benchmarks, which are classic but relatively homogeneous environments. Claiming generality might be overstated given these limited evaluation. For example, the widely accepted generalist algorithm PPO has demonstrated its effectiveness across a broad range of tasks. To establish the generality of this method, additional testing on tasks with some heterogeneity from these benchmarks, such as fine-tuning on large models, would strengthen the claim.
>
> We acknowledge that extending our results to some unique settings, such as fine-tuning large models, would offer some concrete additional evidence at the generality of our method. We have added some remarks about this to the limitations section in the conclusion (Section 6 - limitations).
>
> We have chosen to focus on common RL benchmarks as they allow us to leverage well established baselines and experimental protocol. We do not claim that MR.Q is a fully general-purpose algorithm–our goal is to make meaningful progress towards developing more versatile methods, and we believe that our current results represent an important step in this direction.
>
> > The theoretical motivation presented by the authors is somewhat confusing and challenging to interpret. Please refer to Question 3 for details.
>
> > Equation (7) seems to be derivable directly from Equation (3). Could you clarify the phrase "from this insight"?
>
> Apologies for the confusion. The insight is Theorem 1, Equation 7 is a definition. To be explicit: From this insight (Theorem 1), we connect the definition of value error (Equation 7) to the accuracy of the estimated reward and dynamics (shown in Theorem 2). We have re-written this sentence to eliminate the confusion.
>
> We hope that clarifies any confusion in the theoretical results, but if there are any unclear points, we would love to engage in further discussion to improve the clarity of the paper further.
>
> > The authors describe the linear decomposition of the value function as the motivation, yet the final algorithm design and ablation studies prefer using common nonlinear function approximations. The authors claim that Equations (5-6) are inspired by model-based representations, but related ideas have already appeared in existing model-free algorithms. While this work is innovative and effective from an algorithmic design perspective, the theoretical sections seem somewhat retrofitted to fit the empirical results.
>
> While we agree that our key results are empirical, our theoretical work provides important guidance for designing the algorithm. The existing literature offers many similar ideas, making it hard to choose the best approach. Our theory helps identify which design choices (and approximations) are important. As an additional design experiment, we have included results (in Section 5.2) where we replace the linear model with three non-linear MLPs. Although this increases capacity, we find it slightly harms performance. This helps justify that a linear model is a helpful design choice.

---

> > ### Comment · Reviewer_qhCX · 2024-11-24
> >
> > Thanks for the author’s response; I’ll stick with my original score.

---

### Author Response · Authors · 2024-11-19
**Global Response**

Thank you to all of the reviewers and AC for their time and expertise. We have responded to each of the comments of the reviewers and uploaded a revised version of the paper (adjustments in red).

Changes:
- **PPO**: (b3JB): We’ve included PPO as a baseline in all figures using Stable Baselines 3.
- **Ablations**: (fYrs, b3JB, qhCX): We’ve added a new ablation analyzing the impact of a non-linear model (fYrs, qhCX) and included statistical significance to the main ablation table and added per-environment breakdown of each ablation to Appendix D (b3JB).
- **Clarity and writing improvements**: (qhCX, b3JB, dWbu, fYrs)
    - Clarified writing to avoid misunderstandings in the theoretical section (qhCX, fYrs).
    - Added more detailed description of the hyperparameters of baseline methods (b3JB).
    - Simplified the description of the symexp reward loss (b3JB, fYrs).
    - Expanded on the discussion between model-based vs. model-free and the benefits of planning in current benchmarks (dWbu).
    - Replaced overloaded use of “generalist” (dWbu).
    - Included missing related work (fYrs).
- **Limitations**: Although the original draft included limitations, these were easily overlooked. Therefore, we’ve added an explicit limitations section to the conclusion to highlight limitations and remaining challenges for MR.Q (qhCX, fYrs).

We hope this addresses all of the primary concerns raised by the reviewers, and if not, we would be happy to engage in further discussion and revisions of the paper, to arrive at the best possible version of our work.

---

### Meta-Review · Area_Chair_BPUh · 2024-12-21

**Metareview:**

The paper introduces MR.Q, a model-free reinforcement learning algorithm leveraging model-based representations for general-purpose applications. It demonstrates competitive performance across a variety of RL benchmarks using a single set of hyperparameters, with extensive ablations supporting its design choices. Strengths include clear empirical results, hyperparameter robustness, and detailed ablations. However, limitations include the absence of evaluations on more diverse or challenging benchmarks like Minecraft or Procgen, reliance on well-studied benchmarks, and theoretical motivations being less innovative. Despite these, the paper provides significant empirical contributions, strong baselines, and thoughtful responses to reviewer concerns, warranting its acceptance.

**Additional Comments On Reviewer Discussion:**

During the rebuttal, reviewers raised concerns about hyperparameter tuning, missing baselines (e.g., PPO), theoretical clarity, and evaluation scope. The authors addressed these by clarifying hyperparameter procedures, adding PPO comparisons, improving theoretical explanations, and conducting additional ablations with statistical significance. They also acknowledged limitations in addressing sparse-reward tasks or broader POMDPs and added a dedicated limitations section. These responses resolved key concerns, demonstrated the robustness of MR.Q, and highlighted its practical contributions, leading to the decision to recommend acceptance.

---

### Decision · Program_Chairs · 2025-01-22

Accept (Spotlight)